# The allotetraploid horseradish genome provides insights into subgenome diversification and formation of critical traits

Fei Shen [1,8] ✉, Shixiao Xu [2,8], Qi Shen [3,4,5,8], Changwei Bi [6,8] & Martin A. Lysak [7] ✉

Polyploidization can provide a wealth of genetic variation for adaptive evolution and speciation, but understanding the mechanisms of subgenome evolution as well as its dynamics and ultimate consequences remains elusive. Here, we report the telomere-to-telomere (T2T) gap-free reference genome of allotetraploid horseradish (*Armoracia rusticana*) sequenced using a comprehensive strategy. The (epi)genomic architecture and 3D chromatin structure of the A and B subgenomes differ significantly, suggesting that both the dynamics of the dominant long terminal repeat retrotransposons and DNA methylation have played critical roles in subgenome diversification. Investigation of the genetic basis of biosynthesis of glucosinolates (GSLs) and horseradish peroxidases reveals both the important role of polyploidization and subgenome differentiation in shaping the key traits. Continuous duplication and divergence of essential genes of GSL biosynthesis (e.g., $FMO_{GS-OX}$, *IGMT*, and GH1 gene family) contribute to the broad GSL profile in horseradish. Overall, the T2T assembly of the allotetraploid horseradish genome expands our understanding of polyploid genome evolution and provides a fundamental genetic resource for breeding and genetic improvement of horseradish.

Recurrent polyploidizations or whole-genome duplications (WGDs) have been important drivers of genome evolution among angiosperms[1–3]. WGDs provide rich genetic material for subsequent mutations and lay the foundation for many diverse traits and overall adaptability. For example, the duplicated MADS-box gene family in angiosperms initiated by gamma whole-genome triplication (WGT-γ) is widely understood to have enabled adaptive evolution and species diversification[2,4]. The emergence of important agronomic traits in many crops, such as spinnable fibers in cotton and diversified morphotypes in *Brassica*, has been a consequence of polyploidization[5,6]. Throughout their evolution, allopolyploids undergo dramatic genomic changes to cope with the "genome shock" between the original hybridizing species[7,8]. The subgenome dominance hypothesis states that subgenome dominance can resolve various (epi)genetic conflicts in allopolyploids[8,9]. For example, fewer genomic rearrangements and reduced epigenetic changes in the dominant subgenome have been observed in *Brassica* and cotton allopolyploids[8,10]. However, understanding the mechanisms, dynamics, and ultimate consequences of subgenome evolution remains elusive.

[1]Institute of Biotechnology, Beijing Academy of Agriculture and Forestry Sciences, Beijing, China. [2]Tobacco College, Henan Agricultural University, Zhengzhou, Henan, China. [3]Genome Research Center, Leeuwenhoek Biotechnology Inc., Hong Kong, China. [4]Shangji Biotechnology Inc., Tianjin, China. [5]PheniX, Plant Phenomics Research Centre, Nanjing Agricultural University, Nanjing, China. [6]College of Information Science and Technology, Nanjing Forestry University, Nanjing, China. [7]Central European Institute of Technology and National Centre for Biomolecular Research, Faculty of Science, Masaryk University, Brno, Czech Republic. [8]These authors contributed equally: Fei Shen, Shixiao Xu, Qi Shen, Changwei Bi. ✉e-mail: shenf1028@gmail.com; martin.lysak@ceitec.muni.cz

The Brassicaceae (Cruciferae) is a family of economically important flowering plants that includes species such as rapeseed (*Brassica napus*), cabbage (*B. oleracea*), radish (*Raphanus sativus*), camelina (*Camelina sativa*), and the model organism *Arabidopsis thaliana*. The genus *Brassica* contains important oilseed and vegetable crops and serves as a model for the analysis of the immediate and long-term effects of polyploidization. The "U-triangle" model has been proposed to describe the origin of allotetraploid *Brassica* genomes, such as *B. napus* (AACC), *B. juncea* (AABB), and *B. carinata* (BBCC), from diploidized mesohexaploid genomes (AA, BB and CC)[11]. Both the "diploid" and allotetraploid *Brassica* genomes exhibit subgenome dominance, biased gene retention and differential expression of homoeologous genes[7,12,13].

Horseradish (*Armoracia rusticana*, $2n = 4x = 32$), which also belongs to the Brassicaceae, is a spicy root vegetable that is cultivated worldwide (Fig. 1a, b, Supplementary Fig. 1)[14]. It is highly adaptable, and can even become invasive. Like other cruciferous vegetables, horseradish contains glucosinolates (GSLs), which are enzymatically hydrolyzed by thioglucosidase enzymes to produce isothiocyanates with characteristic pungency[15,16]. Isothiocyanates have been extensively studied for their cancer-preventive properties[17–19]. The dominant GSL in horseradish is sinigrin (allylGSL). It is the precursor of allyl isothiocyanate, the main contributor to the pungent flavor of horseradish and wasabi (*Eutrema japonicum*). The thioglucosidase enzymes that catalyze the formation of isothiocyanate are traditionally termed myrosinases. Myrosinase is a functional term that encompasses several betaglucosidase enzymes capable of accepting thioglucosides as substrates. The group of genes initially identified in *A. thaliana*, referred to as TGGs[20], are commonly known as classical myrosinases[21]. Several other genes with myrosinase activity that were characterized later can be referred to as "atypical myrosinases" simply to indicate that they are less well known biochemically than the classical myrosinases[21]. Atypical myrosinases include the PEN enzymes[22].

The enzyme horseradish peroxidase (HRP) is widely used in molecular biology because it increases the detectability of target molecules[23]. Commercial HRPs are isolated from horseradish roots and consist of several isoenzymes, of which only a tiny fraction has been described (e.g., HRP C)[23,24]. Accordingly, horseradish breeding would benefit from a complete genome sequence that would enable the application of advanced breeding and genetic techniques, including genome editing[25], genome-wide selection[26,27], and molecular marker-assisted breeding[28]. However, limited genomic resources have prevented the application of advanced breeding techniques in horseradish and the elucidation of the genetic basis of its important traits.

Here, we generate the chromosome-level genome assembly of horseradish by combining data from Illumina, Oxford Nanopore Technology (ONT), PacBio HiFi, and Chromatin Conformation Capture (Hi-C) sequencing, corroborating its allotetraploid origin[14]. Dynamics of dominant long terminal repeat (LTR) retrotransposons and DNA methylation are found to play critical roles in subgenome differentiation. Our analysis of the genetic basis of key trait formation in horseradish (e.g., the synthesis of GSLs and HRPs) extends the current understanding of the evolution of allopolyploid genomes, including the effects of WGDs on trait formation and evolution.

## Results

### Genome sequencing reveals an allotetraploid origin of the horseradish genome

Horseradish was confirmed to be an allotetraploid ($2n = 4x = 32$) formed by two structurally nearly identical genomes[14]. We obtained the Illumina sequence data and performed a genome survey by *k*-mer analysis (Fig. 1c, d). The genome size was nearly 636 Mbp, of which 57% were repetitive sequences (Fig. 1c, d). The high proportion (~90.30%) of homozygous *k*-mers and the low proportion (9.70%) of heterozygous *k*-mers indicated high homoeology between the two ancestral

genomes (Fig. 1c, d). We distinguished between an allotetraploid versus autotetraploid WGD based on the different patterns of nucleotide heterozygosity as described in GenomeScope 2.0[29]. Allotetraploids would be expected to have a high proportion of aabb *k*-mers and a low proportion of aaab *k*-mers, as preferential pairing would ensure that two homoeologs from the first subgenome and two homoeologs from the second subgenome are present after recombination[29]. We found that 7.88% of *k*-mers were aabb and 1.79% were aaab (Fig. 1c). Thus, we concluded that the horseradish genome has an allotetraploid origin.

### A T2T gap-free horseradish reference genome

We generated a set of 27.71 Gbp (~43.57-fold genome coverage) HiFi reads, 52.28 Gbp (~82.20-fold coverage) ONT long reads, 55.99 Gbp (~88.03-fold coverage) Illumina reads, and 337.66 Gbp (~530.91-fold coverage) Hi-C data (Supplementary Table 1). To achieve a high-quality genome assembly, we used an integrated approach that combined data from multiple sequencing platforms.

We initially assembled the ONT reads into 292 contigs with an N50 of 7.95 Mbp and a total length of 610.85 Mbp, covering ~96.05% of the estimated genome size (Supplementary Table 2). To improve the single-base accuracy of the genome assembly, we polished the genome assembly with ONT and Illumina reads and finally achieved a quality value (QV) of 39.5. To overcome challenges such as interference between homoeologous subgenomes and chimeric assembly, we adopted a comprehensive scaffolding strategy (Supplementary Fig. 2, Supplementary Note 1). The assembled contigs were organized into 16 pseudochromosomes, which accounted for 94.80% of the total assembled sequence (Supplementary Tables 3 and 4, Supplementary Fig. 3). We distinguished between the two subgenomes (A and B) based on the orthologous genes and subgenome-specific *k*-mers using SubPhaser[30] (Supplementary Figs. 4, 5).

For the PacBio HiFi reads, our preliminary assembly generated 108 contigs with an N50 length of 33.74 Mbp (Supplementary Table 5). Owing to the accuracy of long HiFi reads and the high continuity of the assembly, we performed scaffolding directly and generated a genome assembly with 16 pseudochromosomes that anchored 97.29% of the assembled sequence, leaving only 8 gaps in the genome assembly (Supplementary Table 6, Supplementary Note 2). Comparison of the assembled pseudochromosomes with the published karyotype of horseradish revealed a possible loss of blocks F and R in the ONT-based genome (Supplementary Fig. 6). However, both genomic blocks were confirmed by comparative chromosome painting and also identified in the HiFi-based genome assembly (Supplementary Figs. 7–9, Supplementary Note 1). Considering the superior continuity and integrity of the HiFi-based genome, we selected it as the genome backbone and filled the remaining gaps using the ONT-based genome. This resulted in a gap-free reference genome consisting of 16 pseudochromosomes with a total length of 610.05 Mbp (Supplementary Table 6).

Using the seven base telomere repeats ('CCCTAAA/TTTAGGG') as a sequence query, we identified 31 telomeres (15 pairs plus one singleton) and constructed 15 T2T pseudomolecules (Supplementary Table 7). The approximate location of 16 centromeric regions was estimated from the density of repeats and the Hi-C interaction heatmap. Centromeric regions range from 2.8 to 18.5 Mbp with an average length of 5.26 Mbp (Supplementary Table 8, Supplementary Figs. 10–13).

The horseradish genome assembly exhibits high gene completeness, with 99.44% of genes identified in the Benchmarking Universal Single-Copy Orthologs (BUSCO) dataset (Supplementary Table 9). The mapping rates of Illumina, HiFi, and ONT reads were over 99.40%, 99.91%, and 99.50%, respectively, while an average of 96.94% of RNA-seq reads could be mapped to the genome (Supplementary Table 10). The long terminal repeat retrotransposons (LTR-RTs) Assembly Index (LAI) reached 25, indicating high assembly integrity even in repeat-rich

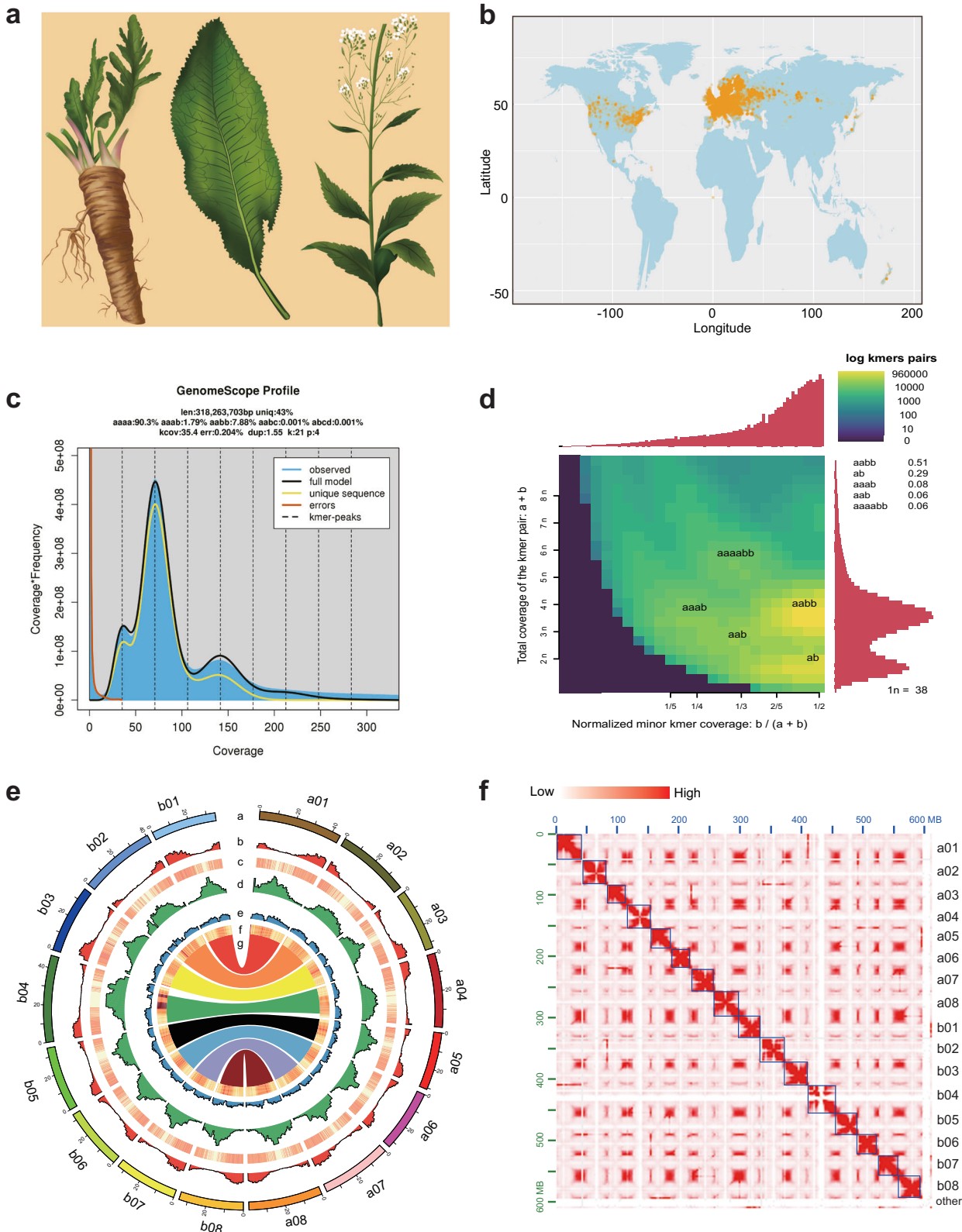

regions. Overall, the T2T gap-free genome assembly of horseradish showed exceptional quality.

## High-quality genome annotation revealed the architecture of the A and B subgenomes

A total of 42,025 protein-coding genes were predicted in the horseradish genome, achieving a BUSCO completeness of 98.33%

(Supplementary Tables 11, 12). Of these genes, 99.60% (41,855) could be functionally annotated or were inferred to be expressed (Supplementary Table 11). The number of genes in the two subgenomes were comparable (Supplementary Table 13). In addition, 1,143 tRNA, 2,600 snoRNA, 557 miRNA, and 195 spliceosomal RNA genes were identified as parts of the non-coding RNA repertoire (Supplementary Table 14).

**Fig. 1 | Genome assembly of *Armoracia rusticana* (horseradish). a** Morphology of horseradish. **b** The world-wide distribution of horseradish. (The source data were adopted from Global Biodiversity Information Facility, https://www.gbif.org/). **c**–**d** *k*-mer spectra and Smudgeplots for the horseradish genome. The genome survey in (**c**) was generated using GenomeScope 2.0. The length of the haploid genome (len) and the percentage of unique sequences (uniq) were 318 Mbp and 43%, respectively. Different types of *k*-mers are indicated as aaaa, aaab, aabb, aabc, and aacd in (**d**). **e** Circular diagram of major genomic characteristics: (a) chromosome, (b) gene density (a 1-Mbp window size), (c) gene expression (genes with higher values are represented by a darker color), (d) long terminal repeat retrotransposon density (a 1-Mbp window size), (e) DNA transposon density (1-Mbp window size). (f) GC content (a 1-Mbp window size; genomic regions with higher values are represented by a darker color), g. pairwise collinear segments. **f** Hi–C interactions between the eight chromosomes of subgenome A (a01–a08) and eight chromosomes of subgenome B (b01–b08) (resolution 500-kbp). Source data are provided as a Source Data file.

A combination of de novo and homoeology-based methods was used to annotate repetitive sequences in the horseradish genome. The proportion of repetitive sequences in the total assembly was 393.26 Mbp, representing 64.44% of the genome (Supplementary Table 15). Retrotransposons were identified as the most abundant repetitive elements, accounting for 69.19% of all repetitive sequences and 44.59% of the entire genome assembly (Fig. 2a, Supplementary Table 15). DNA transposons accounted for 15.28% of the genome (Fig. 2a, Supplementary Table 15). Both the A and B subgenomes had similar proportion of repeats, with -58.66% (-175.07 Mbp) in the A subgenome and -59.99% (-177.03 Mbp) in the B subgenome, including all types of repetitive elements (Fig. 2b, Supplementary Data 1).

Due to the abundance of LTR-RTs in the horseradish genome, we investigated the dynamics of LTR-RTs during horseradish genome evolution. A total of 11,282 intact retrotransposons were identified in the horseradish genome with similar distribution in the two subgenomes (Supplementary Table 16, Supplementary Data 2, Supplementary Note 3). Analysis of the LTR-RT insertion timing revealed that bursts of different LTR-RT types occurred approximately 0.2 million years ago (mya) (Fig. 2c). Phylogenetic analysis revealed four lineages of Ty1/*Copia* retrotransposons (*Ale, Ivana, Maximus,* and *Tork*) and five lineages of Ty3/*Gypsy* retrotransposons (*Athila, CRM, Galadriel, Reina,* and *Tekay*) (Fig. 2d, Supplementary Fig. 14, Supplementary Note 4). The most abundant Ty1/*Copia* retrotransposon was *Ale* (51.4%), followed by *Tork* (35.16%), *Maximus* (10.83%), and *Ivana* (2.60%). As for the Ty3/*Gypsy* retrotransposons, the *CRM* clade occupied 54.93%, followed by *Tekay* (18.53%), *Athila* (16.51%), *Reina* (6.38%), and *Galadriel* (3.65%) clade. We further categorized full-length LTR-RTs into 1949 families, revealing uneven distribution among these families (Supplementary Data 3, Supplementary Note 4). More than 50% of full-length LTR-RTs were found among the top 67 families, indicating their dominant contribution to the LTR-RT population (Supplementary Data 3). Notably, the largest family, FAM961, included 510 members classified as 'unknown' LTR-RTs (Supplementary Data 3).

Using RepeatMasker and a custom library of clustered LTR-RT family sequences, we analyzed the coverage of each LTR-RT family in the horseradish genome. Our results showed that the horseradish repeatome is dominated by specific LTR-RT families with exceptionally high genome coverage (Supplementary Data 3, Supplementary Fig. 15, Supplementary Note 4). Remarkably, the top 150 LTR-RT families ranked by genome coverage accounted for nearly 50% of the total LTR-RT genome coverage (Supplementary Data 3). Among these families, FAM1, classified as 'unknown' LTR-RTs, had the highest coverage (-2.88%) (Supplementary Data 3).

We further explored the potential influence of LTR-RTs on nearby genes (Supplementary Note 5–7). Our analysis identified nearly 1,497 genes located in close proximity (<1000 bp) to intact *Gypsy*/*Copia* LTR-RTs (Supplementary Fig. 16a, b, Supplementary Note 5). The presence of nearby LTR-RTs, particularly *Gypsy* RTs, had noticeable effects on gene expression and methylation levels (Supplementary Fig. 16c, f, Supplementary Note 5). Kyoto Encyclopedia of Genes and Genomes (KEGG) enrichment analysis revealed that the potentially affected genes are associated with vital metabolic pathways/BRITE hierarchies such as glutathione metabolism, starch and sucrose metabolism, fatty acid degradation, and lipid metabolism (Supplementary Fig. 16h, i, Supplementary Note 5). We therefore propose that LTR-RTs may

negatively affect the expression of nearby genes by elevated methylation levels and thus some biological processes.

## Centromeric regions of the horseradish genome

Plant centromeres are usually composed of repetitive sequences, including LTR-RTs and tandem repeats[31]. We combined the Hi-C interaction heatmap and repeat distributions to predict centromeric regions (Supplementary Note 8). Remarkably, the "blank regions" observed in the Hi-C interaction map exhibited a dense accumulation of tandem repeats, particularly in the centromeric regions, consistent with T2T genome assemblies of kiwifruit[32] and faba bean genomes[33] (Fig. 2e, f, Supplementary Figs. 10–13, Supplementary Note 8). Surprisingly, chromosome b04 had a remarkably long centromeric region (-18.5 Mbp) containing megabase-long islands of four highly abundant tandem repeats (Supplementary Data 4, Supplementary Fig. 12). In contrast, we found neither a comparable super-long centromeric region nor such highly abundant tandem repeats on chromosome a04, indicating a stark divergence of the homoeologous centromeric regions (Fig. 2e, f, Supplementary Fig. 12). Visualization of the complex tandem repeats within the centromeric regions revealed their distinct high-order structure in the two subgenomes (Fig. 2g, Supplementary Figs. 17–19).

In our analysis of centromeric regions (Supplementary Note 8), we identified five LTR-RT families of high abundances, classified as *Gypsy-CRM, Copia-Ale,* or 'unknown' elements (Supplementary Data 5). Some LTR-RT families were shared among different chromosomes (Supplementary Data 5). For example, the FAM7 family (classified as 'unknown') occurred on 12 different chromosomes, whereas the FAM13 family was found on seven different chromosomes (Supplementary Data 5). This suggests that these LTR-RT families have undergone amplification and spread to multiple centromeric regions. Interestingly, the FAM1 family ('unknown' classification) occupied nearly 36% of the 18.5-Mbp centromeric region on chromosome b04 (Supplementary Data 5). Hence, both the high density of tandem repeats and the FAM1 LTR-RT family contributed to the centromere expansion on chromosome b04.

## Polyploidization and gene family evolution

To elucidate the process of horseradish genome evolution, we performed a systematic phylogenetic analysis of 18 representative flowering plant species, including 11 species representing the major Brassicaceae lineages (i.e., Lineage I (Cardamineae), II (Brassiceae), III, IV, and Aethionemeae). Based on the phylogenetic tree constructed using the 378 single-copy genes and the estimated divergence time, we found that *Armoracia* diverged from the other Cardamineae species -5 mya (Fig. 3a). The *A. thaliana* genome as well as the most common ancestor of Cardamineae underwent three rounds of WGD, including the whole-genome triplication of eudicots (WGT-$\gamma$), the At-$\beta$ WGD specific to the core Brassicales[34], and the At-$\alpha$ WGD specific to the Brassicaceae[35]. We calculated the $K_s$ value distributions of the different species and horseradish subgenomes using homoeologous genes in the conserved collinear blocks. In the horseradish genome, there was a more recent $K_s$ peak ($K_s$: 0.12–0.13) than those identified in both subgenomes and shared with the *A. thaliana* genome ($K_s$: 0.75–0.81) (Fig. 3b). In addition, pairwise synteny analysis revealed strong macrosynteny between subgenomes A and B, indicating a close

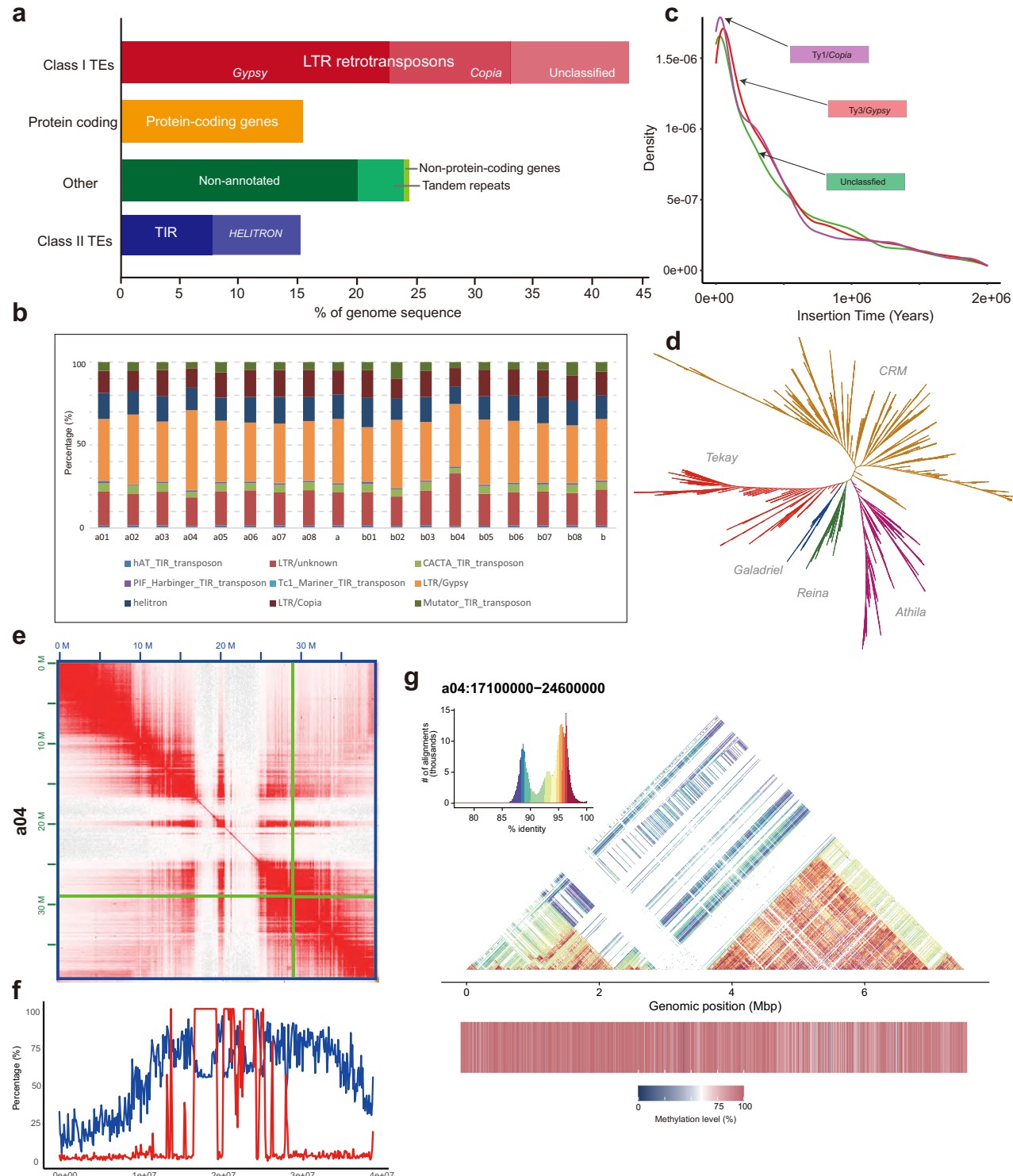

**Fig. 2 | Genomic architecture and the diversity of repetitive sequences in the horseradish genome. a** Summary of the genomic annotation. The "Other" genomic composition includes the non-protein-coding genes, unclassified repeat regions ("No repeat category"), and genomic regions that cannot be annotated as genes/repeats/non-protein-coding genes ("Non-annotated"). **b** Proportion of different repeat types in the 16 horseradish chromosomes (a01–a08, b01–b08) and both subgenomes (a and b columns, respectively). **c** The insertion times of different long terminal repeat retrotransposons (LTR-RTs). **d** Phylogenetic tree of Ty3/*Gypsy* type LTR-RTs. **e** Hi–C interactions in chromosome a04. The boundaries of contigs and scaffold are indicated by green and blue lines, respectively. **f** The density of LTR-RTs (blue) and tandem repeats (red) within 100-kbp window size along chromosome a04. **g** The structure of the centromeric region of chromosome a04. A 500-bp window was used to generate the sequence identify heatmap; methylation levels within each window are also shown. Source data are provided as a Source Data file.

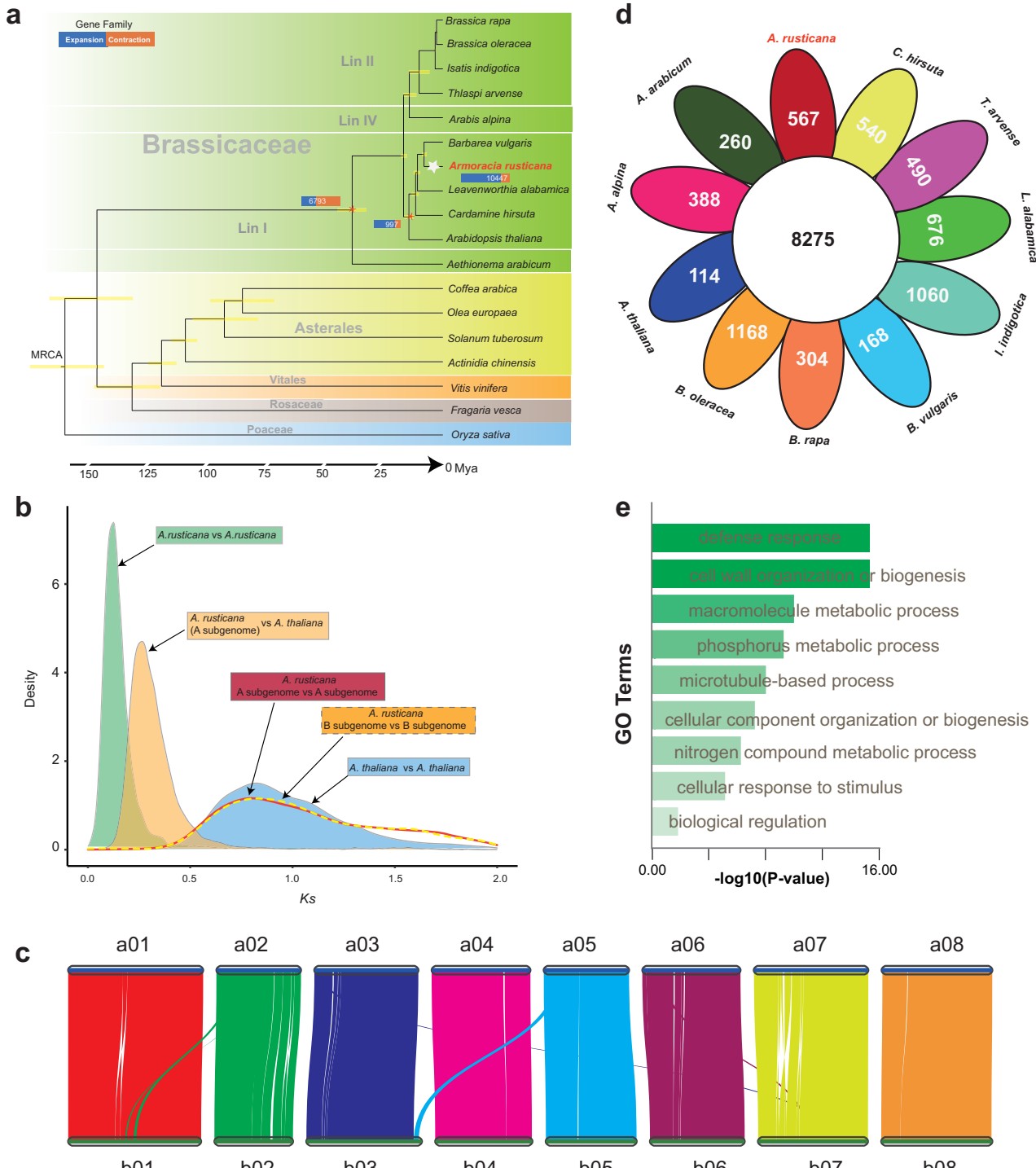

**Fig. 3 | Evolution of the horseradish genome. a** Estimation of divergence times and rapidly evolving gene families. **b** $K_s$ density profiles of *Armoracia rusticana* vs *Armoracia rusticana*, *Armoracia rusticana* (A subgenome) vs *Arabidopsis thaliana*, *Armoracia rusticana* B subgenome vs B subgenome, *Armoracia rusticana* A subgenome vs A subgenome, and *Arabidopsis thaliana* vs *Arabidopsis thaliana*. **c** Macrosynteny visualization of A and B subgenomes. **d** Venn diagram of orthogroups in 11 Brassicaceae genomes. **e** Gene Ontology (GO) enrichment analysis of the significantly expanded gene families in the horseradish genome. A one-sided Fisher's exact test was adopted and adjustments were made for multiple comparisons with Benjamini and Hochberg method. Source data are provided as a Source Data file.

relationship between the parental genomes (Fig. 3c). However, large-scale genomic rearrangements could also be inferred (Fig. 3c, Supplementary Fig. 20), suggesting that large-scale changes followed the polyploidization event or they differentiated the parental genomes.

We classified all 664,387 genes into 35,190 orthologous groups, including 5,409 groups with genes from all 18 species, 12,029 species-specific groups, and 21 single-copy groups (Supplementary Table 17). Among these, we identified 8,275 gene families (orthologous groups) shared by all 11 Brassicaceae species and 567 gene families specific to horseradish (Fig. 3d). To investigate changes in gene family size during evolution, we used the Computational Analysis of Gene Family Evolution (CAFE) software. A total of 10,447 gene families were identified as

changed rapidly at the horseradish node (8,859 expansions and 1,588 contractions) (Fig. 3a), whereby the expanded genes were significantly enriched in Gene Ontology (GO) terms of cell wall organization or biogenesis, defense response, biological regulation, and response to stimuli ($P < 0.01$) (Fig. 3e, Supplementary Data 6).

Comparative chromosome painting and whole-genome sequencing demonstrated that the genomes of *Cardamine* species (tribe Cardamineae) share structurally conserved chromosomes with the Ancestral Crucifer Karyotype (ACK, $n = 8$) but also contain two translocation chromosomes[36–38]. Here, using the high-quality genome assembly, we observed six ancestral chromosomes (AK1, AK2, AK3, AK4, AK5, and AK7) in both subgenomes (Supplementary Figs. 9, 21–23). Two chromosomes (a/b06 and a/b08) originated through a reciprocal translocation involving ancestral chromosomes AK6 and AK8, consistent with the previously published cytogenomic map[14] (Supplementary Fig. 23).

### Epigenetic modifications in the horseradish genome

Epigenetic modification by DNA methylation is conserved and essential for gene regulation and genome stability[39]. We used whole-genome bisulfite sequencing (WGBS) and ONT sequencing to generate genome-wide methylation profiles of different tissues (root, stem, and leaf). We found a high concordance between the two strategies in estimating of overall methylation levels and in generating methylation profiling around genes and LTR-RTs (Figs. 4a–i, Supplementary Fig. 24). WGBS provided higher resolution due to the identification of a greater number of methylated cytosines (Supplementary Data 7). The weighted methylation level of the whole genome in the cytosine context of CpG, CHG and CHH was 64.91%, 32.42%, and 11.82%, respectively (Supplementary Data 7, Supplementary Fig. 25a). Most methylated cytosines were distributed in the CHH context (46.93%–59.44%), followed by CpG (22.61%–31.51%) and CHG (17.95%–21.56%) (Supplementary Fig. 24b). We observed only minor differences in methylation levels between the two horseradish subgenomes (Supplementary Fig. 25a, Supplementary Data 7).

Analyzing methylation patterns around gene regions revealed similar profiles between the two subgenomes, despite differences in cytosine contexts and tissues (Fig. 4d–f, Supplementary Fig. 26). When we examined methylation levels of all genes, we found that only about 4% (1,702–1,749) of genes had extremely high methylation levels (>90%), whereas nearly 14% (5,610–5,838) of genes had relatively low methylation levels (<1%) (Fig. 4a–c, Supplementary Fig. 27). Hypermethylated genes were involved in various biological functions and were significantly enriched for thiamine metabolism, sesquiterpenoid biosynthesis, defense system, isoflavonoid biosynthesis, and fructose and mannose metabolism (Fig. 4j). In contrast, the hypo-methylated genes were significantly enriched in the KEGG metabolic pathways/ BRITE hierarchies of plant hormone signal transduction, monobactam biosynthesis, and transcription factors (Fig. 4j). Similar proportion of hyper- and hypo-methylated genes were observed both subgenomes (Fig. 4a–c, Supplementary Fig. 27).

We observed a strong negative correlation between the methylation level around the genic regions and gene expression ($P < 0.01$), indicating a strong influence of methylation on gene expression (Supplementary Fig. 28). We found that LTR-RT regions exhibited high levels of methylation, with about 90% methylation within these regions (Fig. 4g–i). *Gypsy* elements showed higher level of methylation compared to other LTR-RT families in both subgenomes (Fig. 4g–i, Supplementary Figs. 29–31). Further examination of the methylation levels of fragmented LTR-RTs revealed that these regions have high methylation level even when not actively transposing (Supplementary Figs. 32–34, Supplementary Note 6–7). This suggests that DNA methylation plays a significant role in regulating and maintaining the stability of LTR-RT regions in the horseradish genome. The intricate relationship between LTR-RTs and DNA methylation highlights their joint roles in the dynamic regulation of the horseradish genome.

To identify the difference between the two horseradish subgenomes, we compared the methylation levels of homoeologous gene pairs located in collinear blocks. We defined a gene pair as differentially methylated if at least one gene belonged to hyper- or hypo-methylated genes and the ratio of methylation levels was >2. Among the 23,790 homoeologous gene pairs, we identified 4,483 gene pairs that had differential methylation, with approximately 48.05% (2,154) having a higher level of methylation in the B subgenome (Supplementary Fig. 35a, Supplementary Data 8). These differentially methylated genes were enriched in various biological processes, including starch and sucrose metabolism, as well as plant hormone signal transduction (Supplementary Fig. 35c, d).

Gene expression of the differentially methylated gene pairs showed high concordance with the methylation level, confirming that DNA methylation is associated with differential expression (Supplementary Fig. 35b). We observed that nearly 24% of differentially methylated genes were associated with LTR-RTs (Supplementary Fig. 34, Supplementary Note 7). For example, for one orthologous gene pair (*Arr6274* and *Arr12787*), we observed that one *Gypsy* LTR-RT element was present upstream (−387 bp of *Arr12787*) in the B subgenome and gene expression decreased with a significant increase in methylation in all tissues (Fig. 4k). These results highlight the significant impact of LTR-RTs on subgenomes diversification and the establishment of subgenome-specific methylation profiles. These observations contribute to our understanding of the complex mechanisms underlying genome organization and the influence of transposable elements on genomic diversity.

### 3D chromatin structure of the horseradish genome

Hi-C technology can be used to quantify long-range physical interactions within a nuclear genome[40]. The genome is divided into A/B compartments, and interactions between loci occur largely within the same compartment[40]. The A compartment is always associated with open chromatin, whereas the B compartment is always associated with closed chromatin[40]. We classified 40.78% of the genome as belonging to the A compartment (Supplementary Table 18). In the A subgenome, 39.84% of genomic regions belonged to the A compartment, whereas in the B subgenome, 41.73% belonged to the A compartment (Fig. 5a, Supplementary Table 18). The A compartments were primarily located at chromosome arms, whereas the B compartments were more prevalent near (peri)centromeric regions. We observed quite different genomic features of A and B compartments. The higher gene density, lower transposon element content, and lower methylation levels of the A compartments suggest that chromatin structures have a crucial influence on genomic features and function (Fig. 5b–d). The A/B compartments can be further subdivided into smaller topologically associated domains (TADs). Our analysis identified a total of 3635 TADs (1877 in the A subgenome and 1752 in the B subgenome), with over 75% of TADs being <1 Mbp (Fig. 5e). We analyzed epigenetic modifications around TAD boundaries and found that chromatin surrounding TAD boundaries had relatively lower DNA methylation levels in both subgenomes, suggesting that epigenetic modifications may contribute to the differential activation of genes positioned at TAD boundaries (Fig. 5f–h). We examined TAD boundaries within syntenic genomic blocks. Only about 30% of TAD boundaries were conserved between the two subgenomes (Supplementary Fig. 36). For example, TAD boundaries differed within the syntenic blocks on chromosomes a03 and b03 (Fig. 5j). Long-range chromatin interactions functionally contribute to transcriptional regulation. We identified 183,994 chromatin loops (ranging from 20 kbp to 1 Mbp) in the horseradish genome, of which 51.43% belonged to the A subgenome (Fig. 5i). These interactions were classified into three groups: Genic-Genic (G-G), Intergenic-Genic (I-G), and Intergenic-Intergenic (I-I). Approximately

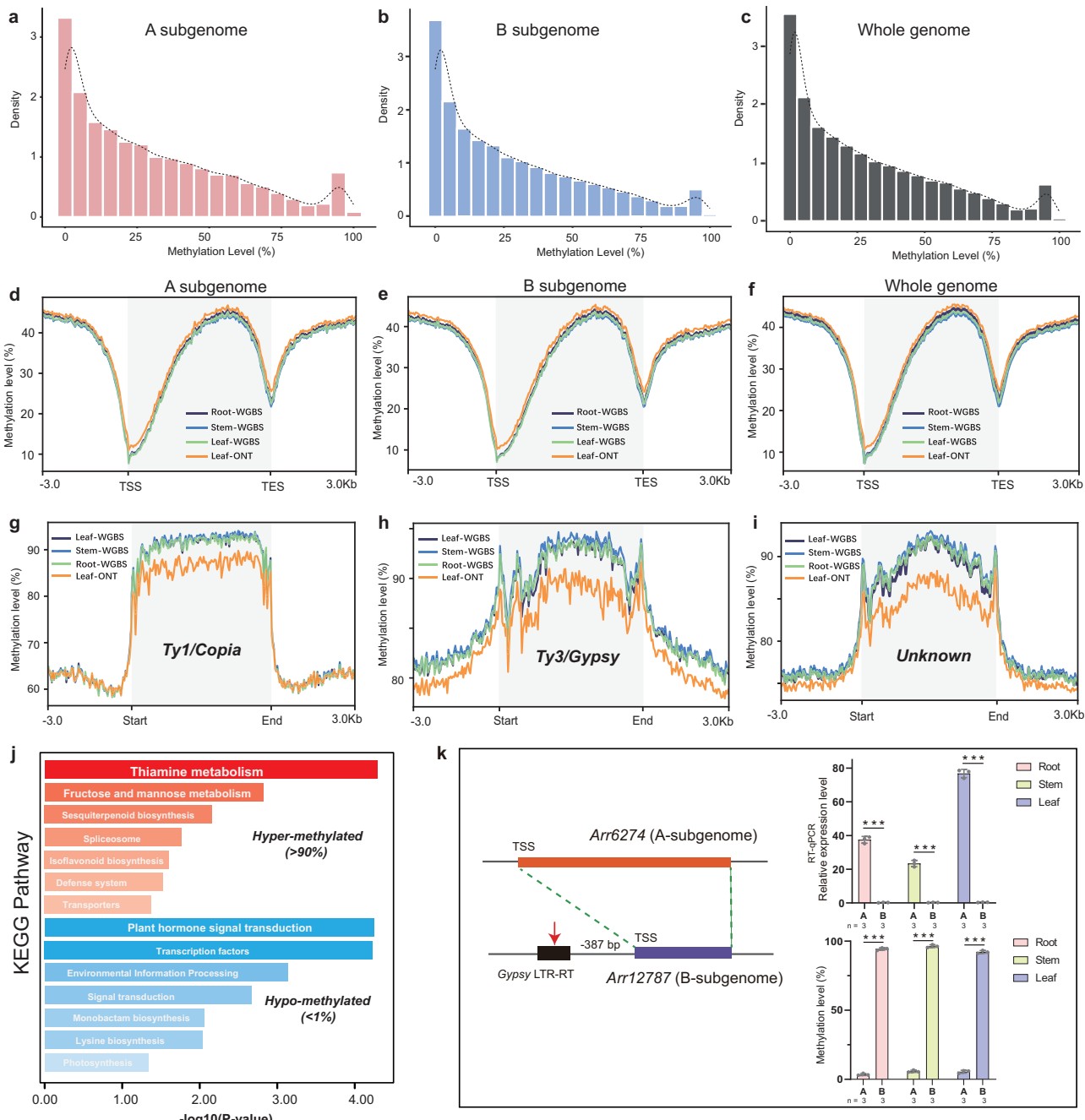

**Fig. 4 | Whole-genome methylation profiling in the horseradish genome.**
**a**–**c** The distribution of average methylation levels around genes in the A sub-genome (**a**), the B subgenome (**b**), and in the whole genome (**c**). **d**–**f** The average methylation levels around genes in the A subgenome (**d**), the B subgenome (**e**), and in the whole genome (**f**). Genic regions were shaded in gray. TSS: transcription start site. TES: transcription end site. WGBS: whole-genome bisulfite sequencing. ONT: Oxford Nanopore Technologies. **g**–**i** The average methylation levels around different types of long terminal repeat retrotransposons (LTR-RTs), such as Ty1/*Copia* (**g**), Ty3/*Gypsy* (**h**), and unclassified elements (**i**). LTR-RT regions are shaded gray.

**j** Kyoto Encyclopedia of Genes and Genomes (KEGG) enrichment analysis of the hypermethylated gene set (methylation level >90%) and hypo-methylated gene set (methylation level < 1%). A one-sided Fisher's exact test was adopted and adjustments were made for multiple comparisons with Benjamini and Hochberg method. **k** The effect of a *Gypsy* LTR-RT on the methylation level and gene expression of *Arr12787* (B subgenome). The homoeologous gene *Arr6274 (*A subgenome) was used as a reference. The number (*n*) of biologically independent samples is shown below. Data are shown as mean ± SD. Significance was tested with two-sided Student's *t*-tests. ***P < 0.001. Source data are provided as a Source Data file.

68.17% (125,437) of these interactions were I–I interactions, followed by I–G (-21.24%; 39,076) and G–G interactions (-10.59%; 19,481) (Fig. 5i). Analysis of genes involved in I-G or G-G interactions revealed enrichment in Gene Ontology (GO) terms and KEGG pathways related to essential and fundamental biological processes, such as chromosome organization, messenger/transfer RNA degradation, DNA repair, and recombination, highlighting the significant impact of chromatin loops on biological processes (Fig. 5k).

## Subgenome dominance in the horseradish genome

We performed a comprehensive analysis to investigate the possible subgenome dominance at different levels. Among the 23,790 homoeologous gene pairs located in collinear blocks, we identified 2,214 gene pairs with a higher homoeolog expression bias (HEB, HEB > 2) in all tissues (Fig. 6a, Supplementary Data 9). More specifically, 1,133 genes belonged to the A subgenome, whereas 1,081 genes belonged to the B subgenome. These gene pairs consistently had higher expression

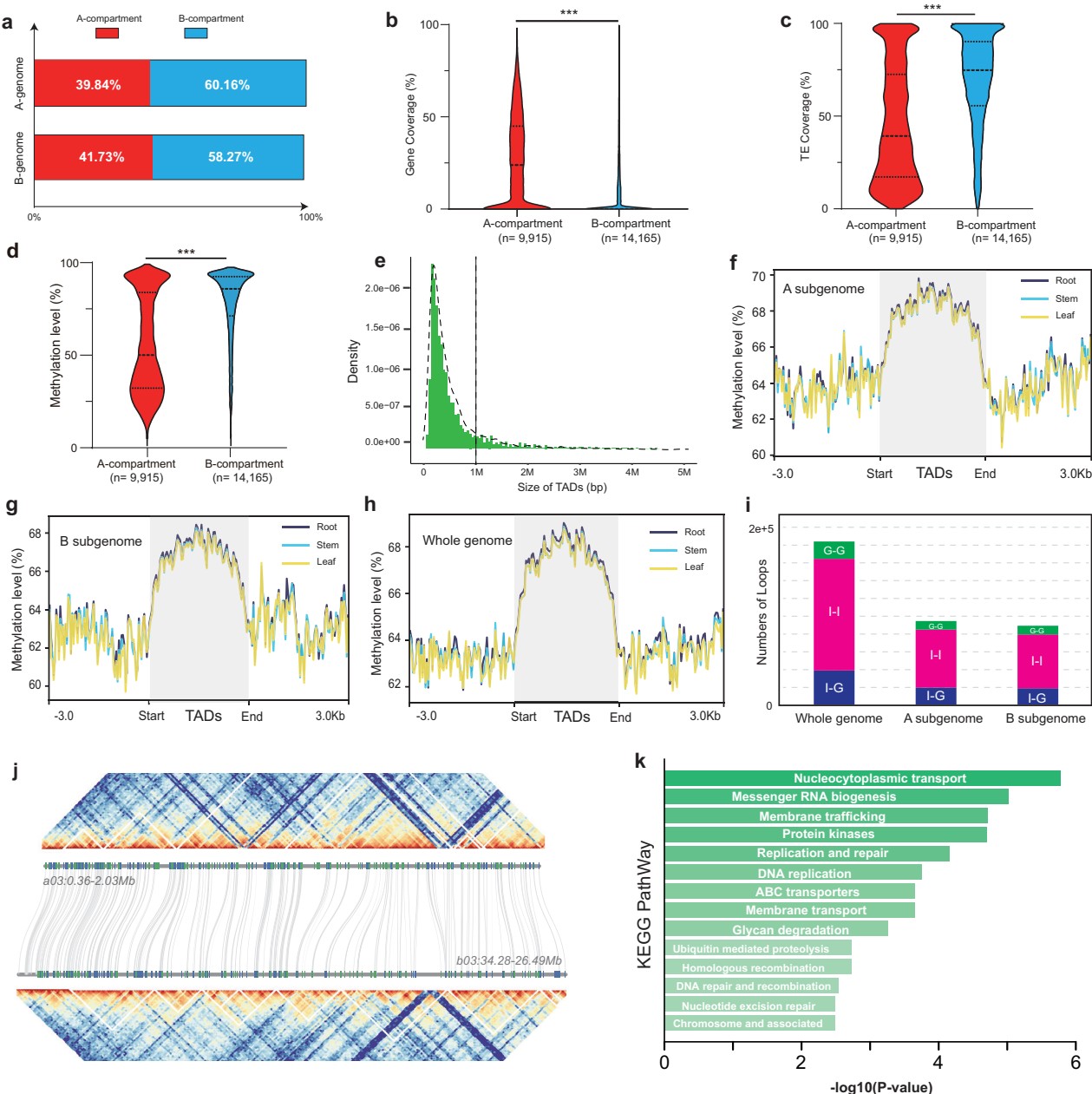

**Fig. 5 | Three-dimensional (3D) chromatin structure of the horseradish genome. a** Proportion of A- and B- compartments in both subgenomes. **b**–**d** (Epi) genomic characteristics of A- and B-compartments, including gene coverage (**b**), transposable element (TE) coverage (**c**), and methylation level (**d**). The number (*n*) of data points for A- and B-compartments is given below. In violin plots, central line: median values; the other two horizontal lines: 25th and 75th percentiles. Significance was tested with two-sided Student's *t*-tests. ***$P < 0.001$. **e** Length distribution of topologically associated domains (TADs). **f**–**h** Methylation level profile around TADs in the A subgenome (**f**), the B subgenome (**g**), and in the whole

genome (**h**). TAD regions are shaded gray. **i** Statistical summary of different types of chromatin loops across the genome. Interactions between chromatin loops were classified into the following three groups: Genic-Genic (G–G), Intergenic–Genic (I–G), and Intergenic–Intergenic (I–I). **j** Microsynteny between chromosomes a03 (A subgenome) and b03 (B subgenome) and identified TADs (white triangles). **k** Kyoto Encyclopedia of Genes and Genomes (KEGG) pathways enriched for chromatin-loop-related genes. A one-sided Fisher's exact test was adopted and adjustments were made for multiple comparisons with Benjamini and Hochberg method. Source data are provided as a Source Data file.

levels in their respective subgenomes (Fig. 6b, Supplementary Data 9). The homoeologs with higher HEB in the A subgenome were highly enriched in organonitrogen compounds, response to biotic stimuli, and defense response (Fig. 6c). In contrast, homoeologs with higher HEB in the B subgenome were significantly enriched in GO terms of defense response, metabolism of sulfur compound, pollination and catabolism of small molecules (Fig. 6c). The above results indicate that different subgenomes may play different roles in biological processes and trait determination.

We also assessed the average $K_a/K_s$ values for genes with higher (dominant) expression in the A subgenome, lower (subordinate) expression in the A subgenome, and neutral expression (i.e., equal expression in both subgenomes). The dominant and subordinate genes had significantly higher $K_a/K_s$ values (median 0.346 and 0.345, respectively) than the neutral genes (median 0.269) (Supplementary Fig. 37a). Average $K_a$ values also followed the same trend as $K_a/K_s$ values (Supplementary Fig. 37b), indicating that dominant and subordinate genes evolved faster than neutral genes.

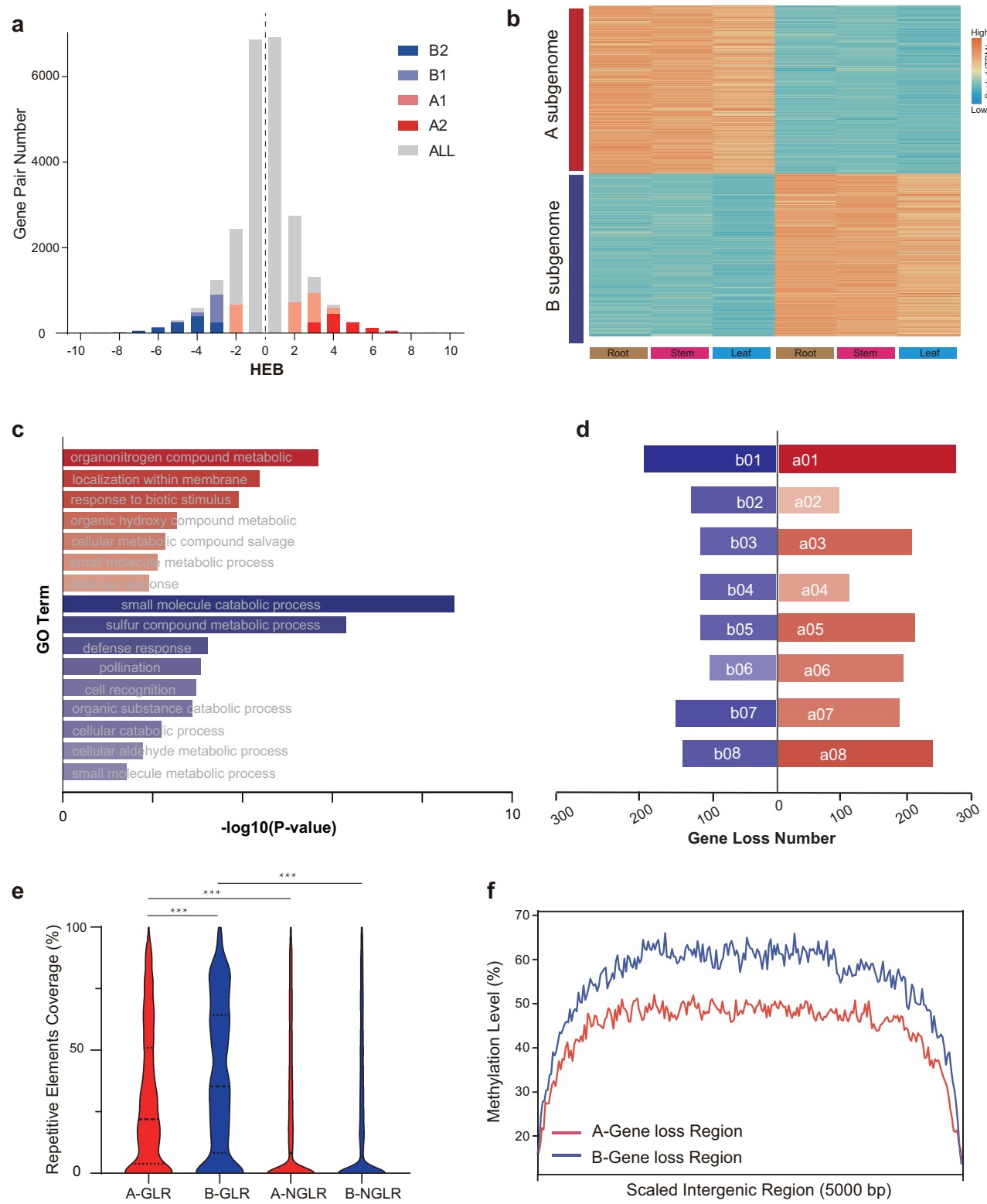

In addition to analyzing differential expression of homoeologs, we examined the possible loss of genes during the evolution of the horseradish genome (Supplementary Fig. 38). Within the conserved collinear genomic regions, a total of 2,653 genes were predictably lost, with 1,563 genes lost in the A subgenome and 1,090 genes lost in the B subgenome. The presumably lost genes showed uneven distribution across both subgenomes (Fig. 6d, Supplementary Fig. 39). The lost genes in the B subgenome were enriched in biological processes related to stress response, response to stimuli, and DNA repair (Supplementary Fig. 40). In contrast, the lost genes in the A subgenome contained more members with essential biological functions such as regulation of molecular functions, homeostatic processes, and system development, suggesting that the different patterns of gene loss played a

**Fig. 6 | Subgenome dominance in the horseradish genome. a** Expression bias between homoeologs in the allopolyploid genome. The gray histogram bars show the distribution of homoeolog expression bias (HEB) for all testable homoeolog pairs. Testable homoeolog pairs (N) are those that could be identified as homoeologous and had at least one read in each tissue sample. Homoeolog pairs significantly biased toward the A-subgenome homoeolog are shown in red, while pairs extremely biased toward the "other" homoeolog are shown in blue. A1 (B1) in lighter color indicates gene pairs with log$_2$(fold change) >1, and A2 (B2) in darker color indicates gene pairs with log$_2$(fold change) >2. **b** Gene expression heatmap of gene pairs with significant HEB. **c** Enriched Gene Ontology (GO) terms among the homoeolog pairs that are significantly biased toward the A (red) or B (blue)

subgenome homoeolog. A one-sided Fisher's exact test was adopted and adjustments were made for multiple comparisons with Benjamini and Hochberg method. **d** The number of predicted lost genes in the A (red) or B (blue) subgenome since the allotetraploidization event. **e** Repetitive element coverage of genomic regions with gene loss (GLR) and regions without gene loss (NGLR) in each subgenome. The number (n) of data points for A- and B-compartments is shown below. In violin plots, central line: median values; other two horizontal lines: 25th and 75th percentiles. Significance was tested with two-sided Student's $t$-tests. ***$P$ < 0.001. **f** Methylation level profiles within genomic regions with gene loss in A and B subgenomes. Source data are provided as a Source Data file.

role in the inferred differential subgenome evolution (Supplementary Fig. 40).

We performed genomic content analysis around the lost genes in the conserved collinear genomic regions. We observed that the presence of interspersed repetitive elements was significantly higher in the gene loss regions (GLRs) (Fig. 6e). In addition, the average methylation of GLRs was significantly higher (Fig. 6f). These findings suggest that the interspersed repetitive elements may have a crucial influence on methylation level, gene expression, and gene loss.

## Both subgenomes contribute to the diversity of horseradish peroxidases

The complete genome sequence enables the identification of genes encoding all horseradish peroxidases. We identified 105 *PER*s (*Peroxidases*) in the horseradish genome, of which 53 were found in the A subgenome and 52 in the B subgenome (Fig. 7a, Supplementary Data 10). Their isoelectric points and molecular weights were 4.43 to 11.09 and 27.09 to 56.86 kDa, respectively (Fig. 7b–d). The vast majority of HRP research has focused on one isoenzyme, C1A, in addition to isoenzymes E1–E6 and five isoenzymes B1-B3, C1, and C2 that have been biochemically characterized[23,24] (Fig. 7a). We constructed a phylogenetic tree of PERs from horseradish, *A. thaliana*, and the previously reported isoenzymes. Phylogenetic analysis retrieved six clades (Fig. 7a). Clade I contained isoenzymes C, B, and E, with most genes (17 of 20) expressed in roots and exhibiting high expression levels (Fig. 7e–g, Supplementary Data 10). Clade VI included the largest number of PERs (30), clustered with six proteins involved in lignin biosynthesis in *A. thaliana* (Fig. 7a, Supplementary Data 10). Among the 63 expressed genes encoding isoenzymes in roots, clades I and VI had a higher proportion of highly expressed genes (Supplementary Data 10). Consistent with gene expression levels, DNA methylation levels of genes were lower in clades I and VI (Fig. 7e–f, Supplementary Data 10). We examined the differential evolution of the *PER*s between subgenomes. A total of 85% (90) of the *PER* genes have homoeologs in the counterpart subgenome, whereas the counterparts of 15 *PER* genes were not identified as typical *PER*s due to differences between the subgenomes. The apparent low $K_a/K_s$ values between homoeologs indicate the conservation of the gene family (Fig. 7d). Conserved motifs were identified among the proteins, particularly in critical structural-functional regions (Fig. 7h, Supplementary Fig. 41). The number of genes in each clade was comparable between two subgenomes (Supplementary Data 10). Only two pairs of homoeologs (*Arr28085* and *Arr31194*; *Arr15137* and *Arr17921*) had significantly higher expression in the B subgenome, while the remaining homoeologs showed balanced expression and similar levels of DNA methylation (Supplementary Data 10). Overall, the *PER*s in both subgenomes remained conserved after the polyploidization event. In addition, genes from specific clades were duplicated after the genome merger. We thus propose that *PER*s of both subgenomes determine the high content of the HRPs in horseradish roots.

## The formation of the glucosinolate biosynthesis and breakdown bioprocess during horseradish genome evolution

GSLs are a major class of secondary metabolites in the Brassicaceae family, playing a role in plant protection[41]. Three independent steps are involved in the biosynthesis of GSLs: (1) side-chain elongation catalyzed by methylthioalkylmalate synthase enzymes (MAMs); (2) development of the core structure; (3) secondary modification of the amino acid side chain[42]. In the tribe Cardamineae, most of the known biosynthetic groups of GSLs are well-established (Fig. 8a, Supplementary Fig. 42)[16,42–44]. Although the GSL profile of horseradish is dominated by sinigrin, the entire profile is remarkably wide and includes most of the biosynthetic groups of the tribe[43,45,46]. Three groups of horseradish GSLs are involved in chain elongation: short chain methionine-derived, long chain methionine-derived, and chain elongated phenylalanine-derived. Two other groups occur independently of chain elongation: tryptophan-derived ("indole GSLs") and the combined group of benzyl GSLs and branched-chain GSLs, which were recently discovered to depend on a committed step catalyzed by CYP79C enzymes[43,47]. The unusual biosynthetic diversity suggests a similarly complex array of biosynthetic genes. We reconstructed the GSL biosynthesis and breakdown bioprocess in horseradish and other 10 species from different Brassicaceae lineages by identifying the syntelogs of the well-characterized GSL genes in *A. thaliana* (Fig. 8a, Supplementary Data 11). In total, we identified 279 genes involved in GSL biosynthesis and breakdown in the horseradish genome (Supplementary Data 11). Compared with other crucifer species, significantly more syntelogs of the GSL biosynthesis were observed in horseradish (Fig. 8b, Supplementary Figs. 43–45). The higher number of gene duplicates in the horseradish genome was associated with the allotetraploidization and tandem duplications (Supplementary Data 12). $Ks$ profiling based on all duplicated genes showed that (i) the duplicated genes were the result of continuous tandem duplications and amplification by WGDs, (ii) the most recent peak of gene duplicates ($Ks$ = -0.25) coincided with the formation of the allotetraploid genome (Fig. 8c, d). The duplicated gene copies could provide a genomic basis for the GSL metabolism and possibly explain the high accumulation of related GSLs. For example, genes encoding MAMs, IPMIs (isopropylmalate isomerases) and BCATs (branched-chain amino acid aminotransferases) in each subgenome were retained or duplicated after the allotetraploidization (Fig. 8e, f, Supplementary Fig. 46, Supplementary Data 11). Interestingly, the gene cluster encoding flavin-containing monooxygenases (FMO$_{GS-OX}$s), key enzymes associated with sinigrin biosynthesis, showed preferential retention after the polyploidization, was dominantly expressed and tandemly duplicated in the B subgenome (Fig. 8e, Supplementary Data 11, 13), consistent with the high accumulation of sinigrin in horseradish[48].

In *A. thaliana*, genes of the glycoside hydrolase 1 (GH1) family (e.g., *TGGs* and *PEN2*) have been identified as genes encoding myrosinase and responsible for the breakdown or turnover of GSLs[20,22,49]. We conducted a systematic analysis of the GH1 gene family in the 11 crucifer genomes and named the clades (I-X) based on homoeologous genes in *A. thaliana*[49] (Supplementary Data 14, Supplementary Figs. 47,

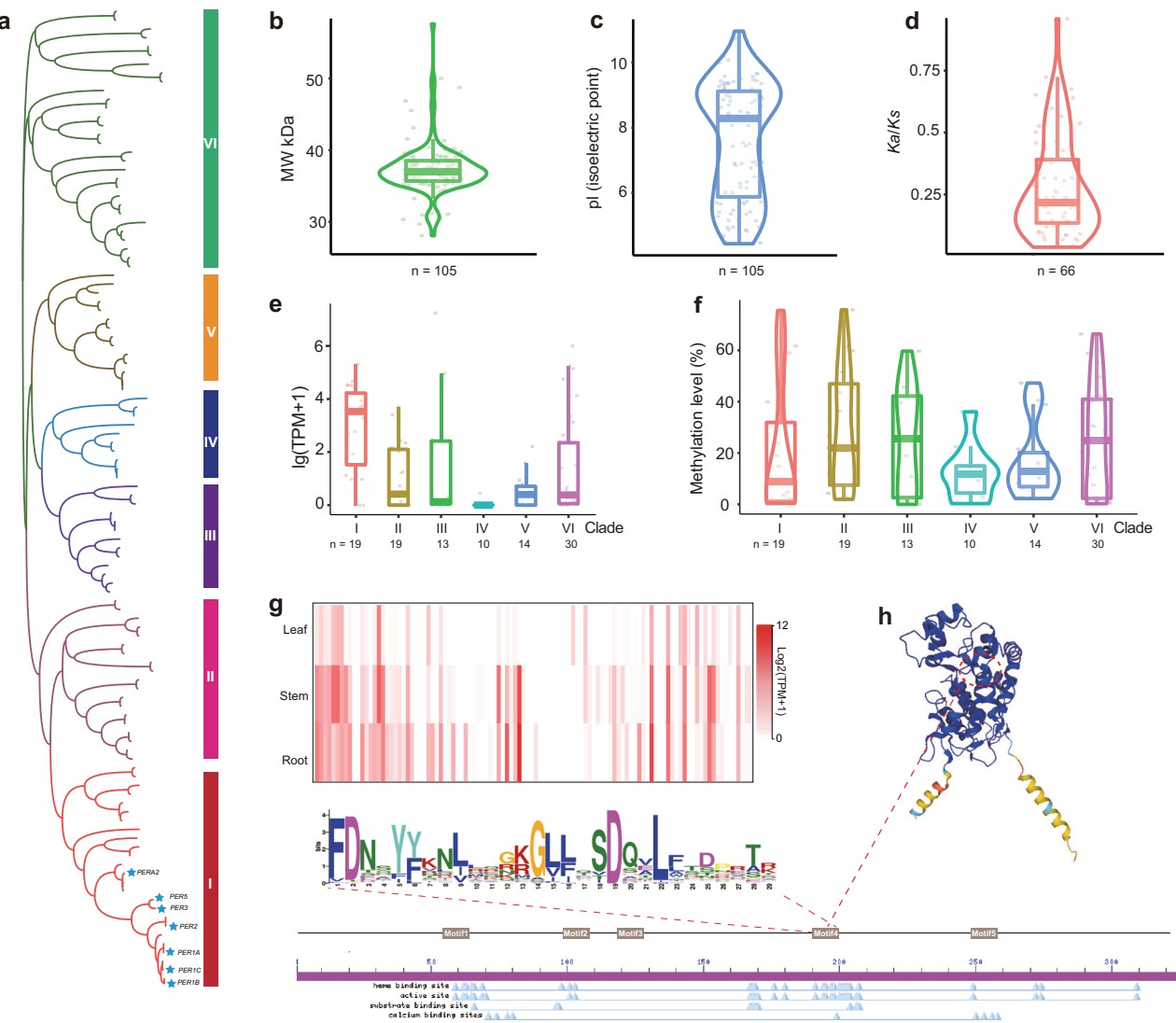

**Fig. 7 | Genes encoding horseradish peroxidases (HRPs). a** Phylogenetic tree of all the identified genes encoding HRP in the horseradish genome including the previously reported isoforms. Previously reported isoenzymes were marked (blue pentagram) in the tree. **b** Molecular weights of HRP isoforms. **c** Isoelectric points of HRP isoforms. **d** The $K_a/K_s$ value distribution of the HRP homoeolog pairs. **e** Expression of HRP genes of six different clades. TPM: transcripts per kilobase

million. **f** Methylation levels of HRP genes. **g** Expression of HRP genes in three different tissues. **h** Protein functional domain, conserved motif, and 3D structure analysis of HRPs. In violin/box plots, dots: data points; central lines: median values; box boundaries: 25th and 75th percentiles. whiskers: 1.5 * IQR (IQR: the interquartile range between the 25th and 75th percentile). The number ($n$) of data points for each violin/box is shown below. Source data are provided as a Source Data file.

48). Among the 11 Brassicaceae genomes analyzed, a total of 863 GH1 genes were identified, with horseradish having the highest number (85 genes) among all species in the Cardamineae tribe (Supplementary Data 14, Supplementary Figs. 47, 48). *TGGs* encoding the classical myrosinase were harbored in Clade VII[50]. In the horseradish genome, Clade VII consisted of 27 genes, making it the largest GH1 gene clade among the 11 genomes analyzed (Supplementary Data 14). Interestingly, genes in both main lineages (TGG1/2 and TGG4/5) of Clade VII in the horseradish genome underwent significant duplication events (Fig. 8g, Supplementary Fig. 47). In the TGG4/5 lineage, a gene cluster containing 11 duplicated genes was identified within a 300-K bp region in the A subgenome (Fig. 8h), while only one similar gene was found in the B subgenome. The pairwise *Ks* distribution of the duplicated genes indicates that the gene cluster has recently emerged and that considerable subgenome diversification has occurred after the allotetraploidization (Fig. 8i). Similarly, almost all genes (12 of 13) in the TGG4/5 lineage are recently tandemly duplicated in both subgenomes (Fig. 8g). The expression patterns of genes encoding classical myrosinase in horseradish were different from those in *A. thaliana*.

*AtTGG1*/2 were not expressed in the root, whereas *AtTGG4/5* were root-specific expressed genes. In horseradish, TGG5 lineage genes showed no or very low expression levels in all three tissues (root, stem, and leaf). In contrast, TGG1 lineage genes were the main expressed members in all tissues (Fig. 8g, Supplementary Data 15).

Genes encoding the atypical myrosinase (*BGLU18-BGLU33*) were distributed in Clades III to VI (Fig. 8g, Supplementary Data 14). Clade IV includes PEN2, which is associated with immunity and indole GSL turnover in intact cells[22,51]. In synteny analysis, we detected an interesting gene expansion of Clade IV in both subgenomes (a04:1.73–1.67 Mb; b04:1.77–1.69 Mb), corresponding to the gene cluster *BGLU15, BGLU28, BGLU29, BGLU17*, and *PEN2* in *A. thaliana*. We analyzed the phylogenetic relationship of these genes (Fig. 8j) and observed that duplicated *PEN2* genes were preferential retained after the polyploidization (Fig. 8k). Of the 85 homoeologous GH1 genes shared by A and B horseradish subgenomes, nine genes showed differential expression between the two subgenomes, with the most of these genes belonging to Clade IV (Supplementary Data 16). No significant gene expansion was observed in the other atypical myrosinase

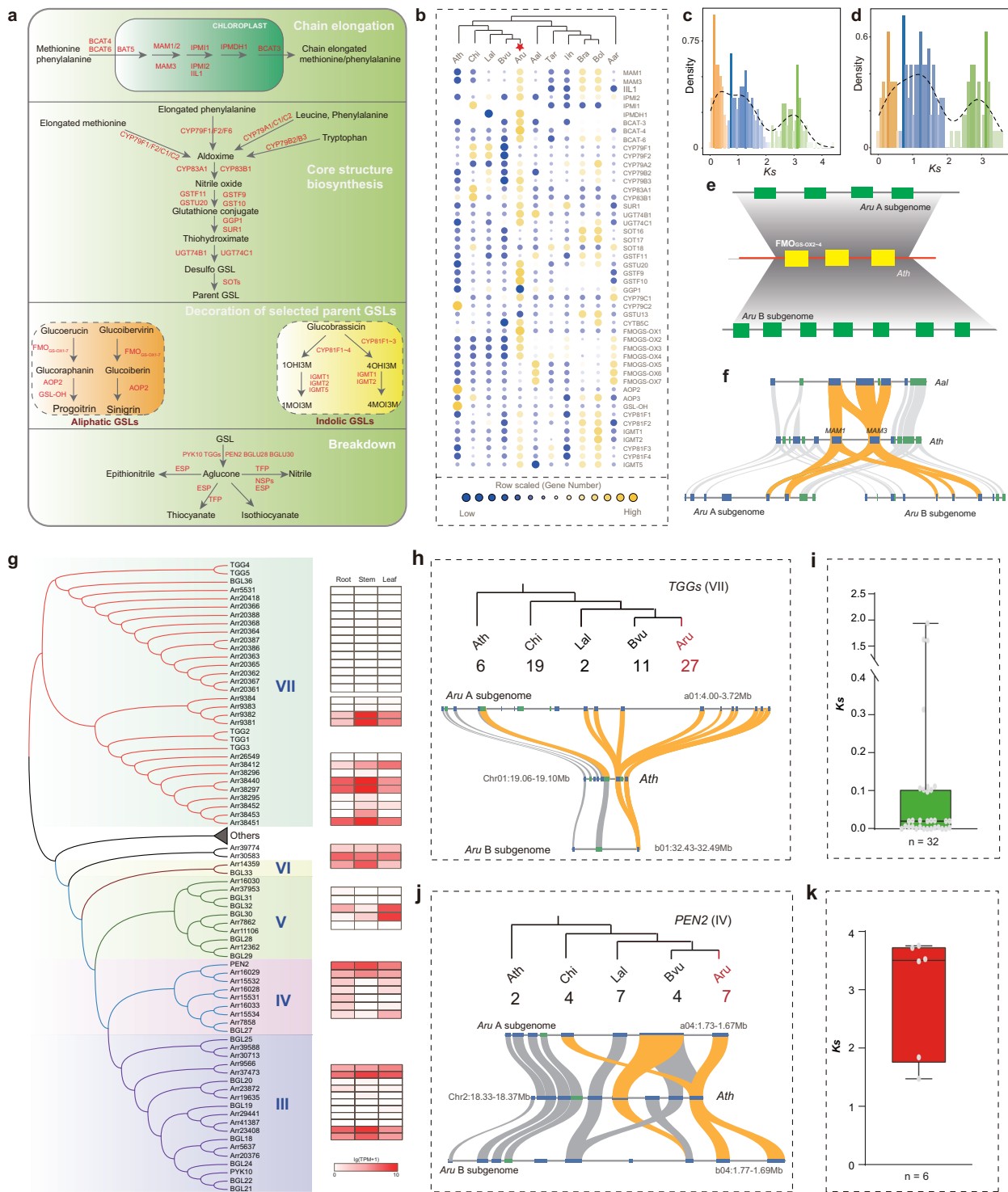

**Fig. 8 | Genome-wide comparison of genes involved in glucosinolate (GSL) metabolism pathways in horseradish and its relatives. a** A simplified diagram of the GSL biosynthesis pathways of methionine-derived GSLs, indolic GSLs, benzyl and branched chain GSLs, and 2-phenylethyl GSLs elucidated in *Arabidopsis thaliana*. **b** The varying copy number of syntelogs involved in GSL biosynthesis in 11 Brassicaceae species, including *Arabidopsis thaliana* (Ath), *Barbarea vulgaris* (Bvu), *Cardamine hirsuta* (Chi), *Leavenworthia alabamica* (Lal), *Armoracia rusticana* (Aru), *Arabis alpina* (Aal), *Brassica rapa* (Bra), *B. oleracea* (Bol), *Isatis indigotica* (Iin), *Thlaspi arvense* (Tar), and *Aethionema arabicum* (Aar). **c** *Ks* value distribution of duplicated GSL genes in the horseradish genome. **d** *Ks* value distribution of tandemly duplicated GSL genes in the horseradish genome. **e** Microsyntenic visualization of flavin-containing monooxygenase (FMO_GS-OX) gene clusters in

horseradish and *Arabidopsis thaliana*. **f** Microsyntenic visualization of methyl-thioalkylmalate synthase enzyme (MAM) genes in horseradish and *Arabidopsis thaliana*. **g** Phylogenetic tree of beta-glucosidase genes in horseradish and *Arabidopsis thaliana* and the gene expression heatmap (right). **h**, **j** Microsyntenic visualization of myrosinase (*TGG*) genes (**h**) and *PEN* genes (**j**) in horseradish and *Arabidopsis thaliana*. The trees indicate phylogenetic relationships between the analyzed species and predicted gene numbers are shown below. **i**, **k** *Ks* value distribution of duplicated *TGG* (**i**) and *PEN* (**k**) genes in the horseradish genome. In the box plots, dots: data points; central lines: median values; box boundaries: 25th and 75th percentiles; whiskers: minimum to maximum. The number (*n*) of data points for each box is shown below. Source data are provided as a Source Data file.

clades, although some genes showed higher expression levels in different tissues (Fig. 8g, Supplementary Data 15). The overall divergence of GH1 family genes in horseradish, including massive expansion and differentiation, reflects the diversity of GSLs in horseradish. These differences in myrosinase substrates possibly have led to changes in the selection pressure on different enzymes.

Examination of the expression and physical location of all GSL-related genes revealed that 65 syntelogs pairs exhibited differential expression between subgenomes, including key genes for side-chain elongation, core structure synthesis, side-chain modification and degradation, such as genes encoding IPMIs, GSTU (glutation *S*-transferase), $FMO_{GS-OX}$, CYP81F1 (cytochrome P450 monooxygenase 81F), IGMTs (indole glucosinolate *O*-methyltransferase) and NSPs (nitrile specifier proteins) (Supplementary Data 16), indicating the distinct roles of the different subgenomes in controlling GSL biosynthesis. For example, the final step of indole GSL methoxylation is methylation, which is catalyzed by IGMTs. IGMT1-4 produce 4-methoxyindol-3-ylmethylGSL (4MOI3M), while IGMT5 produces 1-methoxyindol-3-ylmethylGSL (1MOI3M)[52]. Among the six pairs of *IGMT* syntelogs, five showed preferential expression in the A subgenome. Moreover, methylation levels were closely associated with differential expression of the homoeologs. For example, the methylation level of *Arr21773* (*IGMT1*, 82.61%) was significantly higher than that of its homoeolog *Arr4260* (17.02%), whereas the gene expression of *Arr4260* was significantly higher (average TPM (transcripts per kilobase million), 33.09) than that of *Arr21773* (average TPM, 0.12) (Supplementary Data 16). Overall, although many genes involved in synthesis and degradation are preferentially retained during genome evolution, genes with strict dosage requirements are likely to be particularly constrained.

## Discussion

Horseradish is a vegetable crop known for its pungency and as a source of important raw materials for the biochemical industry (e.g., HRPs and GSLs)[14,23]. The current high-quality horseradish genome is a valuable genomic resource for biologists, biochemists and agronomists. De novo assembly of polyploid genomes is challenging due to the presence of multiple subgenomes, high heterozygosity and complicated assembly of repetitive sequences[53,54]. The previously inferred structure of the tetraploid horseradish genome suggested its origin via autopolyploidization or allopolyploidization[14]. We adopted an integrated strategy and generated the T2T gap-free reference genome of the allotetraploid horseradish, which is believed to be a rare T2T assembly of an allotetraploid genome in the Brassicaceae family.

Subgenome-level analyses have revealed asymmetric evolution of parental subgenomes in many allopolyploid systems, and homoeolog expression bias has also been detected, for example, in the octoploid strawberry genome[9] and in the allotetraploid *Brassica juncea*[7]. Comparison of *Arachis monticola* with diploid and tetraploid genomes revealed that structural variation played an important role in subgenome divergence during peanut domestication[55]. *Arabidopsis* allotetraploids exhibited predominantly epigenetic modifications and non-additive changes in gene expression[8]. Here, we did not observe a clear subgenome dominance, but we found that genes associated with different biological processes showed subgenome-specific bias in gene expression. The significant differences in homoeologous gene expression were strongly correlated with DNA methylation levels and distances to LTR-RTs, highlighting the crucial role of these factors in shaping the horseradish genome and its biological characteristics. In the octoploid strawberry genome, LTR-RTs density differences near homoeologs have also been associated with divergent gene expression[9]. Here, we identified 2,214 differentially methylated gene pairs between the two horseradish subgenomes that exhibited biased expression of homoeologs and were closely associated with LTR-RTs.

We therefore hypothesize that the differences in abundance of LTR-RTs contribute to the observed subgenome dominance.

Horseradish peroxidase has long been the focus of scientific research. However, little progress has been made in deciphering all its isoenzymes, and only the structure and catalytic mechanism of HRP C have been analyzed[23]. To date, only a few isoenzymes have been separated and characterized. Näätsaari et al.[24] conducted transcriptome sequencing and assembled parts of the unigenes encoding horseradish peroxidase. Here, we obtained a high-quality horseradish genome and performed an in-depth analysis of the HRP family. Analysis of *PER* genes showed that genes from both subgenomes enhanced HRP accumulation during horseradish genome evolution. Our results provide important resources for the study of horseradish isoenzymes and an essential reference for studies of interactions between subgenomes.

GSLs in horseradish include a high proportion of sinigrin and many other GSLs with diverse structures[45,56]. Several studies have explored GSL accumulation in cruciferous plants[7,13,57]. Comparative genomics and genome-wide association analyzes have revealed genetic loci responsible for seed oil quality and GSL biosynthesis in the allotetraploid *B. juncea*[58]. A natural variant of the *MAM3* gene is associated with GSL accumulation in *B. rapa*[59]. Comparative transcriptome analyzes of wasabi, horseradish, and mustard have shown that several gene clusters of the GSL biosynthesis have been convergently selected during evolution[48]. To decipher the underlying genetic basis of GSL biosynthesis and breakdown, we performed systematic comparisons of GSL genes in 11 crucifer species. Allopolyploidization and continuous tandem duplications have allowed preferential retention of GSL genes, contributing to the wide diversity of GSLs and their degradation products in horseradish.

## Methods
### Plant materials
The plant materials used in this study were obtained from the experimental orchard of the Genome Research Center, Leeuwenhoek Biotechnology Inc., Hong Kong, China (22°26′N, 114°12′E). The cultivar *Armoracia rusticana* L. cv. 'HD15' was selected for a de novo genome assembly. Three different tissues (root, leaf, and stem) were collected for RNA sequencing in 2018. Fresh samples were harvested and immediately frozen in liquid nitrogen after collection and then preserved at −80 °C.

### Whole-genome sequencing, HiC-sequencing, and RNA-seq
We conducted whole genome DNA sequencing using the Illumina, ONT and PacBio HiFi platforms. The genomic DNA was extracted from young leaves using the QIAGEN® Genomic kit (QIAGEN, Shanghai, China). The quality and quantity of the isolated DNA were estimated by electrophoresis on a 0.75% agarose gel and a NanoDrop™ D-1000 spectrophotometer (NanoDrop Technologies, Wilmington, DE, USA), respectively. We constructed Illumina sequencing libraries (with an insert size of 300 bp) following the manufacturer's standard protocol and sequenced them on the Illumina HiSeq X platform (Illumina, San Diego, CA, USA). We constructed the ONT library according to the protocol provided with the genomic sequencing kit SQK-LSK109 (Oxford Nanopore Technologies, Oxford, UK). The long reads were generated on the PromethION platform, and high-accuracy base calling was conducted using Guppy (v4.0.15) software[60]. For SMRT sequencing, high-molecular weight DNA was extracted using the CTAB method. A standard SMRTbell library was then prepared using 50 mg of DNA and the SMRTbell Express Template Prep Kit 2.0 following the manufacturer's instructions (Pacific Biosciences, CA, USA). The prepared SMRTbell libraries were sequenced on a PacBio Sequel II system.

We used Hi-C sequencing technology to assist scaffolding. The isolated cross-linked DNAs were purified, digested with Dpn II enzyme, blunt-end-repaired, and tagged with biotin. The biotin-containing DNA

fragments were captured and PCR-enriched to construct a Hi-C library. The library was then sequenced on the Illumina HiSeq X platform (Illumina) according to the PE150 strategy.

The root, stem, and leaf tissues were collected and processed following the manufacturer's instructions for library construction. We estimated the quality and quantity of the RNA samples using a Nano-DropTM D-1000 spectrophotometer, a Qubit® 3.0 Fluorometer (Thermo Fisher Scientific, Waltham, MA, USA), and an Agilent Bioanalyzer 2100 (Agilent Technologies, Santa Clara, CA, USA). The paired-end libraries with insert sizes of 300 bp were constructed using the TruSeq Sample Preparation kit and sequenced on the Illumina HiSeq X platform.

### Genome survey, genome assembly, and assessment

We calculated the 21-kmers with jellyfish (v2.3.1)[61] software using the Illumina reads and estimated the genome characteristics using GenomeScope 2.0 software (parameters: -k 21 -p 4 -m −1)[29]. For ONT assemblies, the NextDenovo software (v2.0, https://github.com/Nextomics/NextDenovo) with default parameters was used to assemble the long reads into contigs. The contigs were polished to improve the single-base accuracy using Pilon software (v1.24) with six rounds of iteration[62]. To address the difficulties in assembling polyploid genomes, we followed a comprehensive strategy to mitigate chimeric assemblies and enhance the overall quality of the assembly. First, we performed genome annotation using the MAKER-P pipeline (v2.29+) on the polished contigs and obtained the gene set (see the detailed description below)[63]. Second, we performed a pairwise synteny search of the whole genome and obtained colinear segments using JCVI (v1.2.20)[64]. The pairwise $K_s$ values of homoeologs were calculated using PAML (v4.9j)[65]. Third, we predicted the synteny blocks in the two subgenomes based on $K_s$ value profiling. The Hi-C sequencing reads were aligned to the contigs using Juicer (v1.6)[66] software. We removed the paired reads linking the predicted homoeologous contigs in the two subgenomes and then used the 3D de novo assembly (3D-DNA) pipeline (v201008) to obtain scaffolds[67].

Contigs from HiFi reads were assembled using HiFiasm (v0.17.7)[68] with default parameters. Leveraging the accuracy of HiFi long reads and the high continuity of the assembly, we directly performed scaffolding using Juicer software and 3D-DNA pipeline. Considering the improved continuity and integrity of the HiFi-based genome, we selected it as the genome backbone and filled the remaining gaps using the ONT-based genome with TGS-gapcloser (v1.2.1)[69]. The telomere regions were identified by searching for the seven-base telomere repeat sequence ('CCCTAAA').

We partitioned and phased subgenomes based on repetitive $k$-mers and orthologous genes using SubPhaser (v1.2)[30]. To avoid the effects of large-scale genomic deletions on clustering, we only used chromosomal regions within the syntenic regions for analysis. We ran SubPhaser using different parameters ('-k 13 -q 100 -f 2', -k 15 -q 100 -f 2', '-k 15 -q 200 -f 2', -k 13 -q 200 -f 2') and thus obtained robust results. Orthologous genes were also investigated. We adopted the branch length of the phylogenetic tree based on the orthologous genes. The genome of A. thaliana was used to detect synteny blocks and orthologous gene pairs of each scaffold using JCVI (v1.2.20) software. Orthologous gene pairs were subjected to multiple sequence alignment using MAFFT (v7.490)[70] and concatenated within each scaffold. The phylogenetic trees were built based on the concatenated sequences using FASTTREE (v2.1.11) with default parameters[71].

Multiple methods were utilized to assess the quality of the genome assembly. We evaluated the read mapping ratio (mapped reads number/total reads number) using Illumina, ONT and HiFi reads. The short reads were mapped to the genome using BWA MEM (v0.7.17)[72], and RNA reads were mapped using Hisat2 (v2.2.1) with default parameters[73]. ONT and HiFi reads were mapped using the Minimap2 (v2.24)[74]. BUSCO (v4.0) was used to evaluate the integrity of the genetic region[75]. The repeat sequences of the genome were assessed using the LTR Assembly Index (LAI)[76].

### Genome annotation

The MAKER-P pipeline (v2.29+) was utilized to predict the gene set by incorporating the ab initio prediction, homoeology-based prediction, and transcriptomes[63]. We collected protein sequences from the seven sequenced plants (including *Brassica oleracea*, *Eutrema salsugineum*, *Alyssum linifolium*, *Arabidopsis thaliana*, *Boechera stricta*, *Capsella rubella* and *C. grandiflora*) and a total of 1,440 benchmarking universal single-copy orthologs from the Embryophyta within BUSCO (v4.0)[75]. The protein sequences were mapped to the genome using tBlastn (v2.9.0+), and the Exonerate tool was used to acquire exact intron and exon positions. We assembled the RNA-seq reads from different libraries using Trinity software[77]. We trained the parameters for SNAP (v1.0)[78] and AUGUSTUS (v3.5.0)[79] using the gene modes from the output of the MAKER-P pipeline with only transcriptional data. The consensus gene set was finally obtained using all the above data and predictions. The non-coding RNAs were identified using Infernal (v1.1.3)[80] by searching against the publicly available Rfam database (v14.1)[81]. The gene functions were predicted by aligning the protein sequences to the TrEMBL databases using blast (e-value ≤ 1e − 5)[82]. Interproscan (v5.60-92.0) was used to obtain functional domains by searching against publicly available databases[83].

The Extensive de novo TE Annotator (EDTA) pipeline (v2.0.0) was used to build a high-quality nonredundant repeat sequence library[84]. The nonredundant repeat sequence library was integrated with the Repbase database, and repetitive sequences were further masked using RepeatMasker (v4.1.2). To accurately identify LTR retrotransposons, LTR-retriever (v2.9.0)[85] software was used to integrate the outputs from LTRharvest (v2.9.0)[86] and LTR_FINDER (v1.07)[87] with default parameters. We constructed phylogenetic trees of Ty3/*Gypsy* and Ty1/*Copia* sequences using the reverse transcriptase domain sequences of each LTR-RT element. Sequence alignments were generated using MAFFT[70] with default parameters, and Fasttree[71] with default parameters was used to obtain phylogenetic trees. The iTOL[88] website (v6) was visited to visualize and color the phylogenetic trees.

### Genome evolution and whole-genome duplication

To investigate the evolution of the horseradish genome, we selected 18 representative plant genomes including species of Poaceae (*Oryza sativa*), Vitales (*Vitis vinifera*), Rosaceae (*Fragaria vesca*), Asterales (*Actinidia chinensis*, *Olea europaea*, *Coffea arabica*, *Solanum tuberosum*), and 11 Brassicaceae species (*Brassica rapa*, *B. oleracea*, *Isatis indigotica*, *Thlaspi arvense*, *Arabis alpina*, *Barbarea vulgaris*, *Armoracia rusticana*, *Leavenworthia alabamica*, *Cardamine hirsuta*, *Arabidopsis thaliana*, *Aethionema arabicum*). To identify orthologous groups (orthogroups) and orthologs across the analyzed genomes we used Orthofinder (v2.4.0)[89] with default parameters. The gene families were clustered using the Orthofinder package[89]. Based on the high quality of the single-copy orthologous genes, the species tree was deduced using the Raxml (v8.0.0)[90] package. The divergence times were estimated using the MCMCTREE (v1.0)[65] program and corrected based on the timeline in the TIMETREE database (http://www.timetree.org/)[91]. CAFE[92] (v2.0) software was used to determine the significant expansion and contraction of gene families. JCVI software was used for pairwise and multiple genome syntenic comparisons and visualizations. The polyploidization events in the horseradish genome were estimated by using $K_s$ profiling. We roughly dated the genome duplication events using the $K_s$ values. The peaks of $K_s$ profiles were determined and translated into divergence time (T) in millions of years using the following formula[93]:

$$T = K_s / (2 \times 9.1 \times 10^{-9}) \times 10^{-6} \text{ million years ago} \tag{1}$$

## Whole-genome methylation analysis and 3D-genomic analysis

Oxford Nanopore sequencers are sensitive to base modifications. The reads were base called using the Guppy software. We aligned the base-called reads to the reference genome using Minimap2[74]. The detection of DNA cytosine methylation using ONT sequencing was processed by Nanopolish (v0.8.4)[60]. Then, deepTools (v3.5.1)[94] software was run to visualize methylation levels around the gene loci. The methylation levels were determined by dividing the number of reads covering each mC site by the total number of reads mapped to the specific cytosine base.

We also adopted whole-genome bisulfite sequencing to obtain single-base resolution methylomes. We constructed bisulfite sequencing libraries for three replicates of different tissues, and conducted genome sequencing using the Illumina HiSeq X platform[95]. The reads were aligned to the reference genome using Bismark (v0.24)[96] and HISAT2 (v2.2.1)[73]. The number of methylated and unmethylated reads per cytosine was obtained for each sample.

The clean reads were mapped against their corresponding reference genomes with Juicer[66]. Only the uniquely mapped paired-end reads were used for subsequent analysis. The contact matrices were generated at different resolutions (10, 25, 50, 100 kbp) using hicConvertFormat in HiCExplorer (v3.7.2)[97]. The hicPCA program embedded in HiCExplorer was used to delineate A/B compartments at a 50-kb resolution. TAD-like structures were identified using the hicFindTADs program embedded in HiCExplorer at the different resolutions (10, 25, 50, 100 kbp). The Multiple TAD domain files from hicFindTads were merged using hicMergeDomains. The Fit-Hi-C (v2.05) tool was used to identify Hi-C interaction peaks at the 10-kbp resolution[98].

## Asymmetric evolution of horseradish subgenomes

We investigated the asymmetric evolution of the A and B subgenomes in the horseradish genome at different levels, including the analysis of differential expression of homoeologs and the deletion of homoeologs after the polyploidization event. Clean RNA-seq reads were mapped onto the horseradish genome using Hisat2 (v2.2.1) with default parameters. The gene expression level of individual genes was quantified with StringTie (v2.1.5)[99] using TPM values. Homoeolog expression bias (HEB) analysis was performed within syntenic gene pairs between subgenomes. Differentially expressed gene pairs with greater than two-fold differences across all three tissues were defined as dominant gene pairs.

The unpaired genes within syntenic blocks between subgenomes possibly indicated the deletion of their homoeologs in the counterpart subgenome. Using the unpaired genes as queries, we searched the homoeologs within the syntenic block of the counterpart subgenome to check for gene deletion. Blastp software was used with thresholds of e-value < 10e-5 and coverage >50%. Theoretically, the identified lost genes encompass newly formed genes (known as orphan genes) that arose after the merger of parental (sub)genomes. These genes are expected to be present in horseradish-specific orthogroups. To mitigate the influence of orphan genes, we removed genes from the lost gene dataset that are associated with horseradish-specific orthogroups.

## Analysis of genes encoding horseradish peroxidases

We identified the *PER*s in the horseradish genome using comprehensive methods. First, all the *PER*s from *A. thaliana* were searched against the horseradish protein sequences using Blastp with threshold e-value < 10e-5 and coverage >50%. The candidate proteins with the target functional domains (PF00141 and cd00693) were identified for further investigation. Multiple sequence alignments of all the identified proteins from horseradish and *A. thaliana* were performed using MUSCLE (v5.1) with the default parameters[100]. A phylogenetic tree was constructed using Fasttree software with the default parameters. The isoelectric point and molecular weight were analyzed using Expasy

(https://www.expasy.org/). The MEME server (http://meme-suite.org/tools/meme) was used to identify the conserved motifs[101].

## Comparative genomics with other Brassicaceae species

We reconstructed the group of genes involved in the GSL biosynthesis and breakdown bioprocess in the horseradish genome. First, all GSL genes in *A. thaliana* were collected based on the literature[42,44,47,57]. Second, we conducted the synteny analysis of 10 Brassicaceae genomes using the *A. thaliana* genome as reference using the JCVI software. To identify myrosinase in the horseradish genome, we searched all the candidates in the horseradish genome using the beta-glucosidase protein sequences of *A. thaliana* as queries by using Blastp software with a threshold of e-value < 10e-5. The candidates with the target functional domain (PF00232) were retained for further analysis. The phylogenetic analysis was conducted as described above for *PER*s.

## Bisulfite sequencing PCR analysis

Bisulfite sequencing PCR (BSP) was performed to assay DNA methylation levels in different tissues (root, stem, and leaf). Genomic DNA was extracted from different tissues using a Genomic DNA Extraction Kit (Takara, Shiga, Japan), followed by treatment with sodium bisulfite using the EZ DNA Methylation-Gold Kit (Zymo Research, Orange, CA, USA). PCR Primers were designed using the MethPrimer program (v1.0) (http://www.urogene.org/cgi-bin/methprimer/methprimer.cgi) (Supplementary Data 17). The BSP products were cloned into a pMD19-T simple vector (Takara) and a total of 30 positive clones from each tissue were selected randomly for Sanger sequencing.

## RNA-seq analysis and validation of gene expression

We mapped the RNA-seq reads of different tissues to the reference genome using Hisat2 (v2.2.1) with default parameters. The transcript assembly and quantification were conducted using StringTie (v2.1.5) with default parameters. Total RNA was extracted from different tissues using an RNA Extraction Kit (Biomed, Beijing, China). Quantitative reverse transcription PCR (RT-qPCR) was performed to analyze gene expression using 2 × TB Green Premix Ex Taq II (TaKaRa) with an ABI real-time PCR system (Applied Biosystems, Waltham, MA, USA). Actin was used as an internal control gene for normalization of expression levels. The $2^{(-\Delta\Delta Ct)}$ method was used to calculate the relative transcription levels of each sample analyzed.

## Reporting summary

Further information on research design is available in the Nature Portfolio Reporting Summary linked to this article.

## Data availability

The genome sequence data and genome assembly generated in this study have been deposited in the National Genomics Data Center under the accession PRJCA009966. The genome assembly and annotation files and additional functional annotations are also available publicly at Figshare [https://doi.org/10.6084/m9.figshare.21780176.v2][102]. Source data are provided with this paper.

## Code availability

All codes and pipelines for this study are openly available at Zenodo [https://doi.org/10.5281/zenodo.8058147][103].

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

## Acknowledgements

This work was supported by the fund (grant no. BIO0121) from Shangji Biotechnology, Tianjin, China. We thank colleagues at the genome center of the Fondazione Edmund Mach, San Michele all'Adige, Italy, for their help. We thank T. Mandáková for performing cytogenetic validation. We acknowledge L. Bianco and P. Fontana for their comments and suggestions.

## Author contributions

F.S. and M.A.L. coordinated the project, conceived and designed the experiments. F.S., S.X. and Q.S. assembled and performed bioinformatics analyses of the genomes. C.B. performed the T2T genome assembly. S.X., Q.S. prepared samples for sequencing. F.S., Q.S. and S.X. wrote the manuscript. F.S. and M.A.L. revised the manuscript.

## Competing interests

The authors declare no competing interests.
