## [Peer Review File · Nature Communications]

The allotetraploid horseradish genome provides insights into subgenome diversification and formation of critical traitsReviewers' Comments:

Reviewer #1:

Remarks to the Author:

This review almost solely address the investigation of glucosinolate biosynthetic genes, as requested by the Editor.

First a few other observations.

I could not help being puzzled by the distribution map in figure

1. According to this map, horseradish grows as far north in Canada and Russia as almost the North Pole, yet not at all in Greenland, and also in every single state in USA and Mexico and every province of China, Russia and Egypt, even the desert, but not in Mongolia or Libya. Either I would abandon a map altogether or I would work on a realistic map, e.g. based on dots for confirmed findings or at least smaller political units than countries.

Section plant materials specifies that three "tissues", root, leaves and stem, were sampled for transcriptomics. I am uncertain of the definition of leaves and stem in this context, do you by any chance mean petioles and remaining leaf? Wouldn't you call these major plant part organs rather than tissues?

Figure 1 c morphology of horseradish, it seems to be only part of a horseradish plant, or are the leaves that small and the root that short and blunt? Could there even be flower, so perhaps it is the vegetative stage only?

Glucosinolate-related

I agree that glucosinolate related genes are well suited for investigation and comparison with related species, because this pathway is very well-characterized since it has received so much attention previously. A complete comparison could give clues to consequences of a WGD and perhaps the plant's biochemistry, nutritional and sensory properties and perhaps a particular ecology in the history of horseradish evolution. But a complete gene list and a rational selection of reference species is essential for a relevant comparison.

How did you decide on these exact reference species? The three reference species do not seem like a balanced selection of other crucifers (at least heavily skewed towards Brassica...).

Gene list: The newest reference I could find in Table S2 was from 2010, and indeed, it seems as if the last decade of investigations of glucosinolate biosynthesis and catabolism were not taken into account when compiling the list of genes to search for. I'll list some obvious additional genes to search for at the end of the comments.

There are some chemical errors in the figures and supplementary table, but they can be fixed.

The discussion of the data is rather vague and does not in any way relate to specific papers on glucosinolate biosynthesis and catabolism in horseradish. The mentioned allyl isothiocyanate can be formed using a fraction of the identified genes, and most of them bear no relation to that glucosinolate product or the precursor sinigrin.

The number of myrosinase genes are mentioned (line 348) in connection with the rate of release of ITCs etc, and it is mentioned that 33 myrosinases were identified, which to this reader seemed like the number from horseradish. However, the number of genes from horseradish in that category listed in table S2 is only 12, and they are all listed as homologs of PEN2. The PEN2 gene is considered associated with disease resistance and indole GSL turnover in intact cells but not release of ITCs after tissue fragmentation (ref the Clay et al. and Bednarek et al. twin papers on PEN2 in Science). So it is

a bit of a mystery that no TGG homolog is identified, and I am not sure the current discussion (in results) is spot on. Line 353 "the reinforcement of the glucosinolate breakdown process...", which reinforcement? A homolog of PEN2 is not a likely candidate for the classical myrosinase function, is it?

An obvious number to discuss is the number of TGGs: 4, 6 and 4 in the three reference species, but zero in horseradish. Now THAT is intriguing, and not easily termed a "reinforcement".

Is there any other known crucifer lacking a homolog of the TGGs? What about crucifers more closely related to horseradish, do they have TGGs? Has the TGGs been lost recently in horseradish?

I agree that a multiplication of the PEN family is intriguing, including the gene cluster map (Figure 9d), but I disagree that this can be automatically related to formation of the pungent allyl ITC after crushing horseradish roots.

The stress on the number of myrosinases is also troublesome in another way: The 12 myrosinases are catalogued as PEN2 homologs in Table S2, but according to Figure 9b only two of them can strictly be called closest homologs to PEN2, the remaining are closer homologs to a lot of other Arabidopsis glucosidases. With this observation, to me the entire stress on a high number of myrosinases in horseradish falls apart, and we are left with a plant with no classical myrosinase at all and only an average number of PEN2-like atypical myrosinases.

The comments about epithioester protein (line 351) are mysterious as no such protein exists as far as I know.

Crude numbers of glucosinolate related proteins in the three reference species and horseradish is mentioned in Results (line 335-336), and the number for horseradish is numerically higher than the rest. However, assuming that the numbers were exhaustive, is this comparison fair? The numbers from the reference species have been pruned by decades of functional testing and mutation studies, so would seem to represent a corrected number, while I assume that the number for horseradish is the grand total of candidate homologs, some of them might have other functions. In Discussion, the numbers are not repeated but an attempt is made for a comparison of numbers within functional groups, e.g. breakdown and product diversification. As mentioned above, I think that discussion is flawed from the comparison of the PENs with the TGGs, but still I suggest that in a revised version, a table comparing numbers of GSL related genes /gene candidates should be subdivided into functional groups, to see whether there is a tendency for general doubling of gene numbers or a tendency for only a specific functional group proliferating in gene number. In this grouping, I suggest distinction of classical myrosinases like TGG and atypical or potential myrosinases like PEN, BGLU, BABG.

If the authors will suggest a biological role of a possibly larger number of glucosinolate related genes in this species than in e.g. Brassica rapa and Arabidopsis thaliana, or from those specific genes found, some detailed basis in the current knowledge of horseradish would seem relevant. How does the battery of specifier proteins relate to known GSL products from the species? And what about relations of the biosynthesis battery to the GSL diversity in the species? The discussion of the chemistry of horseradish is almost devoid of phytochemical details and references (lines 402-402, is this true for all tissues and all life stages?). A larger perspective based in solid, published facts would seem to be relevant in the interpretation of the gene list. Line 403 what is "super-accumulation", it is not a concept I have seen in the glucosinolate literature before.

The glucosinolate details in Discussion relate only vaguely to the details in Results, but mainly refer a handful of general results in the area. Glucosinolate results are absent from the Abstract. I feel that no clear conclusion on the large glucosinolate related gene dataset is arrived at.

Conclusion of the review: You present a lot of interesting and valuable data, which will be a fantastic resource for future biologists and biochemists. Yet I suggest repeated, comprehensive search for

glucosinolate-related genes, re-assessment and re-cataloging of suggested functions, re-design of the comparison with other species including rational choice of reference species, and re-analysis of the data. Finally a complete re-writing of the interpretation of glucosinolate related genes, aiming at specific conclusions related to previous knowledge on this chemically very well-characterized species.

Details.

Figure 9.

You miss two additional glucosinolate biosynthesis pathways relevant for horseradish, the combined biosynthesis of benzylGSL and branched chain amino acid-derived glucosinolates starting with CYP79C, and the biosynthesis of 2-phenylethylGSL starting with CYP79F1/2/6.

Pathway from Met:

Between step 1 and 2: there is a transport to the chloroplast needed here

Step 2. There are more genes involved than listed in the chain elongation cycle.

After step 2: there is transport out of the chloroplast needed here, and another aminotransferase step, which might or might not be catalyzed by the same enzyme as in step 1 (BCAT 3 was suggested to play a specific role here, Knill et al., 2008)

Compound five. Currently, this never isolated intermediate is suspected to be a nitrile oxide by many authors, are you sure you want to stick to the aci-nitro hypothesis?

S-alkyl thiohydroximate: In our days, with the alkyl known to be glutathione, this is a funny, outdated name for this intermediate, which I would prefer to term "glutathione conjugate".

Pathway from Trp

You have mixed up the order of intermediates, also here the aci-nitro / nitrile oxide comes right after the oxime. If you wish to swop to nitrile oxide nomenclature, it would be 2-(indol-3-yl)ethanenitrile N-oxide. (N in italics).

The name S-indol-thiohydroximate is completely wrong, the indole group is certainly not on the S, here is the glutathione residue. I would call this intermediate "glutathione conjugate" as for the Met-pathway, because the complete name is quite long.

So, the order from Trp would be: oxime, nitrile oxide, glutathione conjugate, thiohydroximate, desulfoGSL, glucobrassicin.

From glucobrassicin, the intermediate to the left should be named 1HOI3M in analogy with 4HOI3M.

The genes to 1MOI3M are in contrast to the literature (Pfalz et al. 2016).

Not only PEN2 but also TGG hydrolyze indole glucosinolates.

Hydrolysis products: AITC is misleading, you mean isothiocyanates in general. The specifier proteins listed are put in wrong places. ESP means epithiospecifier protein! Thiocyanates are promoted by the thiocyanate forming protein. Etc.

Figure 9 legend: you mention numbers in brackets, but I see no brackets. Is the order of the species in the figure as explained in the legend?

Supplementary Table S2. I support the general table layout with original references for the genes listed and some sorting of the genes in classes. But some of the classes are misleading. Perhaps an additional column with a brief description of the biochemical function of each gene would be useful for the average reader? In particular if you wish to compare in detail with reference species, which seems to be the logic of this Table.

The last heading epithioester protein is a mystery. Does that refer to ESP, which has another meaning? The name is not a good headline for the listed genes. Most of the other proteins listed are specifier proteins, this could be a good heading but does not fit all currently listed.

Missing genes/enzymes discovered since 2010 (a selection, there could be more)

TRANSPORT

GTR1 and GTR2 Search "Nour-Eldin" as author. The best place to start is this:

DOI 10.1016/j.tplants.2015.04.006, make sure you do not present the GTRs as exporters, they are

importers (to phloem companion cells).

THIOGLUCOSIDASES: BrBABG.a and BrBABG.b from *B. rapa*, Klein and Sattely 2017 DOI 10.1073/pnas.1615625114

BGLU 18, 28 and 30

PYK10 Nakano et al 2017 DOI

10.1111/tpj.13377

BIOSYNTHESIS:

Transport of alfa keto acids in and out of the chloroplast by BAT genes Petersen et al 2018 DOI 10.1111/jipb.12705

CYP79F6 from the same tribe as horseradish, relevant for GSLs in horseradish, Byrne 2017 DOI

10.1038/srep40728, functionally characterized by Wang et al 2021 doi 10.1111/tpj.15212

CYP79C1 and CYP79C2 functionally characterized by Wang et al 2020, relevant for GSLs in horseradish such as benzylGSL, DOI 10.3389/fpls.2020.00057

IGMT3, 4, 5 Pfalz 2017 DOI 10.1104/pp.16.01402 Pfalz et al 2011 DOI

10.1105/tpc.110.081711

Special modifications in Arabidopsis (acylation), is this gene in horseradish: SCPL17

HYDROLYSIS PRODUCTS:

The best place to start is here: Wittstock et al 2016 DOI 10.1016/bs.abr.2016.06.006

With this, you should be able to get the products and proteins matched in Figure 9, and arrive at some system in the last block of genes in Table S2.

Here are some Arabidopsis genes to consider: NSP3-5, ESM1,

Consultation of recent reviews is also recommended, the newest general GSL biosynthesis reviews I am aware of are:

Petersen et al 2018 DOI 10.1111/jipb.12705

Harun et al 2020 DOI 10.1021/acs.jafc.0c01916

Chhajed 2020 doi 10.3390/agronomy10111786

But beware of errors in the reviews, e.g. very wrong indication of the GTR transporters as exporters by Harun et al., a blunder you do not want to copy in your manuscript, and also a few connectivity errors in the Chhajed review (the combined biosynthesis overview figure)

There are quite many original papers on horseradish glucosinolates and isothiocyanates. I found 16 hits in web of sciences on "glucosinolate" AND "horseradish" in title, and 61 with these two terms in "topic".

There is even a review of glucosinolate diversity in the tribe to which horseradish belongs, including the horseradish genus in their table 7: Olsen et al 2016 DOI 10.1016/j.phytochem.2016.09.013

Horseradish seems to be characterized by a few dominating GSLs and a quite large number of and structurally diverse set of GSLs. The profile of hydrolysis products has been reported by several authors, also using modern state of the art technology. Is there any relation of the known chemistry and the genes identified?

And there are two or three genomes from that same tribe, *Cardamine hirsute* and *Barbarea vulgaris* (and watercress, I think I heard it was prepared or submitted and could appear during the revision of your manuscript?)

Reviewer #2:

Remarks to the Author:

The authors in this manuscript describe a new reference genome of horseradish by using the up-to-date Nanopore sequencing and Hi-C technologies. They further describe the dynamic characteristics of

the allopolyploid subgenome and explored the potential genetic mechanisms underlying the traits. Overall, the new reference genome and further subgenome investigation will be of great value for polyploid genome evolution and horseradish breeding in the future. However, some of the results are largely descriptive and I still have some concerns as followed.

Considering these concerns, I recommend major revision.

Major concerns:

- 1.The authors need to recheck the analysis results carefully. For example, Line 114, the authors described the LAI value as 24, but Figure 2 shows the LAI value as 17. And, In Figure 2, the author described there are 39162 gene models, but they describe there are 40,290 protein-coding genes in main text (Line 119).
- 2.It seems to have some mistakes for the assembly pipeline and statistics. The assembly pipeline shown in the figure 2a, 2b do not correspond to the results. The authors described in results and methods that they mainly use the Juicer and 3D-DNA pipeline to perform the assembly while it was shown to be ALLHiC and Juicerbox in the fig. 2a. The contig N50 they show in the result is 17.5Mb while it became 7.95 Mb in the Supplementary table S2 and fig. 2a. The authors should carefully check those results and figures.
- 3.The current analysis is too general. For example, the B subgenome had a higher level of methylation (~69.55%) than the A subgenome (63.49%). No details were further provided. And, there is a significant difference in the burst time between Copia and Gypsy LTRs. However, the authors didn't provide further explanations on it. I suggested the authors calculated the distribution of these newly formed LTRs and analyze the consequences of these newly inserted LTRs.
- 4.The methodology used is nevertheless not completely clear: no details have been provided in their method. For example, it is difficult for others to repeat their k-mer analysis and conclude that the horseradish genome had an allotetraploid origin. I strongly suggested that they deposited their scripts pipelines in github. This also ensures that the author's work can be repeated. In addition, version of software and parameters should be provided in the method part.
- 5.Concerning that the levels of methylation modifications are various in different tissues and the methylation data in this paper represents only one tissue type. It is recommended that they provide more data to support the results that 'DNA methylation play important role in subgenome diversification' such as methylation data from leaf root or flower.

Minor concerns:

- 1.Line 54-56, grammar mistake for double use of "is".
- 2.Line 80-81, I didn't find the genome survey results in table S1 as the authors described.
- 3.Line86-88: Please add a reference for this sentence.
- 4.Line214: Please show the exact number of methylated genes with LTRs.
- 5.Line 790-791, Here, it should be the length instant of the ratio.
- 6.Line 791, how to determine aaaa, aaab, aabc types of k-mers? The authors should describe it in the method part.
- 7.In Table S4, I suggested adding the number of contigs.
- 8.The circos plot in Figure 2b is not well displayed, it appears to be shifted to the left.
- 9.The Hi-C contact map has a lot of gray lines and squares, which the authors should explain.
- 10.Figure 2C should include the chromosome id.
- 11.The composition of 'Other' in Figure 3a is not very clear. The author should clarify it.
- 12.Line 144, the sentence is incomplete.
- 13.I didn't find the description of Figure 3e in the main text. Although I found some interesting TE hotspots in Figure 3e.
- 14.I suggested adding Cretaceous–Paleogene boundary in Figure 4a.
- 15.In Figure 4b, there is no description of the Venn diagram in Figure 4b.

Reviewer #3:

Remarks to the Author:

Overview: In this manuscript, "The allotetraploid horseradish genome provides insights into subgenome diversification and formation of critical traits," Shen and colleagues provide a high-quality genome of horseradish (*Armoracia rusticana*) which is in the mustard family (Brassicaceae). Horseradish is a root vegetable that is used as a popular spicy condiment. It is known to possess interesting metabolites that give it a distinctive flavor (e.g., glucosinolates and horseradish peroxidases) and isothiocyanates that may be cancer-preventive. The genome has not been previously published to this reviewer's knowledge and may be an important resource for plant breeders and future comparative studies.

Questions for the authors:

1. Plant materials. Are the seeds from the sequenced cultivar 'HD15' publicly available?
2. Taxon sampling and consistency. How were the 15 outgroup genomes selected? It is not clear these species were chosen (both outside of the mustard family and the seven species within the mustard family seem). For example, there are many more Brassicaceae genomes available. On a related note, why were the various gene family analyses not done with the same species? For example, *Brassica rapa* was used to re-construct the groups of genes involved in glucosinolate biosynthesis; but that genome was not used in other comparisons.
3. Novelty. It is not clear what the main novel results are given how the introduction and discussion are presented. For example, there is a lot of discussion on how much is learned about polyploid genome evolution; however, what is new (versus merely confirmatory) of what was already published in Mandakova and Lysak (2019)(cited reference 9)?
Are there any new insights on sub-genome polyploid evolution?
Similarly, what are the new findings about glucosinolate evolution that were not previously known?
While confirmatory results are valuable, this reviewer struggled to find new insights.
4. Main figures. The manuscript presents extensive results (with nine main figures).
Could many of the results be put into supplemental materials?
Also, some key figure elements are outdated – for example, the phylogeny in Figure 1a is not useful; see Figure 1 in Mandakova and Lysak 2019 or more updated phylogenies (e.g., Walden et al. 2020. *Nature Communications* 11: 3795).
5. References. References 29 and 30 from the Literature Cited are not cited in the main text. Also, some references are used that are inappropriate, for example, the introduction refers to polyploid evolution in cotton and Brassica but then cites a strawberry paper (reference 2). This occurs throughout the introduction and discussion where more recent or specific references would do the current literature justice.

Overall, aside from the comments above, the methods are sound and meet the expected standards. The interpretations and conclusions are not wrong; however, their significance to the field and whether they are noteworthy for the *Nature Communications* audience is questionable. The high-quality annotated genome will be a valuable resource for comparative genomics across species, for population genomics (e.g., GWAS panel for horseradish), and for future functional investigations into specific pathways (e.g., horseradish isoenzymes).

Response to Reviewer #1:

This review almost solely addresses the investigation of glucosinolate biosynthetic genes, as requested by the Editor.

Response: Thank you for your comments, we have benefited greatly from your detailed comments on GSLs, and your practical comments have fundamentally improved this part of our manuscript. We have corrected the errors you pointed out and expanded and completed the in-depth analysis you requested (especially by adding more species and GSL-related gene members).

First a few other observations.

1) *I could not help being puzzled by the distribution map in figure 1. According to this map, horseradish grows as far north in Canada and Russia as almost the North Pole, yet not at all in Greenland, and also in every single state in USA and Mexico and every province of China, Russia and Egypt, even the desert, but not in Mongolia or Libya. Either I would abandon a map altogether or I would work on a realistic map, e.g. based on dots for confirmed findings or at least smaller political units than countries.*

Response: Thank you for this valid comment. In accordance with your suggestion, we have adopted more comprehensive and accurate data and updated the figure in the revised version (Figure 1b). At the same time, we have added an evolutionary tree (Figure S1).

2) *Section plant materials specifies that three “tissues”, root, leaves and stem, were sampled for transcriptomics. I am uncertain of the definition of leaves and stem in this context, do you by any chance mean petioles and remaining leaf? Wouldn't you call these major plant part organs rather than tissues?*

Response: We thank the Reviewer for this concern. Horseradish has a taproot similar to carrots and beets, and it has stems with extremely short internodes that grow only when the plant is bolting. We have taken the tissue of shortened stems between the root and petioles. Wherever possible, we refer to the three samples as “leaf tissue”, “stem tissue”, and “root tissue”.

3) *Figure 1 c morphology of horseradish, it seems to be only part of a horseradish plant, or are the leaves that small and the root that short and blunt? Could there even be flower, so perhaps it is the vegetative stage only?*

Response: To help readers better understand the anatomical structure of horseradish, we have included the illustration of roots, stems, leaves, and flowers in the updated version of the figure (Figure 1a).

Glucosinolate-related

I agree that glucosinolate related genes are well suited for investigation and comparison with related species, because this pathway is very well-characterized since it has received so much attention previously. A complete comparison could give clues to consequences of a WGD and perhaps the plant's biochemistry, nutritional and sensory properties and perhaps a particular ecology in the history of horseradish evolution. But a complete gene list and a rational selection of reference species is essential for a relevant comparison.

Response: We thank you for your comment. Indeed, the identification of GSL-associated genes in the horseradish genome and other Brassicaceae genomes has important implications for understanding the evolution of GSL-associated genes. We have perfected the genes involved in GSL according to your suggestions and selected representative Brassicaceae species for comparative analysis.

How did you decide on these exact reference species? The three reference species do not seem like a balanced selection of other crucifers (at least heavily skewed towards Brassica...).

Response: We agree with the Reviewer that our selection of reference genomes was biased towards Brassica. To address the Reviewer's comment, we expanded the number of Brassicaceae genomes analyzed, taking into account the fact that only a limited number of crucifer genomes have been sequenced and their genomes are available. Our expanded selection followed two criteria: (i) we included genomes of the tribe Cardamineae, where *Armoracia* is placed, and (ii) genomes representing the major Brassicaceae lineages, i.e., Lineage I (Cardamineae), II (*Brassica*), III, IV, and *Aethionema*.

Tribe Cardamineae: *Armoracia rusticana* (horseradish), *Barbarea vulgaris*, *Cardamine hirsuta* and *Leavenworthia alabamica*.

Lin I is covered by *Arabidopsis* and 4 Cardamineae spp. listed above.

Lin II is covered by two *Brassica* spp.

Lin III: no genome is available.

Lin IV: *Arabis alpina*.

Aethionemeae: *Aethionema arabicum*

Gene list: The newest reference I could find in Table S2 was from 2010, and indeed, it seems as if the last decade of investigations of glucosinolate biosynthesis and catabolism were not taken into account when compiling the list of genes to search for. I'll list some obvious additional genes to search for at the end of the comments.

Response: Thank you for your comment and suggestions. Based on the literature you provided, we have added missing genes and pathways (Table S23). The updated gene list includes 113 GSL genes in *Arabidopsis thaliana*: side-chain elongation (9 genes), core structure synthesis (22), side-chain modification (17), co-substrate pathways (16), GSL degradation (21), and GSL transcriptional components and transporters (28).

There are some chemical errors in the figures and supplementary table, but they can be fixed.

Response: Thank you for pointing out the chemical errors and suggesting corrections that we have made in the revised manuscript.

The discussion of the data is rather vague and does not in any way relate to specific papers on glucosinolate biosynthesis and catabolism in horseradish. The mentioned allyl isothiocyanate can be formed using a fraction of the identified genes, and most of them bear no relation to that glucosinolate product or the precursor sinigrin.

Response: We admit that the previous interpretation of this data contained ambiguities. In the updated version, we have improved the gene list and expanded the number and representativeness of species for which genomes were compared. At the same time, a more systematic and comprehensive comparison of genes involved in GSL synthesis/breakdown was made to interpret the key features of GSL synthesis in horseradish. Highlights of the results are briefly summarized below:

1. Compared to other species, the number of key genes related to GSL synthesis in horseradish is advantageous, suggesting the genomic possibility of the strong synthetic ability of GSL in horseradish. Furthermore, we observed that the WGD and tandem duplicates led to the continuous expansion of GSL genes and that there were peaks of bursts at two-time points. We also observed distinct divergence of GSL genes between two subgenomes, suggesting the significant impact of subgenome evolution on the expression of this trait.
2. Horseradish accumulates GSLs dominated by sinigrin (an aliphatic GSL). In addition, many structurally distinct groups of GSLs have been identified in horseradish, indicating a possible more complex genetic basis of GSL biosynthesis. Through comparative genomic studies, we have found evidence supporting the above observations. An interesting example is that the gene cluster encoding flavin-containing monooxygenase (FMO-GSOXs, key enzymes associated with sinigrin biosynthesis) showed preferential retention after the polyploidization, was dominantly expressed, and tandemly duplicated in the B subgenome. Interestingly, we observed that *IGMTs*, key genes related to indole GSL synthesis, were lost in horseradish and its relatives, which may be one of the reasons for the low content of indole GSLs in horseradish.
3. Genes from the glycoside hydrolase (GH1) family (e.g., *TGGs* and *PEN2*) have been identified as genes encoding myrosinase and responsible for the breakdown or turnover of GSLs. We observed well-characterized divergence of GH1 family genes in horseradish, including loss of important gene branches, massive expansion, and differentiation. This also reflects the diversity of GSL in horseradish.

The number of myrosinase genes are mentioned (line 348) in connection with the rate of release of ITCs etc, and it is mentioned that 33 myrosinases were identified, which to this reader seemed like

the number from horseradish. However, the number of genes from horseradish in that category listed in table S2 is only 12, and they are all listed as homologs of PEN2. The PEN2 gene is considered associated with disease resistance and indole GSL turnover in intact cells but not release of ITCs after tissue fragmentation (ref the Clay et al. and Bednarek et al. twin papers on PEN2 in Science). So it is a bit of a mystery that no TGG homolog is identified, and I am not sure the current discussion (in results) is spot on. Line 353 “the reinforcement of the glucosinolate breakdown process...”, which reinforcement? A homolog of PEN2 is not a likely candidate for the classical myrosinase function, is it?

Response: We identified 33 myrosinases based on functional domains and sequence similarity, as detailed in our methods. In comparative genomic analysis, we selected homologous genes (syntelogs) in the collinear region. We believe that the syntelogs have better stability and comparative significance in terms of sequence similarity and conservation of function, although we know that they can also diverge between different species, so the number can be less than 33.

As for the PEN2, we acknowledge that the previous interpretation of the function of PEN2 family genes contained errors. This was now corrected and the results were reinterpreted.

An obvious number to discuss is the number of TGGs: 4, 6 and 4 in the three reference species, but zero in horseradish. Now THAT is intriguing, and not easily termed a “reinforcement”.

Response: Thank you for this comment. We have determined that the syntelogs gene of TGG has also been lost in horseradish and its close relatives. We have corrected our earlier interpretation in the updated version.

Is there any other known crucifer lacking a homolog of the TGGs? What about crucifers more closely related to horseradish, do they have TGGs? Has the TGGs been lost recently in horseradish?

Response: Thank you for this legitimate question. We performed an in-depth analysis of the close relatives of horseradish. TGG4 was lost in all four species of the tribe Cardamineae analyzed, while TGG1, TGG2, and TGG5 were lost in the ancestors of the three Cardamineae species (horseradish, *B. vulgaris* and *L. alabamica*) (Figures 8g-8h). This indicates that the loss of the TGG genes in horseradish was a gradual process that was not unique to horseradish and may also reflect the uniqueness of the GSL breakdown in horseradish or tribe Cardamineae.

I agree that a multiplication of the PEN family is intriguing, including the gene cluster map (Figure 9d), but I disagree that this can be automatically related to formation of the pungent allyl ITC after crushing horseradish roots.

Response: The GH1 family genes actually form a gene cluster in this region. In the updated version, we analyzed the gene cluster in Arabidopsis and horseradish in more detail and divided the homoeologous genes. We admitted that our understanding of the function of the PEN gene was inaccurate, and in the updated version we have corrected this.

The stress on the number of myrosinases is also troublesome in another way: The 12 myrosinases are catalogued as PEN2 homologs in Table S2, but according to Figure 9b only two of them can strictly be called closest homologs to PEN2, the remaining are closer homologs to a lot of other Arabidopsis glucosidases. With this observation, to me the entire stress on a high number of myrosinases in horseradish falls apart, and we are left with a plant with no classical myrosinase at all and only an average number of PEN2-like atypical myrosinases.

Response: Thank you for bringing the problem to our attention. After careful analysis, we found that the gene cluster in this region of horseradish has a more complex differentiation pattern. As shown in Figure 8i, the five members of the GH1 family (BGLU15, BGLU18, BGLU27, BGLU28, and PEN2) in Arabidopsis form a gene cluster, similar to that in horseradish. Based on the phylogenetic tree, we divided these GH1 genes in horseradish into different groups and suspected that the genes with the closer relationship are syntelogs.

The comments about epithioester protein (line 351) are mysterious as no such protein exists as far as I know.

Response: We removed the description of this protein in the updated version. We mean ESP in the previous version but now it is removed. Thanks very much.

Crude numbers of glucosinolate related proteins in the three reference species and horseradish is mentioned in Results (line 335-336), and the number for horseradish is numerically higher than the rest. However, assuming that the numbers were exhaustive, is this comparison fair? The numbers from the reference species have been pruned by decades of functional testing and mutation studies, so would seem to represent a corrected number, while I assume that the number for horseradish is the grand total of candidate homologs, some of them might have other functions.

Response: In the previous version, we had selected two species for comparison, and as you said, such comparison is not necessarily equivalent. To compensate for this bias, we have expanded the number of Brassicaceae genomes analyzed. We admit that we cannot exclude functional divergence of the homoeologous genes. Because the gene sequence and function of the syntelogs are more conserved, we selected the syntelogs for comparison, which makes the comparison more meaningful.

In Discussion, the numbers are not repeated but an attempt is made for a comparison of numbers within functional groups, e.g. breakdown and product diversification. As mentioned above, I think that discussion is flawed from the comparison of the PENs with the TGGs, but still I suggest that in a revised version, a table comparing numbers of GSL related genes /gene candidates should be subdivided into functional groups, to see whether there is a tendency for general doubling of gene numbers or a tendency for only a specific functional group proliferating in gene number. In this grouping, I suggest distinction of classical myrosinases like TGG and atypical or potential myrosinases like PEN, BGLU, BABG.

Response: Following to your suggestion, we have made a more detailed classification of the GH1 family in horseradish (Clade I~IV). Clades I and III harbor atypical myrosinases (PYK10, BGLU28, and BGLU30) that can hydrolyze GSLs to aglucon Clade II includes PEN2, which is associated with immunity and indole-GSL turnover in intact cells. In synteny analysis, we discovered an interesting gene expansion in Clade II and Clade III, which corresponds to the gene cluster (BGLU15, BGLU28, BGLU29, BGLU17, and PEN2) in *A. thaliana*. We analyzed the phylogenetic relationship of these genes (Figure 8j) and found that the PEN2 gene was preferentially retained after the polyploidization and even experienced tandem duplications (Figure 8j). The differentiation and expression of the gene cluster in subgenomes were also variable, e.g., the genes of Clade III were largely duplicated in the B subgenome but were lost in the A subgenome. Overall, we observed well-characterized divergence of GH1 family genes in horseradish, including loss of important gene branches, massive expansion, and differentiation. This also reflects the diversity of GSL substrates in horseradish.

If the authors will suggest a biological role of a possibly larger number of glucosinolate related genes in this species than in e.g. Brassica rapa and Arabidopsis thaliana, or from those specific genes found, some detailed basis in the current knowledge of horseradish would seem relevant. How does the battery of specifier proteins relate to known GSL products from the species? And what about relations of the biosynthesis battery to the GSL diversity in the species?

Response: Horseradish accumulates GSLs dominated by sinigrin (an aliphatic GSL) and gluconasturtiin (2-PE GSL). In addition, many structurally distinct groups of GSLs have been identified in horseradish. Through comparative genomic studies, we found evidence supporting the above observations in the revised manuscript.

The discussion of the chemistry of horseradish is almost devoid of phytochemical details and references (lines 402-402, is this true for all tissues and all life stages?). A larger perspective based in solid, published facts would seem to be relevant in the interpretation of the gene list.

Response: Thank you for this suggestion. Based on the references and notes you provided, we have referenced some references and content about horseradish chemicals in the updated version to make the article more comprehensive and to better interpret the gene list.

Line 403 what is “super-accumulation”, it is not a concept I have seen in the glucosinolate literature before.

Response: We have revised the manuscript to avoid the term.

The glucosinolate details in Discussion relate only vaguely to the details in Results, but mainly refer a handful of general results in the area. Glucosinolate results are absent from the Abstract. I feel that no clear conclusion on the large glucosinolate related gene dataset is arrived at.

Response: We included the GSL results in the updated abstract:
“Investigation of the genetic basis of glucosinolate (GSL) biosynthesis and horseradish

peroxidases revealed both the important role of polyploidization and subgenome evolution in shaping key characteristics. Continuous duplication and divergence of essential genes of GSL biosynthesis (e.g., *FMO-GSOX*, *IGMTs*, and *GH1* gene family) determined the GSL composition in horseradish.”

Conclusion of the review: You present a lot of interesting and valuable data, which will be a fantastic resource for future biologists and biochemists. Yet I suggest repeated, comprehensive search for glucosinolate-related genes, re-assessment and re-cataloging of suggested functions, re-design of the comparison with other species including rational choice of reference species, and re-analysis of the data. Finally a complete re-writing of the interpretation of glucosinolate related genes, aiming at specific conclusions related to previous knowledge on this chemically very well-characterized species.

Response: Thank you. Based on your suggestions, we improved the gene list and re-selected the species for comparison with the horseradish results. We have reinterpreted and carefully summarized the comparative genomic results.

Details.

Figure 9.

You miss two additional glucosinolate biosynthesis pathways relevant for horseradish, the combined biosynthesis of benzylGSL and branched chain amino acid-derived glucosinolates starting with CYP79C, and the biosynthesis of 2-phenylethylGSL starting with CYP79F1/2/6.

Response: Thank you for your comments. According to your suggestion, we have added benzylGSL and 2-PE GSL synthesis pathways. For convenience, we have simplified the main figure, but a more detailed synthesis pathway is presented as a supplementary figure.

Pathway from Met:

Between step 1 and 2: there is a transport to the chloroplast needed here.

Step 2. There are more genes involved than listed in the chain elongation cycle.

After step 2: there is transport out of the chloroplast needed here, and another aminotransferase step, which might or might not be catalyzed by the same enzyme as in step 1 (BCAT 3 was suggested to play a specific role here, Knill et al., 2008)

Compound five. Currently, this never isolated intermediate is suspected to be a nitrile oxide by many authors, are you sure you want to stick to the aci-nitro hypothesis?

S-alkyl thiohydroximate: In our days, with the alkyl known to be glutathione, this is a funny, outdated name for this intermediate, which I would prefer to term “glutathione conjugate”.

Response: Thank you. We added all functional genes, transcription factors, and transporters in the revised main figure and supplementary Figure/Table. In addition, the name errors have been corrected.

Pathway from Trp

You have mixed up the order of intermediates, also here the aci-nitro / nitrile oxide comes right after the oxime. If you wish to swap to nitrile oxide nomenclature, it would be 2-(indol-3-yl)ethanenitrile N-oxide. (N in italics).

The name S-indol-thiohydroximate is completely wrong, the indole group is certainly not on the S, here is the glutathione residue. I would call this intermediate "glutathione conjugate" as for the Met-pathway, because the complete name is quite long.

So, the order from Trp would be: oxime, nitrile oxide, glutathione conjugate, thiohydroximate, desulfoGSL, glucobrassicin.

From glucobrassicin, the intermediate to the left should be named 1HOI3M in analogy with 4HOI3M.

The genes to 1MOI3M are in contrast to the literature (Pfalz et al. 2016).

Not only PEN2 but also TGG hydrolyze indole glucosinolates.

Hydrolysis products: AITC is misleading, you mean isothiocyanates in general. The specifier proteins listed are put in wrong places. ESP means epithiospecifier protein! Thiocyanates are promoted by the thiocyanate forming protein. Etc.

Figure 9 legend: you mention numbers in brackets, but I see no brackets. Is the order of the species in the figure as explained in the legend?

Response: Thank you for pointing out the errors. We have modified the names of the corresponding intermediate products and updated the simplified version of the main and the full version of the attached figure.

Supplementary Table S2. I support the general table layout with original references for the genes listed and some sorting of the genes in classes. But some of the classes are misleading. Perhaps an additional column with a brief description of the biochemical function of each gene would be useful for the average reader? In particular if you wish to compare in detail with reference species, which seems to be the logic of this Table.

Response: Thank you. We have added simple descriptions of gene references and functions to the table you mentioned, in the hope that readers will better understand its.

The last heading epithioester protein is a mystery. Does that refer to ESP, which has another meaning? The name is not a good headline for the listed genes. Most of the other proteins listed are specifier proteins, this could be a good heading but does not fit all currently listed.

Response: We have removed "epithioester protein" in the updated table. We mean ESP in the previous version but now it has been removed. Thanks very much.

Missing genes/enzymes discovered since 2010 (a selection, there could be more)

TRANSPORT

GTR1 and GTR2 Search "Nour-Eldin" as author. The best place to start is this:

DOI 10.1016/j.tplants.2015.04.006, make sure you do not present the GTRs as exporters, they are

importers (to phloem companion cells).

THIOGLUCOSIDASES: BrBABG.a and BrBABG.b from B. rapa, Klein and Sattely 2017 DOI 10.1073/pnas.1615625114

BGLU 18, 28 and 30

PYK10 Nakano et al 2017 DOI

10.1111/tpj.13377

BIOSYNTHESIS:

Transport of alfa keto acids in and out of the chloroplast by BAT genes Petersen et al 2018 DOI 10.1111/jipb.12705

CYP79F6 from the same tribe as horseradish, relevant for GSLs in horseradish, Byrne 2017 DOI 10.1038/srep40728, functionally characterized by Wang et al 2021 doi 10.1111/tpj.15212

CYP79C1 and CYP79C2 functionally characterized by Wang et al 2020, relevant for GSLs in horseradish such as benzylGSL, DOI 10.3389/fpls.2020.00057

IGMT3, 4, 5 Pfalz 2017 DOI 10.1104/pp.16.01402 Pfalz et al 2011 DOI 10.1105/tpc.110.081711

Special modifications in Arabidopsis (acylation), is this gene in horseradish: SCPL17

HYDROLYSIS PRODUCTS:

The best place to start is here: Wittstock et al 2016 DOI 10.1016/bs.abr.2016.06.006

With this, you should be able to get the products and proteins matched in Figure 9, and arrive at some system in the last block of genes in Table S2.

Here are some Arabidopsis genes to consider: NSP3-5, ESM1,

Consultation of recent reviews is also recommended, the newest general GSL biosynthesis reviews I am aware of are:

Petersen et al 2018 DOI 10.1111/jipb.12705

Harun et al 2020 DOI 10.1021/acs.jafc.0c01916

Chhajed 2020 doi 10.3390/agronomy10111786

But beware of errors in the reviews, e.g. very wrong indication of the GTR transporters as exporters by Harun et al., a blunder you do not want to copy in your manuscript, and also a few connectivity errors in the Chhajed review (the combined biosynthesis overview figure)

Response: Based on your suggestion, we have refined and updated the gene list. Thanks very much.

There are quite many original papers on horseradish glucosinolates and isothiocyanates. I found 16 hits in web of sciences on “glucosinolate” AND “horseradish” in title, and 61 with these two terms in “topic”.

There is even a review of glucosinolate diversity in the tribe to which horseradish belongs, including the horseradish genus in their table 7: Olsen et al 2016 DOI 10.1016/j.phytochem.2016.09.013

Horseradish seems to be characterized by a few dominating GSLs and a quite large number of and structurally diverse set of GSLs. The profile of hydrolysis products has been reported by several authors, also using modern state of the art technology. Is there any relation of the known chemistry and the genes identified?

Response: Thank you comment. The GSLs in horseradish contain the highest amount of sinigrin and many other GSLs with different structures. In the revised manuscript, we succeeded in linking the GSLs in horseradish to genome evolution and explaining the features of GSL synthesis/breakdown to some extent.

And there are two or three genomes from that same tribe, Cardamine hirsute and Barbara vulgaris (and watercress, I think I heard it was prepared or submitted and could appear during the revision of your manuscript?)

Response: Thank you. In the updated manuscript version, we have included several species of the tribe Cardamineae/Brassicaceae species.

Response to Reviewer #2

The authors in this manuscript describe a new reference genome of horseradish by using the up-to-date Nanopore sequencing and Hi-C technologies. They further describe the dynamic characteristics of the allopolyploid subgenome and explored the potential genetic mechanisms underlying the traits. Overall, the new reference genome and further subgenome investigation will be of great value for polyploid genome evolution and horseradish breeding in the future. However, some of the results are largely descriptive and I still have some concerns as followed.

Considering these concerns, I recommend major revision.

Response: Thank you for your comments and suggestions. In the revised manuscripts, we have carefully addressed your concerns including conducting whole-genome bisulfite sequencing of different tissues and a more in-depth analysis of LTR-RTs. We have also addressed all minor concerns.

Major concerns:

1. *The authors need to recheck the analysis results carefully. For example, Line 114, the authors described the LAI value as 24, but Figure 2 shows the LAI value as 17. And, In Figure 2, the author described there are 39162 gene models, but they describe there are 40,290 protein-coding genes in main text (Line 119).*

Response: We apologize for these discrepancies between the preliminary and final versions of the genome assembly. In the revised manuscript, we double-checked the accuracy and consistency of all reported values.

2. *It seems to have some mistakes for the assembly pipeline and statistics. The assembly pipeline shown in the figure 2a, 2b do not correspond to the results. The authors described in results and methods that they mainly use the Juicer and 3D-DNA pipeline to perform the assembly while it was shown to be ALLHiC and Juicerbox in the fig. 2a. The contig N50 they show in the result is 17.5Mb while it became 7.95 Mb in the Supplementary table S2 and fig. 2a. The authors should carefully check those results and figures.*

Response: Thank you for bringing these inadequacies to our attention. We admit that some discrepancies arose from combining statistics for different versions of the genome assembly. All discrepancies have been corrected in the revised manuscript.

3. *The current analysis is too general. For example, the B subgenome had a higher level of methylation (~69.55%) than the A subgenome (63.49%). No details were further provided. And, there is a significant difference in the burst time between Copia and Gypsy LTRs. However, the authors didn't provide further explanations on it. I suggested the authors calculated the distribution of these newly formed LTRs and analyze the consequences of these newly inserted LTRs.*

Response: Thank you for your comments. We admit that some parts of the previous version, as

you mentioned, were not very detailed. In the revised manuscript, we have tried to focus more on the biological significance of our findings.

For methylation, we tried to investigate the methylation status together with the subgenome diversification and give a different perspective to the subgenome diversification by obtaining the methylomes of horseradish at single-base resolution. First, we systematically investigated the methylation status of subgenomes, including genic and repetitive regions. To further explore methylation differences between subgenomes, we identified the distinct methylation status between syntelogs and summarized subgenome methylation differences at the gene level. In addition, we have successfully linked epigenetic diversification to the dynamics of repetitive sequences and changes in genomic sequences. Overall, we take an incremental approach to elucidating the horseradish methylomes and subgenome diversification. In contrast to traditional approaches to analyze subgenome diversification, here we provide insights into DNA methylation and 3D chromatin structure generated by edge-cutting sequencing platforms. All this allowed us to systematically decipher the asymmetric evolution of horseradish subgenomes.

For the LTR-RTs, we first systematically identified the intact LTR-RTs in the subgenomes and characterized the insert time distribution patterns. Indeed, we observed that there was a younger burst peak in the *Copia* LTR-RTs and concluded that the differences were due to the difference in activity, which was confirmed by the following methylation level. Interestingly, based on the methylation profiling around LTR-RTs (Figures 4g-4i), we observed that the methylation levels around *Gypsy* LTR-RTs were always higher than around *Copia* LTR-RTs. According to your suggestions, we calculated the distribution of these newly formed LTR-RTs and found that there are many genes that have a rather small distance to the LTR-RTs, which possibly may influence the transcription of these genes (Figures S11-S12).

In the revised manuscript, studies on the formation of critical traits such as horseradish peroxidases and glucosinolate biosynthesis were improved. In particular, we systematically conducted the comparative genome analysis of the 11 representative Brassicaceae species and concluded that the number of key genes related to GSL synthesis in horseradish is increased. Further deciphering of the pathway provided us with the genomic basis of the GSL diversity in horseradish. Horseradish accumulates GSLs dominated by sinigrin (an aliphatic GSL) and gluconasturtiin (2-PE GSL). In addition, many structurally distinct groups of GSLs have been identified in horseradish, indicating a possible more complex genetic basis for GSL biosynthesis. Through comparative genomic studies, we have found evidence supporting the above observations. An interesting example is that the gene cluster encoding flavin-containing monooxygenase (FMO-GSOXs, key enzymes associated with sinigrin biosynthesis) showed preferential retention after the polyploidization, was dominantly expressed, and tandemly duplicated in the B subgenome. Interestingly, we observed that IGMTs, key genes related to indole GSL synthesis, were lost in horseradish and its relatives, which may be one of the reasons for the low content of indole GSLs in horseradish. Overall, the in-depth analysis of key horseradish traits provides us with a new understanding of the evolution of important traits in tetraploid plant genomes.

4. The methodology used is nevertheless not completely clear: no details have been provided in their method. For example, it is difficult for others to repeat their k-mer analysis and conclude that the horseradish genome had an allotetraploid origin. I strongly suggested that they deposited their scripts pipelines in github. This also ensures that the author's work can be

repeated. In addition, version of software and parameters should be provided in the method part.

Response: Thank you for your suggestion. In the revised manuscript, we have provided a more detailed description of the methodology including the version of the software and parameters. Also, all codes and pipelines have been uploaded to GitHub <https://github.com/maypoleflyn/horseradish-genome>.

5. *Concerning that the levels of methylation modifications are various in different tissues and the methylation data in this paper represents only one tissue type. It is recommended that they provide more data to support the results that 'DNA methylation play important role in subgenome diversification' such as methylation data from leaf root or flower.*

Response:

Thank you very much for your suggestions. At your request, we performed whole-genome bisulfite sequencing (WGBS) of different tissues. In the revised manuscript, we obtained single-base resolution methylomes of horseradish using Nanopore sequencing and WGBS. Based on the methylomes, we first systematically investigated the methylation status of subgenomes, gene regions, and repetitive regions, deciphered the consistent trends in methylation distribution in different tissues, and also revealed the differences in methylation levels between subgenomes. In addition to the above general observation, we further defined and explored the differentially methylated genes between subgenomes. Essential genes involved in many vital biological processes exhibited quick distinct methylation status, indicating epigenetic diversification between subgenomes. Furthermore, we linked methylation status to repetitive sequences and found that the changes in methylation status were due to repetitive sequences, indicating the crucial role of repetitive sequences in the shaping and diversification of subgenomes. Overall, horseradish methylomes at single-base resolution provide a new perspective for the study of subgenome changes, particularly for the complex, newly formed allotetraploid genome.

Minor concerns:

1. *Line 54-56, grammar mistake for double use of "is".*

Response: We have corrected this sentence (Line 59 in the revised manuscript).

2. *Line 80-81, I didn't find the genome survey results in table S1 as the authors described.*

Response: The results of the genome survey are presented in Figure 1 and we have added the reference in the revised manuscript (Line 85 in the revised manuscript).

3. *Line86-88: Please add a reference for this sentence.*

Response: We have added the reference for this sentence "For allotetraploids, a high proportion of aabb k-mers and a low proportion of aaab k-mers would be expected because preferential pairing would ensure that two homologs from the first subgenome and two homologs from the second subgenome are present after recombination" (Line 93 in the revised manuscript).

4. *Line214: Please show the exact number of methylated genes with LTRs.*

Response: We added the exact numbers of methylated genes associated with LTRs in the revised manuscript (Line 239 in the revised manuscript).

5. *Line 790-791, Here, it should be the length instant of the ratio.*

Response: We have changed it in the revised manuscript (Line 892 in the revised manuscript).

6. *Line 791, how to determine aaaa, aaab, aabc types of k-mers? The authors should describe it in the method part.*

Response: Thank you for your question. We conducted the K-mer analysis using jellyfish and Genomescope 2.0. First, we calculated the 21-kmers with jellyfish (v2.3.1) software using the Illumina short reads. Second, we run the Genomescope 2.0 software (“-k 21 -p 4 -m -1”), and then we obtained results including different types of k-mers. To make this part clearer, we have added the detailed parameters and the link to Genomescope 2.0 (Line 506 in the revised manuscript).

7. *In Table S4, I suggested adding the number of contigs.*

Response: We have added the numbers of the contigs in Table S4.

8. *The circos plot in Figure 2b is not well displayed, it appears to be shifted to the left.*

Response: The circos plot (Figure 1e in the revised manuscript) has been adjusted in the revised manuscript.

9. *The Hi-C contact map has a lot of gray lines and squares, which the authors should explain.*

Response: In the previous version, the gray lines and squares were caused by the removal of HiC-contact reads that linked the syntenic contigs. In the revised manuscript, the final HiC-contact map has been updated as Figure 1f.

10. *Figure 2C should include the chromosome id.*

Response: We included chromosome IDs in the HiC-contact map (Figure 1d in the revised manuscript).

11. *The composition of ‘Other’ in Figure 3a is not very clear. The author should clarify it.*

>>> The “Other” genomic composition included the “Non-protein-coding genes” (e.g., non-coding RNAs), unclassified repeat regions (“No repeat Category”), and genomic regions that cannot be annotated as genes/repeat/non-protein-coding genes. To clarify this, we added a detailed description in the figure legend (Lines 903-905 in the revised manuscript).

12. *Line 144, the sentence is incomplete.*

Response: Thank you. The sentence has been changed to “The long terminal repeat (LTR) retrotransposon Assembly Index (LAI) was up to 24, indicating that the assembly integrity was high even in the repeat-rich sequence regions” (Line 123 in the revised manuscript).

13. *I didn't find the description of Figure 3e in the main text. Although I found some interesting TE hotspots in Figure 3e.*

Response: We added a more detailed description of Figure 3e (Figure 2e in the revised

manuscript) in the main text (lines 151-152) “(Retro)transposons were not evenly distributed across the genome, and the hotspots were located in the centromere regions (Figure 2e)”.

14. I suggested adding Cretaceous–Paleogene boundary in Figure 4a.

Response: Thank you for your suggestion. In the latest version of evolutionary analysis, we include more representative Brassicaceae species, including species of the tribe Cardamineae, where horseradish is located, which also refines our time scale to the level of species differentiation. As the Cretaceous–Paleogene boundary is dated to 65-66 Mya, it has very limited relevance to the divergence of Brassicaceae lineages and the diversification of the Lineage I, including the Cardamineae. Therefore, the K-Pg boundary was not included in the revised Figure 4.

15. In Figure 4b, there is no description of the Venn diagram in Figure 4b.

Response: We have added references to the revised Venn diagram figure in the text of the revised manuscript, Line 181-184 “Compared to the other ten Brassicaceae species, we identified 8,205 gene families shared by all 11 species and 604 gene families specific to horseradish (Figure 3d).”

Response to Reviewer #3:

Overview: In this manuscript, “The allotetraploid horseradish genome provides insights into subgenome diversification and formation of critical traits,” Shen and colleagues provide a high-quality genome of horseradish (*Armoracia rusticana*) which is in the mustard family (*Brassicaceae*). Horseradish is a root vegetable that is used as a popular spicy condiment. It is known to possess interesting metabolites that give it a distinctive flavor (e.g., glucosinolates and horseradish peroxidases) and isothiocyanates that may be cancer-preventive. The genome has not been previously published to this reviewer’s knowledge and may be an important resource for plant breeders and future comparative studies.

Response: Thanks very much for your comments on our manuscript. Based on your suggestions, we have improved the manuscript substantially and all the concerns have been addressed.

Questions for the authors:

1. Plant materials. Are the seeds from the sequenced cultivar ‘HD15’ publicly available?

Response: Yes, the cultivar is publicly available.

2. Taxon sampling and consistency. How were the 15 outgroup genomes selected? It is not clear these species were chosen (both outside of the mustard family and the seven species within the mustard family seem). For example, there are many more *Brassicaceae* genomes available. On a related note, why were the various gene family analyses not done with the same species? For example, *Brassica rapa* was used to re-construct the groups of genes involved in glucosinolate biosynthesis; but that genome was not used in other comparisons.

Response: Thank you for your questions. In the revised manuscript, we selected 18 representative plant genomes including species in Poaceae (*Oryza sativa*), Vitales (*Vitis vinifera*), Rosaceae (*Fragaria vesca*), Asterales (*Actinidia chinensis*, *Olea europaea*, *Coffea arabica*, *Solanum tuberosum*), and 11 *Brassicaceae* species (*Brassica rapa*, *B. oleracea*, *Isatis indigotica*, *Thlaspi arvense*, *Arabis alpina*, *Barbarea vulgaris*, *Armoracia rusticana*, *Leavenworthia alabamica*, *Cardamine hirsuta*, *Arabidopsis thaliana*, and *Aethionema arabicum*).

We expanded the number of *Brassicaceae* genomes analyzed, taking into account the fact that only a limited number of crucifer genomes have been sequenced and their genomes are available. Our expanded selection followed two criteria: (i) we included species/genomes of the tribe Cardamineae, where *Armoracia* is placed, and (ii) genomes representing the major *Brassicaceae* lineages, i.e., Lineage I (Cardamineae), II (*Brassica*), III, IV, and *Aethionema* (Aethionemeae).

Tribe Cardamineae: *Armoracia rusticana* (horseradish), *Barbarea vulgaris*, *Cardamine hirsuta* and *Leavenworthia alabamica*.

Lin I is covered by *Arabidopsis* and 4 Cardamineae spp. listed above.

Lin II is covered by two *Brassica* spp.

Lin III: no genome is available.

Lin IV: *Arabis alpina*.

Aethionemeae: *Aethionema arabicum*.

As for consistency, in the earlier version, we only compared the glucosinolate biosynthesis in three species. In the revised manuscript, all 11 Brassicaceae species were used.

3. Novelty. It is not clear what the main novel results are given how the introduction and discussion are presented. For example, there is a lot of discussion on how much is learned about polyploid genome evolution; however, what is new (versus merely confirmatory) of what was already published in Mandakova and Lysak (2019)(cited reference 9)? Are there any new insights on sub-genome polyploid evolution? Similarly, what are the new findings about glucosinolate evolution that were not previously known? While confirmatory results are valuable, this reviewer struggled to find new insights.

Response: Thank you for these comments and concerns. In our study, we successfully obtained the high-quality genome assembly of the allotetraploid horseradish. We previously investigated the structure of the horseradish genome using comparative chromosome painting (Mandakova and Lysak (2019) Plant Physiology 179). While chromosome painting is an efficient method to infer (comparative) structure of crucifer genomes, it cannot provide sequence information or insights into gene evolution and function. The novelty of our research lies in analyses that go several steps beyond the macrostructure of the 16 chromosomes revealed by chromosome painting, in particular, our insights into GSL biosynthesis in horseradish.

In contrast to traditional approaches to analyze subgenome diversification, here we provide insights through a combined strategy, including DNA methylation and 3D chromatin structure generated by edge-cutting sequencing platforms. By investigating the asymmetric evolution of the subgenomes at different levels such as expression, gene loss, methylation level, and 3D chromatin structure, we concluded that the distinct and vital biological processes were affected by the different two subgenomes. We also establish the role of the dominant long terminal repeat retrotransposons and DNA methylation in the asymmetric evolution of subgenomes.

Regarding glucosinolate evolution, we conducted a more systematic and comprehensive comparison of genes involved in GSL synthesis/breakdown to interpret the key features of GSL synthesis in horseradish. Compared with other Brassicaceae species, the number of key genes related to GSL synthesis in horseradish was increased, explaining the GSL diversity in horseradish. Furthermore, we observed that the WGDs and tandem duplications led to the continuous expansion of GSL genes. The distinct divergence of GSL genes between two horseradish subgenomes suggests the significant impact of subgenome evolution on the expression of this trait(s).

Horseradish accumulates GSLs dominated by sinigrin (an aliphatic GSL). In addition, many structurally distinct groups of GSLs have been identified in horseradish. In our study, we found a genomic explanation for the GSL composition in horseradish. Through comparative genomic studies, we found evidence supporting the above observations. An interesting example is that the gene cluster encoding flavin-containing monooxygenase (FMO-GSOXs, key enzymes associated with sinigrin biosynthesis) showed preferential retention after the polyploidization, was dominantly expressed, and tandemly duplicated in the B subgenome. Interestingly, we observed that IGMTs, key genes related to indole GSL synthesis, were lost in horseradish and its relatives,

which may be one of the reasons for the low content of indole GSLs in horseradish. Genes from the glycoside hydrolase (GH1) family (e.g., *TGGs* and *PEN2*) have been identified as genes encoding myrosinase and responsible for the breakdown or turnover of GSLs. We observed well-characterized divergence of GH1 family genes in horseradish, including loss of important gene branches, massive expansion, and differentiation. Overall, we successfully linked the origin and evolution of the tetraploid horseradish genome with key traits of this root crop species.

4. Main figures. The manuscript presents extensive results (with nine main figures).

Could many of the results be put into supplemental materials?

Also, some key figure elements are outdated – for example, the phylogeny in Figure 1a is not useful; see Figure 1 in Mandakova and Lysak 2019 or more updated phylogenies (e.g., Walden et al. 2020. Nature Communications 11: 3795).

Response: Thank you. We have simplified the content of the main images and eight figures are presented in the main text now. The outdated phylogeny (now as Supplementary Figure 1) and the distribution map (Figure 1) were updated; figures summarizing GSL-related data and analyses were updated too.

5. References. References 29 and 30 from the Literature Cited are not cited in the main text. Also, some references are used that are inappropriate, for example, the introduction refers to polyploid evolution in cotton and Brassica but then cites a strawberry paper (reference 2). This occurs throughout the introduction and discussion where more recent or specific references would do the current literature justice.

Response: Thank you. We double-checked the literature cited throughout the manuscript and all the inappropriate citations have been corrected.

Overall, aside from the comments above, the methods are sound and meet the expected standards. The interpretations and conclusions are not wrong; however, their significance to the field and whether they are noteworthy for the Nature Communications audience is questionable. The high-quality annotated genome will be a valuable resource for comparative genomics across species, for population genomics (e.g., GWAS panel for horseradish), and for future functional investigations into specific pathways (e.g., horseradish isoenzymes).

Reviewers' Comments:

Reviewer #1:

Remarks to the Author:

I am generally very satisfied with the revision and the responses to my comments. I have no more comments to the general outline and logic. However, I am concerned about the concluded lack of TGG genes, and can find numerous small errors and communication problems in the GSL biosynthesis and biochemistry parts. Hence, I suggest a second, less extensive revision with focus on optimizing bits and pieces in the present framework. The framework as such is a scientific work of much importance and relevance for crucifer biology, biochemistry and breeding.

Major

Lack of TGGs in this and some related species. This observation is very interesting, but will cause quite some problems in our current understanding of GSL biochemistry. If reliable, it could also lead to renewed fundamental biochemical research in this species, which is well known to have a high potential conversion of GSLs to ITCs, i.e. high myrosinase activity. So far, in no plant species this general activation of GSLs is known to not happen catalyzed by a TGG enzyme, but I do agree that you have provided alternative candidates. Hence I suggest to reinspect the data once more, to be completely sure of your case before publishing. You suggest the loss to have happened before the split of several species, including *Barbarea vulgaris*. Liu et al 2016 (Front Pl Sci 2016 – vol 7 – article 83), in a transcriptome analysis of that species, reported several transcripts denoted "myrosinase", which would usually be understood as a synonym of TGG. I could not find any reference to TGG in that paper, but I suggest double checking this paper and underlying data for any TGG-like sequences. Could some lineage-specific change of sequence have led you to overlook TGGs also in horseradish?

Major related to GSL biosynthesis in Figures 8 and S32

Figure 8

Spelling errors CPY (-> CYP) and "chlproplast", "Tytophan". Please decide either capital first letters or not for metabolites.

The figure misses Leucine as precursor amino acid in horseradish. It has the exact same entry as benzylGSL, and the same Wang reference is valid. The figure also misses that for horseradish (but not any other crucifer), we have quite good evidence for general chain elongation of phenylalanine (not only to homophenylalanine but up to four cycles of elongation, so "chain elongated phenylalanine" would be better than just "homophenylalanine". The figure does correctly illustrate entry of the other four precursor amino acids, but to this reader it was initially confusing to see homophenylalanine standing "alone" in a visually more prominent place than the remaining.

You could redesign this to a more general figure. You could make the upper part of the figure with the chloroplast a restricted box and show chain elongation of methionine and phenyl alanine to chain elongated methionine and chain elongated phenylalanine, from left to right, as a common set of reactions for methionine and phenylalanine. This upper box could be called "Chain elongation" with white font.

A major box below could be called Core structure biosynthesis in white font.

You could replace homophenylalanine with "amino acid", with a box to the side listing the five precursors: Chain elongated methionine, Chain elongated phenylalanine, Leucine, Phenylalanine, Tryptophan. I would introduce a reaction arrow from desulphoGSL to "parent glucosinolate".

In the present box with yellow and orange sub-boxes, I would put with white font "Decoration of selected parent glucosinolates" .

In the bottom box, I would put "glucosinolate" as starting point of the reaction arrow, and place all five gene classes at the reaction arrow (from PYK10 to PEN2), without visual distinction of PEN2. This box named "Breakdown" in white font as present.

In summary, the suggested re-design would have four boxes, with names in white font (counted from above): Chain elongation, Core structure biosynthesis; Decoration of selected parent glucosinolates; Breakdown.

With the suggested design, there is no need to single out benzylGSL and 2-phenylethylGSL as presently, so these names could be deleted from the decoration box.

Figure S32

In figure S32, there are more biochemical errors. I wish I could point and draw, but here is a one page text attempt to point out the problems. Sorry.

Spelling: Chloroplast! (o missing). IMDH1 in both chloroplasts should be IPMDH1. Tryptophan with r. Sulfate and sulfite with f: place make a choice of either US or British spelling, if you write desulphoGSL with pH you must also write sulphate with ph, I think. If the journal is British, there may be a standard. 3-phosphosulfate5'-phosphosulfate lowermost in both chloroplasts: you have the same f / ph problem, but more important, this metabolite is called 3-phosphoadenine-5'-phosphosulfate (note adenine!). The abbreviation is PAPS, which has wrongly been put as an enzyme name next to a reaction arrow, so delete red font PAPS in both chloroplasts.

Synonyms: 2-keto acid is a synonym of 2-oxo acid, so you should choose. I suggest 2-keto acid also in the right side chloroplast.

Colors, logic etc: sulfate enters twice in each chloroplast, the second can be deleted in both. The APS3 enzyme is wrongly in black font once in each chloroplast. "GSH" in red font has been forgotten in the middle of each chloroplast, has no meaning there. In two biosynthesis to the right, the E configuration has been given for the aldoxime, but not in the two to the left. I suggest leaving out the "(E)" also in the right hand side. (if you keep it, it should be changed to italics).

Curved lines starting from gamma-Glu-Cys-Gly in chloroplasts: these are arrows signifying transport of Glu-Cys-Gly out of the chloroplast. But the lines end in the wrong place, and miss an arrow. They should end not at nitrile oxide but at the reaction arrow from nitrile oxide to glutathione conjugate. The red "GSH" along the line/arrow should be deleted, as GSH is not a transporting enzyme but the abbreviation of gamma-Glu-Cys-Gly. (You were probably misled by the absurd graphical symbols used in the Harun et al review figure).

The lines from 3-phosphoadenine-5'-phosphosulfate to desulphoGSL (no hyphen!) should also be arrows, so they each miss an arrowhead.

Below, I suggest corrections that will make the figure broader. To find space for this, I suggest splitting figure 32 in two, still on the same page.

The caption will be changed to: The biosynthesis of glucosinolates (GSLs) includes the methionine-derived and indolic (a) and benzyl, branched-chain amino acids and 2-phenylethyl (b) pathways in *A. thaliana*.

Figure S32a is identical to the present "Aliphatic GSL" and "Indolic GSL" part.

Figure S32b is identical to the twin part presently at the right hand side. The move to below is needed because some reactions have been left out.

In figure S32b, The enzymes channeling phenylalanine to benzylGSL also channel branched chain amino acids (valine and leucine) to branched chain aliphatic GSLs. So the heading "Benzyl GSL" should be changed to "Benzyl and branched chain GSLs".

In the biosynthesis below, the starting amino acid should be changed from "Phenylalanine" to "Phenyl alanine and branched chain amino acids". For this reason, some metabolite names below should be generalized, but most are already general. The (E)-2-phenylethanal oxime: change to aldoxime. From there, all is good until: Desulfo benzyl GSL: change to DesulfoGSL (or Desulfo GSL if you prefer a

space).

End point: Benzyl GSL, 1-methylethylGSL, 2-methylpropylGSL.

Due to these changes, the Phenylalanine used for transport to the chloroplast should be a separate word, not the same used for the previously discussed pathway. From that new Phenylalanine, there should be an arrow in the white space to "2-keto acid". Above this arrow BCAT4 and BCAT6 in red exactly as for Methionine in Fig S32a. From this new "2-keto acid" there should be an arrow showing transport over the chloroplast membrane (BAT5 in red above this arrow). The arrow ends in the same name (2-keto acid) now within the chloroplast (as presently). The next three reaction arrows are kept as presently, but above the fourth arrow, symbolizing transport out of the chloroplast, the red "BCAT3" should be deleted. You are right that there are controversial evidence for the possible involvement of BCAT3 in the biosynthesis of the methionine derived GSLs (Figure S32a), but not for the benzyl and 2-phenylethylGSLs.

Also in this Fig S32b, remember editing the curved and straight lines leading from the chloroplast, as explained for Fig S32a.

Minor

Line 29: This line in a way states the obvious. I would rephrase to: .. family) may have caused the unusually broad GSL profile of horseradish.

Line 61. A single GSL and GSLs in general seems to be mixed. I would rephrase to: ... glucosinolates (GSLs), which are hydrolysed by thioglucosidase enzymes to produce isothiocyanates with..

Line 63. Reference 17: is this particular paper suitable as single reference for such vast literature?

Line 63 after dot. IF you wish to include sinigrin in the introduction, it could be done with a sentence here, such as: "The dominant GSL in horseradish is sinigrin (allylGSL). It is the precursor of allyl isothiocyanate, the main responsible for the pungent aroma of horseradish and wasabi."

Line 77: GSLs

Line 164. It is confusing to variously used tribes and genera here. Should Brassica be Brassiceae?

Line 183: I wondered: what is the definition of "gene family" used here? In general biochemistry, as far as this reviewer knows, gene families are common to multiple species, genera, families and orders, and not usually species specific.

Line 353-362: I find this section a bit unprecise. I would put it as follows: "In the tribe C., most of the known biosynthetic groups of GSLs are well-established (16, 36-38). [Move this sentence to here: Three independent steps (1) (2) (3) ... (37)]. Although the GSL profile of horseradish is dominated by sinigrin, the entire profile is remarkably wide, including most of the biosynthetic groups of the tribe (36, 39, 41). Three groups of horseradish GSLs include chain elongation: short chain methionine-derived, long chain methionine-derived, and chain-elongated phenylalanine derived. A further two groups occur independent of chain elongation: tryptophan derived ("indole GSLs") and the combined group of benzyl GSL and branched chain GSLs, which were recently discovered to depend on a committed step catalyzed by CYP79C enzymes (36, 90). The unusual biosynthetic diversity suggests a similarly complex array of biosynthetic genes. We reconstructed the GSL.... (keep the existing text from here)

Line 376-383: I think this bit is imprecise at places, and suggest the following:

GSL names: remember that in chemical names, hyphens are ONLY used before and after numbers or atomic symbols, and that atomic symbols in names should be in italics. Line 376 O-methyltransferase: italic O.

Line 376-377: methoxylation is catalyzed in two steps, first a hydroxylation, then a methylation.

Reword e.g. as follows: "The final step in indole GSL methoxylation is methylation catalyzed by indole GSL O-methyl transferases (IGMTs). IGMT 1-4 produce 4-methoxyindol-3-ylmethylGSL (4MOI3M) (43), while IGMT 5 produces 1-methoxyindol-3-ylmethylGSL (1MOI3M) (Pfalz et al., 2016 Plant Physiology). The syntelogs of"

Line 382-383: The references are only on horseradish, but anyway I don't see the literature stating a particularly low level of indoles in Barbarea. I would reword the last sentence as this: The loss of IGMT 5 may explain the absence of 1MOI3M in horseradish and its presence in Barbarea (36, 39).

Line 404-405: I would insert ". Possibly" as follows: "... substrates in horseradish. Possibly, the differences in substrates led to...."

Line 423: add "agronomists" or "breeders"?

Line 466-467: I suggest a slight adjustment to ". Cardamineae, e.g. the ancient loss of TGGs and very recent loss of IGMT5. The evolution ... GH1 family may reflect the diversity...

Reviewer #2:

Remarks to the Author:

The authors in this newest version of the manuscript have responded to all my questions in the previous review and I am satisfied with most of the answers and revisions. However, I have still felt that it is sometimes hard to get the main points from many parts of the results, either because of the quite descriptive findings or because the analysis for supporting some conclusions are inadequate. Below are some concerns about this revised manuscript.

Major concerns:

1. Inadequate information for Hi-C scaffolding in Figure 1f. It seems many small segments of the chromosome do not show any Hi-C contact information with any other part of the genome. The Hi-C contact maps for each chromosome couldn't be found in the supplementary figures for checking the details. The authors should carefully check those segments. In addition, they should give a complete Hi-C contact map of all the estimated genome including those unanchored contigs for checking the quality of scaffolding process.

2. Some of the results lack adequate analysis. Line 151-152, the authors concluded that LTR hotspots are located in centromere regions, but I didn't observe that the authors annotated the centromere regions from the genome. Besides, simply performing GO and KEGG analysis of genes around LTR retrotransposons (a) and DNA transposons (b) (as shown in Figure S12) does not provide enough evidence to support the conclusion of a possible influence of repeats on gene expression. More examples of these issues are presented in the comments below.

3. The present manuscript contains numerous errors, some of the numbers in the main text do not match with the tables and figures provided by the author. And, some of the tables and figures referenced in the main text do not match to the corresponding results. I pointed out many such issues in the subsequent comments. I remembered that this point was highlighted in the previous manuscript review.

4. The current analysis is still overly descriptive and lacks explanations for the results. For example, the authors mention a significant difference in burst time between Copia and Gypsy LTRs without providing further elaboration. Additionally, they note an enrichment of certain TE families based on phylogenetic analysis, but they only make a general conclusion that the rapid expansion of certain families played an important role in genome expansion, without providing any evidence to support this conclusion. I remember that I pointed out these issues in the previous review, but the resubmitted manuscript does not seem to address them adequately.

5. The analysis for "lost genes" should be reorganized and more clearly explained. The authors should provide a more compelling explanation for why they consider these genes to be "lost" rather than newly formed on the A and B subgenomes after speciation.

Minor concerns:

6. The reference on line 119-120 is incorrect as Table S6 does not provide the mapping ratio of Illumina and Nanopore reads.

7. I noticed that the mapping ratio of Nanopore reads is 99.55% instead of 95.55%. I suggest that the authors should carefully check the figures displayed in the present manuscript for any inaccuracies.

8. The circos plot in Figure 1e is not clear. For example, the circular diagram of gene expression appears to have many maximum values and the author does not provide any description in the legend.

9. Table S14 was not presented in the main text.

10. Line 97, 24Gbp of Illumina paired-end short reads do not match the 88-fold genome coverage given the reference genome is estimated around 610Mb.
11. Line 148-149, This conclusion is too broad and lacks any results to support it.
12. Line 152, This conclusion appears to be based on the author's subjective belief rather than evidence, as there is no supporting data or results that demonstrate the hotspots were located in the centromere regions.
13. Line 153, It is not clear about the 'certain families' here.
14. Line 153-154, the authors cannot conclude that the 'certain families' played an important role in genome expansion. They didn't conduct analysis between the 'certain families' and genome size.
15. Line 155, The Figure S11 does not illustrate that "nearly 3000 genes were predicted to be located in the vicinity" as stated in the text. There is no clear connection or correlation between the results presented in the text and the data displayed in Figure S11.
16. Line 158, There is no data or results provided to demonstrate 'a possible influence of repeats on gene expression'.
17. L155-158, the study focuses on genes with intact transposable elements (TEs) inserted in the flanking regions, but it's not clear why the authors choose to limit their analysis in this way. As the intact TE only constitutes a small proportion of the total transposons.
18. Line 174, The authors should indicate it clearly in the Figure 3b, so that the readers can easily understand the information and get the correct interpretation of the figure.
19. Line 175, the result is too general. There is no detail about the existence of segmental duplications contribute to the additional Ks peak.
20. Line 181-182, There is an obvious error here. The reference of Figure 3b should not appear here.
21. Line 183-184, It appears that there is an error in the text and Figure 3d, where the number of gene families described in the text does not match the number shown in the figure. The figure shows 8208, but the authors described 8205 gene families.
22. Line 183-184, Why are 11 species described here? They employed 18 species in the present study.
23. Line 195-195, There is an obvious error here. The reference of Figure S15-S17 should not appear here.
24. Line 365-367, it appears that there is a discrepancy in the reference numbers given in the text. The first sentence refers to Table S23, but the second sentence immediately shifts to Table S33-35. This inconsistency makes it difficult to follow the information provided and understand the context.
25. Line 302-303, the sentence is too general. I only observed a different number of 'lost genes' between A and B subgenomes.
26. Line 317-318, I suggest stating here that there are 3 more genes on the scaffold.

Reviewer #3:

Remarks to the Author:

This is a revised manuscript, "The allotetraploid horseradish genome provides insights into subgenome diversification and formation of critical traits". I was reviewer 3 on the original submission and am satisfied that most of my concerns have been addressed. However, reviewer 1 and reviewer 2 had more critical and substantial concerns of the original submission so I bow to their reviews of this revision.

Reviewer #1 (Remarks to the Author):

I am generally very satisfied with the revision and the responses to my comments. I have no more comments to the general outline and logic. However, I am concerned about the concluded lack of TGG genes, and can find numerous small errors and communication problems in the GSL biosynthesis and biochemistry parts. Hence, I suggest a second, less extensive revision with focus on optimizing bits and pieces in the present framework. The framework as such is a scientific work of much importance and relevance for crucifer biology, biochemistry and breeding.

Response: Thank you once again taking the time to provide us with valuable feedback on our revised manuscript. We have carefully reviewed your feedback and revised accordingly.

As for the GSLs, we have conducted thorough proofreading, and addressed the minor errors and communication issues you pointed out. Major revisions include redrawing Figure 8 and Figure S42 (the former Figure S32), and correcting the inaccurate chemical names.

In response to the conclusions regarding TGGs (or any other lineage-specific gene losses), we conducted a thorough investigation, focusing primarily on improving the quality of the genome assembly and the specificity of sequence differences. We used a new sequencing platform for genome reassembly and annotation, resulting in more comprehensive genome assembly. Building on the improved genome assembly, we systematically re-classified the glycoside hydrolase 1 (GH1) gene family rather than relying solely on syntelog results (see detailed information below). By correcting previous inaccuracies, we present more precise and reliable conclusions. We sincerely hope that these improvements will meet your expectations.

Major

Lack of TGGs in this and some related species. This observation is very interesting, but will cause quite some problems in our current understanding of GSL biochemistry. If reliable, it could also lead to renewed fundamental biochemical research in this species, which is well known to have a high potential conversion of GSLs to ITCs, i.e. high myrosinase activity. So far, in no plant species this general activation of GSLs is known to not happen catalyzed by a TGG enzyme, but I do agree that you have provided alternative candidates. Hence I suggest to reinspect the data once more, to be completely sure of your case before publishing. You suggest the loss to have happened before the split of several species, including *Barbarea vulgaris*. Liu et al 2016 (Front Pl Sci 2016 – vol 7 – article 83), in a transcriptome analysis of that species, reported several transcripts denoted “myrosinase”, which would usually be understood as a synonym of TGG. I could not find any reference to TGG in that paper, but I suggest double checking this paper and underlying data for any TGG-like sequences. Could some lineage-specific change of sequence have led you to overlook TGGs also in horseradish?

Response: We greatly appreciate the careful questions and suggestions you have made. We also realize that we need to be careful with the issues related to gene loss, especially in a polyploid genome. Our primary concern was and is the quality of our genome assembly. Despite using comprehensive strategies to address challenges such as assembling polyploid chimeras, we were limited by the inherent single-base errors of Nanopore sequencing. Consequently, we cannot completely avoid some gaps or losses in the assembly as described in originally submitted

manuscript version, in which we identified missing large genomic segments that were however confirmed by FISH (Fluorescence In Situ Hybridization) experiments (Figures S6, S7, and Supplementary Note 1). The final evaluation of the gene set yielded a BUSCO score of 95.56% for ONT-based genome assembly, indicating room for improvement.

In the current manuscript version, we implemented an integrated strategy combining multiple platforms, including PacBio HiFi, ONT, HiC-seq, and Illumina, to achieve more comprehensive genome assembly (Supplementary Notes 1). For the PacBio HiFi reads, our preliminary assembly generated 108 contigs with an N50 length of 33.74 Mbp (Table S5). Leveraging the accuracy of HiFi long reads and the high continuity of the assembly, we directly performed scaffolding, resulting in a genome assembly comprising 16 pseudochromosomes that anchored 97.25% of the assembled sequence, leaving only 8 gaps (Table S6). Notably, nine of the 16 pseudochromosomes consisted of a single contig, highlighting the advantage of high-accuracy HiFi reads in resolving polyploid complex genomes (Table S6). Considering the superior continuity and integrity of the HiFi-based genome, we selected it as the genome backbone and filled the remaining gaps with ONT-based genome sequences. As a result, we successfully obtained a telomere-to-telomere (T2T) gap-free assembly of the horseradish genome. This is the first T2T assembly of an allotetraploid genome in the Brassicaceae family. Subsequent genome assessments confirmed the improved contiguity and completeness of the T2T genome assembly. For instance, the BUSCO score of the genome assembly was up to 99.44%, which was significantly higher than that of the ONT-based genome assembly (Table S9).

We extended the analysis to all GH1 members instead using only the syntelogs. Based on the complete genome assembly, we conducted a systematic analysis of the GH1 gene family in the 11 crucifer genomes and named the clades (I-X) based on homoeologous genes in *A. thaliana* (Xu et al., 2004). In contrast to previous observations, we identified the TGGs clade in the T2T gap-free genome assembly of horseradish. In addition, we followed the reason for the failed identification of TGGs in our previous study (ONT-based assembly). We re-identified all GH1 genes in the ONT-based assembly and constructed a phylogenetic tree using all identified GH1 genes and those from *A. thaliana*. The genes in the ONT-based genome assembly still did not cluster with TGGs in *A. thaliana* (Figure R1, below). We extracted two genome sequences that encompassed the TGG gene cluster in the T2T genome assembly and attempted to align them with the ONT-based genome assembly. However, we could not detect the two genomic segments accurately in the ONT-based assembly. This observation implies that the ONT reads have limitations in assembling tandemly duplicated genes. To verify the TGG genes in the horseradish genome, we aligned the HiFi long reads to T2T genome and visualized the alignment using IGV (Integrative Genomics Viewer). We could see that the genome sequences covering TGGs could be accurately aligned by the HiFi reads (Figures R2-R3). The RT-qPCR experiment also supported the expression of TGG genes (Figure R4). Therefore, based on the latest findings, we revised the results related to gene losses in the current manuscript version.

Due to the important role of the GH1 gene family in GSL metabolism, we conducted a systematic classification of the GH1 gene family based on the latest genome assembly results. Among the 11 Brassicaceae genomes analyzed, horseradish stood out with the highest number of GH1 genes (85 genes) in the tribe Cardamineae. Clade VII harbored TGGs encoding classical myrosinase, with 27 genes in horseradish, making it the largest GH1 gene clade among the studied genomes. Interestingly, significant gene duplication events were observed in both main lineages

(TGG1/2 and TGG4/5) of Clade VII in horseradish. Additionally, distinct expression patterns were found for genes encoding classical myrosinase in horseradish compared with *A. thaliana*, with tissue-specific expression and differential expression levels observed in different lineages. For the genes encoding the atypical myrosinase (BGLU18-BGLU33), we obtained similar result as for ONT-based assembly. PEN2 is known for its involvement in immunity and indole GSL turnover in intact cells. By synteny analysis, we made an interesting discovery: a gene expansion in Clade IV across both subgenomes (a04:1.73-1.67Mb; b04:1.77-1.69Mb), corresponding to the gene cluster BGLU15, BGLU28, BGLU29, BGLU17, and PEN2 in *A. thaliana*. Analyzing the phylogenetic relationship of these genes, we found that the duplicated PEN2 genes were preferentially retained after the polyploidization event. We also analyzed the homoeologous GH1 genes that showed differential expression between the two subgenomes, with the majority of these genes belonging to Clade IV. The overall divergence of GH1 family genes in horseradish, including massive expansion and differentiation, reflects the diversity of GSL substrates in horseradish.

Similarly, we have examined and validated the absence of the *IGMT* gene in horseradish. In the latest version of the genome, we found that *IGMT* is not deleted but rather underwent a tandem duplicated event. We visualized the alignment of HiFi reads for these genes and selected specific genes for verification using RT-qPCR (Figures R4-R5).

We apologize for the changes in some of our conclusions due to improvements of the genome assembly. To ensure accuracy, we performed a careful verification and made the necessary modifications. We thank the Reviewers for their suggestions that eventually led to an improved horseradish genome assembly.

Figure R1. The phylogenetic tree of all identified glycoside hydrolase 1 (GH1) genes in the ONT-based genome assembly of horseradish and *Arabidopsis thaliana*. The TGG clades are marked in red. The gene names starting with “Arr” were identified in the horseradish genome assembly, whereas the remaining genes belong to the *Arabidopsis thaliana* genome.

Figures R1~R5 are only included in the Response.

Figure R2. Visualization of PacBio HiFi reads mapped near the selected TGG genes in the T2T gap-free horseradish genome assembly. TGG genes (*Arr20386*, *Arr20387*, *Arr20388*, *Arr20361*, *Arr20362*, *Arr20363*, *Arr20364*, *Arr20365*, *Arr20366*, and *Arr20367*) are marked with red.

Figure R3. Visualization of PacBio HiFi reads mapped near the selected TGG genes in the T2T gap-free horseradish genome assembly. TGG genes (*Arr38440*, *Arr38441*, *Arr38451*, *Arr38452*, and *Arr38453*) are marked with red.

Figure R4. Quantitative reverse transcription PCR (RT-qPCR) of randomly selected TGG and IGMT genes.

Figure R5. Visualization of PacBio HiFi reads mapped near the selected IGMT genes in the T2T gap-free horseradish genome assembly. IGMT genes (*Arr21774*, *Arr21771*, *Arr21773*, *Arr21772*, *Arr4259*, *Arr4260*, *Arr4258*, and *Arr4261*) are marked with red.

Major related to GSL biosynthesis in Figures 8 and S32

Figure 8

Spelling errors CPY (-> CYP) and “chlproplast”, “Tytophan”. Please decide either capital first letters or not for metabolites.

Response: Thank you for pointing out these errors. They were revised accordingly.

The figure misses Leucine as precursor amino acid in horseradish. It has the exact same entry as benzylGSL, and the same Wang reference is valid. The figure also misses that for horseradish (but not any other crucifer), we have quite good evidence for general chain elongation of phenylalanine (not only to homophenylalanine but up to four cycles of elongation, so “chain elongated

phenylalanine” would be better than just “homophenylalanine”. The figure does correctly illustrate entry of the other four precursor amino acids, but to this reader it was initially confusing to see homophenylalanine standing “alone” in a visually more prominent place than the remaining.

Response: Thank you very much for your professional input. We have revised our statement to "chain elongated phenylalanine".

You could redesign this to a more general figure. You could make the upper part of the figure with the chloroplast a restricted box and show chain elongation of methionine and phenyl alanine to chain elongated methionine and chain elongated phenylalanine, from left to right, as a common set of reactions for methionine and phenylalanine. This upper box could be called “Chain elongation” with white font.

Response: Thank you for your suggestion. We have redesigned the figure based on your suggestion.

A major box below could be called Core structure biosynthesis in white font. You could replace homophenylalanine with “amino acid”, with a box to the side listing the five precursors: Chain elongated methionine, Chain elongated phenylalanine, Leucine, Phenylalanine, Tryptophan. I would introduce a reaction arrow from desulphoGSL to “parent glucosinolate”. In the present box with yellow and orange sub-boxes, I would put with white font “Decoration of selected parent glucosinolates”. In the bottom box, I would put “glucosinolate” as starting point of the reaction arrow, and place all five gene classes at the reaction arrow (from PYK10 to PEN2), without visual distinction of PEN2. This box named “Breakdown” in white font as present.

Response: Thank you for your suggestion. We have revised the figure based on your suggestions.

In summary, the suggested re-design would have four boxes, with names in white font (counted from above): Chain elongation, Core structure biosynthesis; Decoration of selected parent glucosinolates; Breakdown.

Response: Thank you for your suggestion. We have revised this figure accordingly.

With the suggested design, there is no need to single out benzylGSL and 2-phenylethylGSL as presently, so these names could be deleted from the decoration box.

Response: Thank you. We have revised this figure accordingly.

Figure 8. Genome-wide comparison of genes involved in glucosinolate (GSL) metabolism pathways in horseradish and its relatives.

(a). The simplified diagram of GSL biosynthetic pathways for the biosynthesis of aliphatic GSLs, indolic GSLs, benzyl and branched-chain GSLs, and 2-phenylethyl GSLs. (b). The varying copy number of syntenic involved in GSL biosynthesis in 11 Brassicaceae species, including *Arabidopsis thaliana* (Ath), *Barbarea vulgaris* (Bvu), *Cardamine hirsuta* (Chi), *Leavenworthia alabamica* (Lal), *Armoracia rusticana* (Aru), *Arabis alpina* (Aal), *Brassica rapa* (Bra), *B. oleracea* (Bol), *Isatis indigotica* (Iin), *Thlaspi arvense* (Tar), and *Aethionema arabicum* (Aar). (c). K_s value distribution of duplicated GSL genes in the horseradish genome. (d). K_s value distribution of the tandemly duplicated GSL genes in the horseradish genome. (e). Microsyntenic visualization of flavin-containing monooxygenase (FMOGS-OX) gene clusters in horseradish and *Arabidopsis thaliana*. (f). Microsyntenic visualization of methylthioalkylmalate synthase enzymes (MAM) genes in horseradish and *Arabidopsis thaliana*. (g). Phylogenetic tree of beta-glucosidase genes in

horseradish and *Arabidopsis thaliana*, and the gene expression heatmap (right). Roman numerals refer to the names of different gene clades. (h) and (j). Microsyntenic visualization of myrosinase (*TGG*) genes (h) and *PEN* genes (j) in horseradish and *Arabidopsis thaliana*. The trees indicate the phylogenetic relationships between the analyzed species, and the predicted gene numbers are indicated below the species acronyms. (j) and (k). *Ks* value distribution of duplicated *TGG* genes (j) and *PEN* (k) genes in the horseradish genome.

Figure S32

In figure S32, there are more biochemical errors. I wish I could point and draw, but here is a one page text attempt to point out the problems. Sorry.

Response: Thank you for investing your time in providing detailed corrections for our figure. This is greatly appreciated. We have carefully considered your suggestions and have changed every detail of the illustration accordingly. We are grateful for the opportunity to improve our work based on your comments.

Spelling: Chloroplast! (o missing). IMDH1 in both chloroplasts should be IPMDH1. Tryptophan with r. Sulfate and sulfite with f: please make a choice of either US or British spelling, if you write desulphoGSL with pH you must also write sulphate with ph, I think. If the journal is British, there may be a standard. 3-phosphosulfate5'-phosphosulfate lowermost in both chloroplasts: you have the same f / ph problem, but more important, this metabolite is called 3-phosphoadenine-5'-phosphosulfate (note adenine!). The abbreviation is PAPS, which has wrongly been put as an enzyme name next to a reaction arrow, so delete red font PAPS in both chloroplasts.

Response: Thank you for these corrections. We have corrected the spelling errors and inaccuracies in the names. We appreciate your attention to detail and the opportunity to improve the accuracy of statements.

Synonyms: 2-keto acid is a synonym of 2-oxo acid, so you should choose. I suggest 2-keto acid also in the right side chloroplast.

Response: Thank you for your suggestion. We have revised to "2-keto acid" accordingly.

Colors, logic etc: sulfate enters twice in each chloroplast, the second can be deleted in both. The APS3 enzyme is wrongly in black font once in each chloroplast. "GSH" in red font has been forgotten in the middle of each chloroplast, has no meaning there. In two biosynthesis to the right, the E configuration has been given for the aldoxime, but not in the two to the left. I suggest leaving out the "(E)" also in the right hand side. (if you keep it, it should be changed to italics).

Curved lines starting from gamma-Glu-Cys-Gly in chloroplasts: these are arrows signifying transport of Glu-Cys-Gly out of the chloroplast. But the lines end in the wrong place, and miss an arrow. They should end not at nitrile oxide but at the reaction arrow from nitrile oxide to glutathione conjugate. The red "GSH" along the line/arrow should be deleted, as GSH is not a transporting enzyme but the abbreviation of gamma-Glu-Cys-Gly. (You were probably misled by the absurd graphical symbols used in the Harun et al review figure).

The lines from 3-phosphoadenine-5'-phosphosulfate to desulphoGSL (no hyphen!) should also be arrows, so they each miss an arrowhead.

Response: Thank you for your corrections. We have revised the figure based on your comments.

Below, I suggest corrections that will make the figure broader. To find space for this, I suggest splitting figure 32 in two, still on the same page.

Response: We redesigned the figure based on your comments.

The caption will be changed to: The biosynthesis of glucosinolates (GSLs) includes the methionine-derived and indolic (a) and benzyl, branched-chain amino acids and 2-phenylethyl (b) pathways in *A. thaliana*.

Response: Thank you for your suggestion. We revised the caption accordingly.

Figure S32a is identical to the present “Aliphatic GSL” and “Indolic GSL” part.

Response: The figure was revised accordingly.

Figure S32b is identical to the twin part presently at the right hand side. The move to below is needed because some reactions have been left out.

Response: This was revised accordingly.

In figure S32b, The enzymes channeling phenylalanine to benzylGSL also channel branched chain amino acids (valine and leucine) to branched chain aliphatic GSLs. So the heading “Benzyl GSL” should be changed to “Benzyl and branched chain GSLs”.

Response: Thank you for your suggestion. We have changed the “Benzyl GSL” to “Benzyl and branched chain GSLs” accordingly.

In the biosynthesis below, the starting amino acid should be changed from “Phenylalanine” to “Phenyl alanine and branched chain amino acids”. For this reason, some metabolite names below should be generalized, but most are already general. The (E)-2-phenylethanal oxime: change to aldoxime. From there, all is good until: Desulfo benzyl GSL: change to DesulfoGSL (or Desulfo GSL if you prefer a space).

Response: Thank you for your suggestion. Based on you suggestion, we have changed the metabolite names accordingly.

End point: Benzyl GSL, 1-methylethylGSL, 2-methylpropylGSL.

Due to these changes, the Phenylalanine used for transport to the chloroplast should be a separate word, not the same used for the previously discussed pathway. From that new Phenylalanine, there should be an arrow in the white space to “2-keto acid”. Above this arrow BCAT4 and BCAT6 in red exactly as for Methionine in Fig S32a. From this new “2-keto acid” there should be an arrow showing transport over the chloroplast membrane (BAT5 in red above this arrow). The arrow ends in the same name (2-keto acid) now within the chloroplast (as presently). The next three reaction arrows are kept as presently, but above the fourth arrow, symbolizing transport out of the chloroplast, the red “BCAT3” should be deleted. You are right that there are controversial evidence for the possible involvement of BCAT3 in the biosynthesis of the methionine derived GSLs (Figure S32a), but not for the benzyl and 2-phenylethylGSLs.

Response: Thank you for your suggestions. We have revised the figure accordingly.

Also in this Fig S32b, remember editing the curved and straight lines leading from the chloroplast, as explained for Fig S32a.

Response: The figure was revised accordingly.

Figure S42. The biosynthesis of glucosinolates (GSLs) in *Arabidopsis thaliana* includes the methionine-derived pathway and the indolic pathway (a), as well as the benzyl pathway, the branched-chain amino acid pathway, and the 2-phenylethyl pathway (b). The enzymes and transporters are displayed in red.

Minor

Line 29: This line in a way states the obvious. I would rephrase to: .. family) may have caused the unusually broad GSL profile of horseradish.

Response: We rephrased the sentence based on your suggestion.

Line 61. A single GSL and GSLs in general seems to be mixed. I would rephrase to: ... glucosinolates (GSLs), which are hydrolysed by thioglucosidase enzymes to produce isothiocyanates with..

Response: Thank you. We revised the GSL to GSLs based on your comments.

Line 63. Reference 17: is this particular paper suitable as single reference for such vast literature?

Response: Thank you for your suggestion. In the revised manuscript, we added the below three representative academic papers on the cancer preventive properties of isothiocyanates (Wu *et al.*, 2009; Abbaoui *et al.*, 2018; Sundaram *et al.*, 2022).

Line 63 after dot. IF you wish to include sinigrin in the introduction, it could be done with a sentence here, such as: “The dominant GSL in horseradish is sinigrin (allylGSL). It is he precursor of allyl

isothiocyanate, the main responsible for the pungent aroma of horseradish and wasabi.”

Response: Thank you very much for your kind. We have revised this part accordingly.

Line 77: GSLs

Response: The line revised accordingly.

Line 164. It is confusing to variously used tribes and genera here. Should Brassica be Brassiceae?

Response: Thank your very much for your comment. We selected 11 genomes representing major Brassicaceae lineages, i.e., Lineages I, II, III, and IV, and the sister tribe Aethionemeae. Lineage I contained five species from the tribe Cardamineae, Lineage II contained four species from the tribe Brassiceae. We changed “*Brassica*” to Brassiceae in the revised manuscript version.

Line 183: I wondered: what is the definition of “gene family” used here? In general biochemistry, as far as this reviewer knows, gene families are common to multiple species, genera, families and orders, and not usually species specific.

Response: Thank you very much for your question. The term "gene family" is referred to as "orthogroup." We used Orthofinder, a popular software tool, to cluster genes across various species (Emms & Kelly, 2015, 2019). This clustering process led to the identification of orthogroups and orthologs. An orthogroup is the set of genes that descend from a single gene in the last common ancestor of all species considered (Chen *et al.*, 2006; Altenhoff *et al.*, 2011; Powell *et al.*, 2014; Zdobnov *et al.*, 2021). Here, an orthogroup by definition contains both orthologs and paralogs, and in this context is frequently used as a unit of comparison in comparative genomics (Wapinski *et al.*, 2007; Waterhouse *et al.*, 2011; Simola *et al.*, 2013). Species-specific orthogroups always represent the lineage-specific divergence or orphan genes. For example, in the paper of the updated OrthoFinder, the authors presented results for ten metazoan species and identified a large number of species-specific orthogroups (Emms & Kelly, 2019).

Given that the term "gene family" is more widely utilized and has been commonly used to refer to "orthogroups" in numerous studies, we have included this clarification in the main text. Furthermore, we have provided a detailed explanation of the term "orthogroup" in the Methods section to ensure a comprehensive understanding of its definition and usage in this study.

Line 353-362: I find this section a bit unprecise. I would put it as follows: “In the tribe C., most of the known biosynthetic groups of GSLs are well-established (16, 36-38). [Move this sentence to here: Three independent steps (1) (2) (3) ... (37)]. Although the GSL profile of horseradish is dominated by sinigrin, the entire profile is remarkably wide, including most of the biosynthetic groups of the tribe (36, 39, 41). Three groups of horseradish GSLs include chain elongation: short chain methionine-derived, long chain methionine-derived, and chain-elongated phenylalanine derived. A further two groups occur independent of chain elongation: tryptophan derived (“indole GSLs”) and the combined group of benzyl GSL and branched chain GSLs, which were recently discovered to depend on a committed step catalyzed by CYP79C enzymes (36, 90). The unusual biosynthetic diversity suggests a similarly complex array of biosynthetic genes. We reconstructed the GSL.... (keep the existing text from here)

Response: Thank you for your feedback on this section. We have carefully considered your suggestions and have revised the text accordingly to make it more precise and clear.

Line 376-383: I think this bit is imprecise at places, and suggest the following:

GSL names: remember that in chemical names, hyphens are ONLY used before and after numbers or atomic symbols, and that atomic symbols in names should be in italics. Line 376 O-methyltransferase: italic O.

Response: Thank you for your note. We have changed the chemical names accordingly.

Line 376-377: methoxylation is catalyzed in two steps, first a hydroxylation, then a methylation. Reword e.g. as follows: “The final step in indole GSL methoxylation is methylation catalyzed by indole GSL O-methyl transferases (IGMTs). IGMT 1-4 produce 4-methoxyindol-3-ylmethylGSL (4MOI3M) (43), while IGMT 5 produces 1-methoxyindol-3-ylmethylGSL (1MOI3M) (Pfalz et al., 2016 Plant Physiology). The syntelogs of”

Response: Thank you for your suggestion. As mentioned above, we have revised our conclusions regarding the loss of *IGMTs*. We have also analyzed in more detail the differential expression of *IGMTs* across two subgenomes. We have incorporated your suggestions to provide a more accurate description of *IGMTs* functionality. We have reworded the text in the revised manuscript (lines 464-466 in the revised manuscript).

Line 382-383: The references are only on horseradish, but anyway I don't see the literature stating a particularly low level of indoles in *Barbarea*. I would reword the last sentence as this: The loss of IGMT 5 may explain the absence of 1MOI3M in horseradish and its presence in *Barbarea* (36, 39).

Response: Thank you for your suggestion. As we have revised our conclusion regarding the loss of *IGMTs* in horseradish based on the new genome assembly, we removed the corresponding content accordingly.

Line 404-405: I would insert “. Possibly” as follows: “... substrates in horseradish. Possibly, the differences in substrates led to.....”

Response: Revised accordingly.

Line 423: add “agronomists” or “breeders”?

Response: Revised accordingly.

Line 466-467: I suggest a slight adjustment to “.. Cardamineae, e.g. the ancient loss of TGGs and very recent loss of IGMT5. The evolution ... GH1 family may reflect the diversity...”

Response: Thank you. Based on recent results, we have thoroughly investigated and revised the loss of these two genes. We sincerely apologize for any changes in our conclusions that have resulted from the improvement of the genome assembly.

References

- Abbaoui B, Lucas CR, Riedl KM, Clinton SK, Mortazavi A. 2018.** Cruciferous Vegetables, Isothiocyanates, and Bladder Cancer Prevention. *Molecular Nutrition and Food Research*. **62**: e1800079.
- Altenhoff AM, Schneider A, Gonnet GH, Dessimoz C. 2011.** OMA 2011: Orthology inference among 1000 complete genomes. *Nucleic Acids Research* **39**: D289-294.
- Chen F, Mackey AJ, Stoeckert CJ, Roos DS. 2006.** OrthoMCL-DB: querying a comprehensive multi-species collection of ortholog groups. *Nucleic acids research* **34**: D363-368.
- Emms DM, Kelly S. 2015.** OrthoFinder: solving fundamental biases in whole genome comparisons dramatically improves orthogroup inference accuracy. *Genome Biology* **16**: 157.
- Emms DM, Kelly S. 2019.** OrthoFinder: phylogenetic orthology inference for comparative genomics. *Genome Biology* **20**: 238.
- Powell S, Forslund K, Szklarczyk D, Trachana K, Roth A, Huerta-Cepas J, Gabaldón T, Rattei T, Creevey C, Kuhn M, et al. 2014.** EggNOG v4.0: Nested orthology inference across 3686 organisms. *Nucleic Acids Research* **42**: D231-239.
- Simola DF, Wissler L, Donahue G, Waterhouse RM, Helmkampf M, Roux J, Nygaard S, Glastad KM, Hagen DE, Viljakainen L, et al. 2013.** Social insect genomes exhibit dramatic evolution in gene composition and regulation while preserving regulatory features linked to sociality. *Genome Research* **23**: 1235–1247.
- Sundaram MK, R P, Haque S, Akhter N, Khan S, Ahmad S, Hussain A. 2022.** Dietary isothiocyanates inhibit cancer progression by modulation of epigenome. *Seminars in Cancer Biology*.
- Wapinski I, Pfeffer A, Friedman N, Regev A. 2007.** Natural history and evolutionary principles of gene duplication in fungi. *Nature* **449**: 54–61.
- Waterhouse RM, Zdobnov EM, Kriventseva E V. 2011.** Correlating traits of gene retention, sequence divergence, duplicability and essentiality in vertebrates, arthropods, and fungi. *Genome Biology and Evolution* **3**: 75–86.
- Wu X, Zhou QH, Xu K. 2009.** Are isothiocyanates potential anti-cancer drugs? *Acta Pharmacologica Sinica* **30**: 501–512.
- Xu Z, Escamilla-Treviño LL, Zeng L, Lalgondar M, Bevan DR, Winkel BSJ, Mohamed A, Cheng CL, Shih MC, Poulton JE, et al. 2004.** Functional genomic analysis of Arabidopsis thaliana glycoside hydrolase family 1. *Plant Molecular Biology* **55**: 343–367.
- Zdobnov EM, Kuznetsov D, Tegenfeldt F, Manni M, Berkeley M, Kriventseva E V. 2021.** OrthoDB in 2020: Evolutionary and functional annotations of orthologs. *Nucleic Acids Research* **49**: D389–D393.

Reviewer #2 (Remarks to the Author):

The authors in this newest version of the manuscript have responded to all my questions in the previous review and I am satisfied with most of the answers and revisions. However, I have still felt that it is sometimes hard to get the main points from many parts of the results, either because of the quite descriptive findings or because the analysis for supporting some conclusions are inadequate. Below are some concerns about this revised manuscript.

Response: Thank you for taking the time to review our manuscript again. We are very grateful for your valuable comments, which have contributed greatly to improving the quality of our manuscript. We have carefully considered all the issues and suggestions you raised. Based on your feedback, we have made fundamental revisions in two main aspects: 1) further improving the quality of the genome assembly; 2) providing more compelling evidence for some of the conclusions criticized in your review

Regarding the genome quality, the previous version of the genome relied mainly on ONT-based sequences. Although a number of data indicators indicated high genome quality, there was still a risk of missing genome fragments (Figures S6-S7). To address these concerns, we added HiFi-based assembly locally while maintaining the ONT-based assembly as the basic framework. The PacBio HiFi sequencing platform provided us with long sequencing reads with ultra-high accuracy, which is an effective solution to overcome chimeric sequences and repetitive regions in genome assemblies. To further improve the genome quality and address concerns raised by both Reviewers, we combined PacBio HiFi reads, ONT-based sequences, and HiC-seq data. As a result, we successfully constructed a T2T gap-free genome assembly in horseradish, the first T2T gapless assembly of an allopolyploid genome in the Brassicaceae family.

We have noted your concerns, particularly regarding the analysis of repetitive sequences, especially LTR retrotransposons. We realized that the limited number of words negatively affected the interpretation of some findings. In the current version, we have made significant improvements by providing comprehensive supplementary explanations (Supplementary Notes) and presenting additional evidence, specifically addressing the areas of your concern. We hope that our effort will meet your expectations and welcome your potential feedback to further improve the overall quality of the article. We have addressed your specific concerns point by point as follows.

Major concerns:

1. Inadequate information for Hi-C scaffolding in Figure 1f. It seems many small segments of the chromosome do not show any Hi-C contact information with any other part of the genome. The Hi-C contact maps for each chromosome couldn't be found in the supplementary figures for checking the details. The authors should carefully check those segments. In addition, they should give a complete Hi-C contact map of all the estimated genome including those unanchored contigs for checking the quality of scaffolding process.

Response: Thank you for raising this point. We also noted the presence of "blank regions" in the Hi-C interaction maps, where there is no Hi-C contact information with another part of the genome

(Figure S3, the former Figure 1f). Upon closer examination, we found that these regions contain a high density of repetitive sequences, indicating their possible association with centromeric regions. The presence of abundant repetitive sequences in these regions may result in higher GC contents, which are not favorable for amplification and sequencing on Illumina sequencing platforms (Kozarewa *et al.*, 2009; Chen *et al.*, 2013). Consequently, genomic regions with high GC content may be underrepresented in Illumina sequence data. In addition, the presence of highly repetitive sequences can pose a challenge for read alignment, making it difficult to obtain unique mapping positions. For example, researchers studying the high-quality genomes of kiwifruit (Han *et al.*, 2023; Yue *et al.*, 2023) and faba bean (Jayakodi *et al.*, 2023) obtained similar results and observed the absence of Hi-C signals in centromeric regions and regions with simple repeats.

To further improve the quality of the genome and prove the above conclusion, we performed a more rigorous and careful genome assembly process. Unlike the previous version of the genome, the updated version is combining the advantages of both PacBio HiFi and Nanopore technologies. By assembling the PacBio HiFi reads, we achieved higher contiguity and completeness of the assembly, with a contig N50 of 33.74 Mbp (Table S5). By Hi-C scaffolding, we found that 9 of 16 chromosomes were represented by single contigs (Table S6). The assembly, which was predominantly based on PacBio HiFi reads, served as the backbone, while the gaps were filled with ONT-generated sequences. Ultimately, we obtained a high-quality reference genome at the T2T gap-free level. Subsequent evaluations have confirmed the accuracy of this genome in terms of genome completeness, contiguity, and genome structure.

We further examined the "blank regions" in the Hi-C interaction maps using the T2T-level reference genome. Similarly, we observed regions devoid of any signal in the Hi-C interaction maps (Figure 2e, Figures S10-S13). Plant centromeres are generally composed of repeated sequences, including LTR-RTs and tandem repeats. We combined the Hi-C interaction heatmap and repeat distributions to predict centromeric regions of the horseradish genome (Supplementary Notes 3). Approximate locations of 16 centromeric regions were estimated using the density of repeats and Hi-C interaction heatmap. They ranged from 2.8 to 18.5 Mbp, with an average length of 5.26 Mbp (Table S8, Figures S10-S13). Of note, the "blank regions" observed in the Hi-C map had a dense accumulation of tandem repeats, which was consistent with the T2T genome assemblies of kiwifruit (Han *et al.*, 2023) and faba bean (Jayakodi *et al.*, 2023) genome. In addition, we conducted a detailed analysis of repetitive sequences within the centromeric regions. We observed that the abundant tandem repeat monomers and LTR-RT families contributed to the expansion of centromeric region (Supplementary Notes 3); these results are now detailed in Supplementary Notes.

In conclusion, in order to address the concerns raised by both Reviewers regarding the quality of our genome assembly, we took a comprehensive approach to significantly improve the quality of the sequenced horseradish genome. As a result, we achieved a high-quality genome at T2T gap-free level and we elucidated the nature of white signals observed in Hi-C interaction maps of different plant genomes. Our work could serve as a reference for obtaining similar high-quality T2T genome assemblies in polyploid species of the mustard family.

Figure 2

Genomic architecture and the diversity of repetitive sequences in the horseradish genome.

(a). Summary of the genomic annotation. The “Other” genomic composition includes the non-protein-coding genes, unclassified repeat regions (“No repeat category”), and genomic regions that cannot be annotated as genes/repeats/non-protein-coding genes (“Non-annotated”). (b). Proportion of different repeat types in the 16 horseradish chromosomes (a01 – a08, b01 – b08) and both subgenomes (a and b columns). (c). The insertion times of different LTR retrotransposons. (d). Phylogenetic tree of Ty3/Gypsy type LTR-RTs. I. The Hi-C interactions in chromosome a04. (f). The density of LTR-RTs (blue) and tandem repeats (red) within the window size of 100 kbp along chromosome a04. (g). The structure of the centromeric region of the chromosome a04. A 500 bp window of 500 bp was used to generate the sequence identify heatmap and methylation levels within each window are also shown.

2. Some of the results lack adequate analysis. Line 151-152, the authors concluded that LTR hotspots are located in centromere regions, but I didn't observe that the authors annotated the centromere regions from the genome.

Besides, simply performing GO and KEGG analysis of genes around LTR retrotransposons (a) and DNA transposons (b) (as shown in Figure S12) does not provide enough evidence to support the conclusion of a possible influence of repeats on gene expression. More examples of these issues are presented in the comments below.

Response: Thank you for bringing this up. We have made additional clarification of these results in the current version. Relevant content has been presented concisely in the main text, while detailed information has been provided in supplementary notes. We recognized that the **identification/annotation of centromeric regions**, and the **potential impact of LTR-RTs on gene and genome diversification** have not yet been explained in detail.

Identification and annotation of centromeric regions:

Based on the T2T gap-free genome assembly, we combined the Hi-C interaction heatmap and repeat distributions to predict centromeric regions (Supplementary Notes 3). Approximate location of 16 centromeric regions was estimated using the density of repeats and Hi-C interaction heatmap.

To elucidate the repetitive elements within the centromeric regions, we clustered monomer elements of identified tandem repeats. The most abundant tandem repeat clusters were identified as the most abundant tandem repeat monomers in centromeric regions (Table S20). We observed that the “blank regions” in Hi-C interaction maps contained high densities of tandem repeats in centromeric regions. Visualization of complex tandem repeats within the centromeric regions revealed that high-order structures are distinct between centromeres of subgenome A and subgenome B (Figure 2g, Figures S17, S18). As for the LTR-RTs in the centromeric regions, we scanned the five LTR-RT families with the highest coverage in the centromeric regions of all chromosomes. We found that the LTR-RT families with the highest coverage were shared among centromeres on different chromosomes of both subgenomes (Table S21), suggesting that these LTR-RTs have undergone amplification and spread into multiple centromeric regions

All detailed results and discussion can be found in Supplementary Notes 3.

The potential impact of LTR-RTs on gene expression and genome diversification:

When examining the influence of intact LTR-RTs, a total of approximately 1,497 genes were found in close proximity (<1,000 bp) to intact LTR-RTs (Figures S16a-16b). **To assess the gene expression levels near the different types of LTR-RTs, we calculated the average expression level and methylation level in the different tissue types (root, stem, and leaf) using the deeptools software with a sliding window size of 50 bp. Our results showed that LTR-RTs, particularly *Gypsy* elements, had a negative impact on the expression levels of nearby genes (Figures S16a-16b).** Consistently, we observed increased methylation levels in the vicinity of LTR-RTs, with the *Gypsy* LTR-RTs having significantly higher methylation levels compared with the *Copia* type elements (Figures S16e-16f). In addition, we quantified the expression of different

retrotransposons and found a higher proportion of LTR-RTs with high expression levels among *Copia* LTR-RTs (Figure S16g). **We therefore propose that LTR-RTs may negatively affect the expression of nearby genes by elevated methylation levels and thus some biological processes.**

In addition, we investigated the impact of fragmented LTR-RTs on nearby genes using a comprehensive strategy (Supplementary Notes 2.5). To investigate the distribution of fragmented LTR-RTs around genes, the occurrence of high-confidence fragmented LTR-RTs with a length greater than 200 bp was calculated across different genic regions, including exons, introns, and upstream regions (-500 bp). The analysis revealed that fragmented LTR-RTs were present in genic regions of 9,336 genes. Specifically, they were found in exons of 1,769 genes, introns of 2,383 genes, and upstream regions of 6,951 genes (Figure S34, Supplementary Notes 2.5). **These results suggest that the structure or expression of a large number of genes could be affected by the presence of fragmented LTR-RTs.**

In conclusion, to address the Reviewer's concerns, we systematically sought to provide additional evidence and examine various aspects, including gene expression, DNA methylation, and transposon activity, in order to elucidate potential impacts of transposons on gene functionality. In addition, we have expanded our analyses to include fragmented LTR-RTs to increase the robustness of our arguments.

Figure S16. Characterization of long terminal repeat retrotransposons (LTR-RTs) and analysis of their nearby gene expression.

a-b, Statistics of the gene numbers near *Copia* (a) and *Gypsy* (b) type LTR-RTs; c-d, Distribution of expression levels near different types of LTR-RTs. The average expression level is based on values obtained by analysis of three different tissues (root, stem and leaf). The calculation was performed using deepTools, and the size of each sliding window was 40 bp. e-f, Distribution of methylation levels near different types of LTR-RTs. The methylation level was the average value of different tissues (root, stem, and leaf). g, Distribution of different types of LTR-RTs expression level. The average expression level is based on values obtained by analysis of three different tissues (root, stem, and leaf). The read count of each LTR-RT locus was obtained using the telescope software. We used the ratio of read counts and the length of LTR-RT to measure the expression level. h-i, The Kyoto Encyclopedia of Genes and Genomes (KEGG) enrichment analysis of the genes nearby different types of LTR-RTs. FPKM: Fragments Per Kilobase of transcript per Million mapped reads.

3. The present manuscript contains numerous errors, some of the numbers in the main text do not

match with the tables and figures provided by the author. And, some of the tables and figures referenced in the main text do not match to the corresponding results. I pointed out many such issues in the subsequent comments. I remembered that this point was highlighted in the previous manuscript review.

Response: Thank you. This is correct. In the current version, we have thoroughly reviewed the results and carefully cross-checked the content referenced in the figures, tables and the main text. To help readers understand the tables, we have provided detailed annotations in the updated version of the tables.

4. The current analysis is still overly descriptive and lacks explanations for the results. For example, the authors mention a significant difference in burst time between *Copia* and *Gypsy* LTRs without providing further elaboration.

Additionally, they note an enrichment of certain TE families based on phylogenetic analysis, but they only make a general conclusion that the rapid expansion of certain families played an important role in genome expansion, without providing any evidence to support this conclusion. I remember that I pointed out these issues in the previous review, but the resubmitted manuscript does not seem to address them adequately.

Response:

Thank you for your constructive comments. In the previous version, due to limited word count and our focus on subsequent sections on genome evolution and its implications for important traits, we indeed neglected the interpretation of the diversity of repetitive sequences, particularly LTR-RTs. In the current version, we performed a comprehensive and systematic analysis of transposable elements as already mentioned above.

For the question about difference in burst time between *Copia* and *Gypsy* LTRs:

We detected recent bursts of different LTR-RT types that occurred within the last 0.2 million years (Figure 2c). We also observed variations in burst time between different LTR-RT types (Figure 2c). The *Copia* retrotransposons had a more recent burst of activity compared with *Gypsy* RTs. **We provided more evidence to elaborate the differences, including the difference of methylation level, and the expression level of LTR-RTs.**

Accordingly, we observed increased methylation levels in the vicinity of LTR-RTs, with *Gypsy* LTR-RTs having significantly higher methylation levels compared with the *Copia* type (Figures S16e-16f). **In addition, we quantified the expression of different retrotransposons and found a higher proportion of LTR-RTs with high expression levels among *Copia* LTR-RTs (Figure S16g). Thus, we proposed that the distinct activity of various types of LTR-RTs results in variations in the burst time.**

For the Question about the expansion of certain families:

To further characterize LTR-RT families and classify the clades of different LTR-RT families, we developed an in-house pipeline called *valcano* (<https://github.com/maypoleflyn/valcano>). Based on the phylogenetic analysis, **we classified full-length LTR-RTs into 1,949 families, with an average of 5.8 members per family (Table S19)**

After that, we used RepeatMasker with a custom library created from clustered LTR-RT family sequences to assess the coverage of each LTR-RT family in the genome. Our results showed that the horseradish repeatome is dominated by specific LTR-RT families with exceptionally high genome coverage (Table S19, Figure S14, Supplementary Notes 2.2). Remarkably, **the top 150 LTR-RT families, ranked by genome coverage, accounted for nearly 50% of the total LTR-RT genome coverage (Figure S15, Table S19)**. All clustered sequences of LTR-RT families can be found in github (<https://github.com/maypolefly/HG>). All detailed descriptions in the revised manuscript can be found in Supplementary Notes 2.2 and the Results section.

Figure S15. Cumulative frequency of the LTR-RT families in the sequenced horseradish genome.

5. The analysis for "lost genes" should be reorganized and more clearly explained. The authors should provide a more compelling explanation for why they consider these genes to be "lost" rather than newly formed on the A and B subgenomes after speciation.

Response: Thank you very much for your suggestion. Theoretically, we should have identified genes that show differences between subgenomes, including **both lost genes and newly formed genes** (orphan genes). However, compared to gene loss or functional degradation, the emergence of orphan genes from non-coding DNA is relatively uncommon (Chandrasekaran & Betrán, 2008; Singh & Wurtele, 2020).

To better demonstrate the status of genes with differences between the subgenomes, in the

current manuscript version, we focused on identifying orphan genes that arose in one or other horseradish subgenome. **We obtained a total of 1,151 genes in 567 orthogroups that were unique to horseradish but absent in other crucifer species. We found that none of the horseradish-specific genes belonged to the identified 2,653 “lost genes”.** This comparison corroborates our earlier conclusions about “lost genes” in the horseradish genome. We admit that the analysis may not have been clearly explained in the previous manuscript version, therefore we have included a detailed explanation in the Methods section (Line 684-691) to address this issue.

Minor concerns:

6. The reference on line 119-120 is incorrect as Table S6 does not provide the mapping ratio of Illumina and Nanopore reads.

Response: Thank you for your question. We have thoroughly reviewed the accuracy of the citations in the latest version. The information regarding the alignment rate can be found in Table S10.

7. I noticed that the mapping ratio of Nanopore reads is 99.55% instead of 95.55%. I suggest that the authors should carefully check the figures displayed in the present manuscript for any inaccuracies.

Response: Thank you for your suggestion. This was correct. In the latest version, we have conducted another thorough check to ensure the accuracy of numbers, including the alignment rate.

8. The circos plot in Figure 1e is not clear. For example, the circular diagram of gene expression appears to have many maximum values and the author does not provide any description in the legend.

Response: Thank you for raising the question. We have redesigned the figure representing the expression levels using a heatmap and provided detailed explanations in the figure caption.

9. Table S14 was not presented in the main text.

Response: Thank you. In the latest manuscript version, we have double checked all figures cited in the main text.

10. Line 97, 24Gbp of Illumina paired-end short reads do not match the 88-fold genome coverage given the reference genome is estimated around 610Mb.

Response: Thank you for this. In the latest version, we have made corrections to the description of Illumina data.

11. Line 148-149, This conclusion is too broad and lacks any results to support it.

Response: Thank you. In the revised manuscript, we provide more evidence to support the hypothesis. We investigated the methylation levels around the different types of LTR-RTs and **observed increased methylation levels in the vicinity of Gypsy LTR-RTs compared to Copia LTR-RTs (Figures S16e-16f). In addition, we quantified the expression of different retrotransposons and found a higher proportion of LTR-RTs with high expression in Copia LTR-RTs (Figure S16g).** All the evidences may reflect the distinct activity of different LTR-RTs.

12. Line 152, This conclusion appears to be based on the author's subjective belief rather than evidence, as there is no supporting data or results that demonstrate the hotspots were located in the

centromere regions.

Response: Thank you. Plant centromeres generally consist of repeated sequences, including long terminal repeat (LTR) retrotransposons and tandem repeats (Ma *et al.*, 2007). Based on the T2T gap-free genome assembly, we combined the Hi-C interaction heatmap and repeat distributions to predict centromeric regions (Supplementary Notes 3). Approximate locations of the 16 centromeric regions were estimated using the density of repeats and Hi-C interaction heatmap, ranging from 2.8 to 18.5 Mbp with an average length of 5.26 Mbp (Table S8, Figures S10-S13). Similar methods were also used in the recent identification of centromeric regions of T2T genome assemblies in kiwifruit (Han *et al.*, 2023; Yue *et al.*, 2023) and watermelon (Deng *et al.*, 2022).

13.Line 153, It is not clear about the ‘certain families’ here.

14.Line 153-154, the authors cannot conclude that the ‘certain families’ played an important role in genome expansion. They didn’t conduct analysis between the ‘certain families’ and genome size.

Response: Thank you for this comment. The two points are related and are actually very similar to Question 2 above. We used RepeatMasker with a custom library created from clustered LTR-RT family sequences to assess the coverage of each LTR-RT family in the horseradish genome. Our findings show that specific LTR-RT families dominate the horseradish repeatome and have exceptionally high genome coverage (Table S19, Figure S15, Supplementary Notes 2.2). It is worth noting that the top 150 LTR-RT families, ranked by genome coverage, account for nearly 50% of the total LTR-RT genome coverage (Table S19). To provide further details, we included a comprehensive list of the LTR-RT families with the highest coverage in Table S19. In addition, all corresponding sequences have been uploaded to GitHub for easy access and reference.

15.Line 155, The Figure S11 does not illustrate that "nearly 3000 genes were predicted to be located in the vicinity" as stated in the text. There is no clear connection or correlation between the results presented in the text and the data displayed in Figure S11.

16.Line 158, There is no data or results provided to demonstrate ‘a possible influence of repeats on gene expression’.

Response: Thank you. As these two points are related, we discuss them together here.

Re Question 15, we have reanalyzed the situation regarding the neighboring genes of LTR-RTs, instead of presenting the quantity of LTR-RTs around genes as before. Detailed results can be found in the updated version of Figure S16.

Re Question 16, to assess the expression levels near different types of LTR-RTs, we calculated the average expression value across various tissues (root, stem, and leaf) using deeptools software with a sliding window size of 50 bp. Our findings revealed that LTR-RTs had a negative impact on the expression levels of nearby genes, particularly *Gypsy* LTR-RTs (Figure S16a-16b). Consistently, we observed increased methylation levels in the vicinity of LTR-RTs, with *Gypsy* LTR-RTs exhibiting significantly higher methylation levels compared with *Copia* type RTs (Figures S16e-16f). Furthermore, we quantified the expression of different retrotransposons and noted a higher proportion of LTR-RTs with high expression of *Copia* LTR-RTs (Figure S16g).

17. L155-158, the study focuses on genes with intact transposable elements (TEs) inserted in the

flanking regions, but it's not clear why the authors choose to limit their analysis in this way. As the intact TE only constitutes a small proportion of the total transposons.

Response: Thanks for your question. Despite the intact transposable elements constitute a small proportion of the genome, intact and active transposable elements are able to amplify within genome, and alter genome structure genetically and epigenetically. The activity level of intact transposable elements can partially reflect the dynamic expansion and genomic changes within a genome. Studying the state and patterns of active transposable elements can provide crucial insights into understanding the formation of the current state of the genome. Moreover, current research on transposon transposition mechanisms and suppression mechanisms primarily focuses on active transposable elements, which could provide valuable references for understanding our observations. Therefore, our study primarily focuses on intact transposable elements, particularly LTR-RTs (Supplementary Notes 2.3).

In the updated manuscript, we also conducted a systematic study on fragmented LTR-RTs to investigate their impact on the upstream regions of genes as well as gene structures. Besides, we also investigated the methylation levels around the fragmented LTR-RTs. Overall, All the observation suggested a potential association between the presence of fragmented LTR-RTs and the observed differential methylation patterns in the subgenomes, indicating their possible role in shaping subgenome-specific methylation profiles. All the detailed description could be found in the supplementary Notes 2.5.

18.Line 174, The authors should indicate it clearly in the Figure 3b, so that the readers can easily understand the information and get the correct interpretation of the figure.

19.Line 175, the result is too general. There is no detail about the existence of segmental duplications contribute to the additional Ks peak.

Response: Thank you for your question. As these two questions (Question 18 and Question 19) are related, we address them together here. In the latest version of the T2T-level assembly, we did not observe the mentioned *Ks* peak. Furthermore, upon further analysis of the earlier assembly results, we found that many of these additional gene peaks were distributed in regions with repetitive sequences. Examination of the Hi-C data revealed the presence of potential assembly chimeras. Therefore, we speculate that the origin of these peaks was likely caused by potential assembly errors. In summary, in the latest version, we have removed these specific results and discussion related to the *Ks* peak.

20.Line 181-182, There is an obvious error here. The reference of Figure 3b should not appear here.

Response: Thank you. We have revised the manuscript including all references to figures.

21.Line 183-184, It appears that there is an error in the text and Figure 3d, where the number of gene families described in the text does not match the number shown in the figure. The figure shows 8208, but the authors described 8205 gene families.

Response: Thank you for pointing out the issue. In the updated version of the manuscript, this was corrected.

22.Line 183-184, Why are 11 species described here? They employed 18 species in the present study.

Response: Thank you for your question. We are focusing here on the gene families specific for species of the Brassicaceae family rather than the gene families that are shared by all 18 species. The conserved gene families shared by all 11 crucifer species could help decipher the evolution of conserved traits in the Brassicaceae family.

23.Line 195-195, There is an obvious error here. The reference of Figure S15-S17 should not appear here.

Response: In Figures S15-S16 (Figures S21-S22 in the latest manuscript version), the collinearity relationship between the horseradish genome, Arabidopsis genome, and ACK is presented, while Figure S17 (Figures S23 in the latest manuscript version) represents a synthesis of information from multiple figures. Therefore, we believe the citation of these figures in the given context is appropriate. To avoid confusion, we have added a reference to the supplementary material section for this part.

24.Line 365-367, it appears that there is a discrepancy in the reference numbers given in the text. The first sentence refers to Table S23, but the second sentence immediately shifts to Table S33-35. This inconsistency makes it difficult to follow the information provided and understand the context.

Response: We apologize for the oversight. In the latest version, we double checked all tables and figures cited in the main text.

25.Line 302-303, the sentence is too general. I only observed a different number of 'lost genes' between A and B subgenomes.

Response: Thank you. By analyzing the positional distribution of lost genes on chromosomes, we have identified differences in the presence of lost genes between different chromosomes and subgenomes. We have provided further explanation and elaboration on this in the current manuscript version. We have added a density plot showing the distribution of lost genes along the chromosomes (Figure S39).

26.Line 317-318, I suggest stating here that there are 3 more genes on the scaffold.

Response: Thank you. Due to the improved genome assembly, we have discovered that all genes are distributed across both subgenomes.

References

Chandrasekaran C, Betrán E. 2008. Origins of New Genes and Pseudogenes. *Nature Education* **1**: 181

Chen YC, Liu T, Yu CH, Chiang TY, Hwang CC. 2013. Effects of GC Bias in Next-Generation-Sequencing Data on De Novo Genome Assembly. *PLoS ONE* **8**: e62856

Deng Y, Liu S, Zhang Y, Tan J, Li X, Chu X, Xu B, Tian Y, Sun Y, Li B, et al. 2022. A telomere-to-telomere gap-free reference genome of watermelon and its mutation library provide important resources for gene discovery and breeding. *Molecular Plant* **15**: 1268–1284.

Han X, Zhang Y, Zhang Q, Ma N, Liu X, Tao W, Lou Z, Zhong C, Deng XW, Li D, et al. 2023. Two haplotype-resolved, gap-free genome assemblies for *Actinidia latifolia* and *Actinidia chinensis*

shed light on the regulatory mechanisms of vitamin C and sucrose metabolism in kiwifruit. *Molecular Plant* **16**: 452–470.

Jayakodi M, Golicz AA, Kreplak J, Fehete LI, Angra D, Bednář P, Bornhofen E, Zhang H, Bousageon R, Kaur S, et al. 2023. The giant diploid faba genome unlocks variation in a global protein crop. *Nature* **615**: 652–659

Kozarewa I, Ning Z, Quail MA, Sanders MJ, Berriman M, Turner DJ. 2009. Amplification-free Illumina sequencing-library preparation facilitates improved mapping and assembly of (G+C)-biased genomes. *Nature Methods* **6**: 291–295

Ma J, Wing RA, Bennetzen JL, Jackson SA. 2007. Plant centromere organization: a dynamic structure with conserved functions. *Trends in Genetics* **23**: 134–139.

Singh U, Wurtele ES. 2020. Genetic novelty: How new genes are born. *eLife* **9**: e55136.

Yue J, Chen Q, Wang Y, Zhang L, Ye C, Wang X, Cao S, Lin Y, Huang W, Xian H, et al. 2023. Telomere-to-telomere and gap-free reference genome assembly of the kiwifruit *Actinidia chinensis*. *Horticulture Research* **10**: uhac264.

Reviewer #3 (Remarks to the Author):

This is a revised manuscript, "The allotetraploid horseradish genome provides insights into subgenome diversification and formation of critical traits". I was reviewer 3 on the original submission and am satisfied that most of my concerns have been addressed. However, reviewer 1 and reviewer 2 had more critical and substantial concerns of the original submission so I bow to their reviews of this revision.

Response: Thank you for the insightful questions you asked earlier. In the updated version of the article, we have focused on improving the quality of the genome. As you can see, we have successfully assembled the first allopolyploid-level T2T reference genome in the Brassicaceae family. Based on this, we have thoroughly addressed the concerns raised by Reviewer 1 and Reviewer 2. We believe that the updated manuscript is a significant improvement over the previous version and hope that it is to your satisfaction.

Reviewers' Comments:

Reviewer #1:

Remarks to the Author:

3rd review horseradish genome Shen et al.

In order to test some surprising characteristics of a previous genome that was almost but not quite complete, the authors have significantly increased the completeness of the genome characterization. This has allowed the important discovery that the mysteriously missing TGGs were in fact there, but had escaped detection in the earlier version of the genome. This progress solidifies the impression of the resource now submitted. Obviously, this changes the manuscript quite a bit, and eliminates the previous surprising lack of TGGs, but this is what science is about, and we now have, I think, a very reliable sequence resource and a highly reliable overview of potential GSL biochemistry in this species. A rare resource of much scientific value.

So the major experimental revision has evidently improved the manuscript dramatically.

Previous versions likewise showed various problems regarding details in GSL biochemistry, they have been solved as far as I can see. In fact, I could find little to complain about despite quite a bit of scrutiny.

The remaining trouble is minor, but here is a list.

Line 35: Abbreviations. This reader would have enjoyed a list of abbreviations in the beginning of the paper.

Line 51 canola, is that *B napus ssp. oleifera* to be specific?

Line 59 GSL biochemistry:

The terms myrosinase, conventionalmyrosinase and atypicalmyrosinase are used a lot in the results section (line 422-472). But they are not explained in the introduction, where the term thioglucosidase is used. This could be fixed in many ways, and the authors can decide. I suggest keeping the discussion as it is and adding something like this in the intro line 66.

First add a section shift, to put HRPs in a section for themselves. In the myrosinase ITC section, add something like: The isothiocyanate forming thioglucosidase enzymes are traditionally termed myrosinases. Myrosinase is a functional term, and several betaglucosidase enzymes are myrosinases, i.e., accepting thioglucosides as substrates. The group of genes first characterized in *A. thaliana*, TGGs (Barth and Jander, 2006 *Plant J.* 46, 549– 562, were termed the classical myrosinases (I think the first user of this terminology was Nakano et al 2017 *Plant journal* 10.1111/tpj.13377). Several other genes with myrosinase activity but characterized later can be termed "atypical myrosinases" simply to indicate that they are yet less well biochemically known than the classical myrosinases (Nakano et al., 2077). The atypical myrosinases include the PEN enzymes (Bednarek et al., 2009 *Science*, 323, 101– 106).

Line 88 versus 89 K-mer or k-mer? (capital k?)

Line 211: names of tribes, should they be in italics or not?

Line 452 should "not" be deleted? Double negation in this sentence.

Line 460: write GS-OX as lower case letters to match the name in figure 8a.

Line 461 format explanation of CYP as the other explanations, e.g. (monooxygenase) in parentheses or perhaps (cytochrome P450 clade 81F)

Legend to figure 8: I would expand with a few explanations as follows: a. simplified diagram of the GSL biosynthesis pathways as elucidated in *Arabidopsis thaliana* of methionine-derived GSLs, indolic GSLs, ...

Reasoning: In Brassica, other gene symbols are known, and in the text (line 406) you specifically point out that you compared the horseradish sequences to *A. thaliana*. Also using methionine derived rather than aliphatic corresponds better to figure 8a and line 399.

Artwork figure 3a: The white font is a bit difficult for reading in the left hand side of the figure, could this be improved some way? Also spelling error Caradmine.

Reference formatting. In many reference titles, species names have not been spelled with capital genus name or italicized, such as 6, 7, 11, 12, 13, 15, 29, 32, 34, 35, 38, 39, 41, 42, 44, 48... please go on yourself for numbers higher than 50.

In some reference titles, there are superfluous capital letters inherited from the use of capital letters by some journals in the titles just for show, I suggest removing the superfluous capital letters in 18, 39, 49... please go on yourself for numbers higher than 50.

Reviewer #2:

Remarks to the Author:

I'm satisfied with the modifications.

Reviewer #1 (Remarks to the Author):

3rd review horseradish genome Shen et al.

In order to test some surprising characteristics of a previous genome that was almost but not quite complete, the authors have significantly increased the completeness of the genome characterization. This has allowed the important discovery that the mysteriously missing TGGs were in fact there, but had escaped detection in the earlier version of the genome. This progress solidifies the impression of the resource now submitted. Obviously, this changes the manuscript quite a bit, and eliminates the previous surprising lack of TGGs, but this is what science is about, and we now have, I think, a very reliable sequence resource and a highly reliable overview of potential GSL biochemistry in this species. A rare resource of much scientific value.

So the major experimental revision has evidently improved the manuscript dramatically.

Previous versions likewise showed various problems regarding details in GSL biochemistry, they have been solved as far as I can see. In fact, I could find little to complain about despite quite a bit of scrutiny.

Response: Thank you again for taking your valuable time to comment on our manuscript. You put forward a series of constructive comments during the review process, which played a huge role in improving the quality of our manuscripts. We are very grateful for your suggestions, especially on GSL, which have improved the professionalism of our manuscripts. According to your new comments, we all accepted and made revisions. Hope you are all satisfied.

The remaining trouble is minor, but here is a list.

Line 35: Abbreviations. This reader would have enjoyed a list of abbreviations in the beginning of the paper.

Response: Thank you for your suggestion. We also think that Abbreviations can better help readers understand some content in the article. However, depending on the format requirements of the journal, we may not be able to provide a list of abbreviations at the beginning of the article. We have added the content you mentioned to the supplementary notes.

Line 51 canola, is that *B napus* ssp. *oleifera* to be specific?

Response: Thank you very much for reminding. We adopted a more common name “rapeseed” for *Brassica napus*.

Line 59 GSL biochemistry:

The terms myrosinase, conventionalmyrosinase and atypicalmyrosinase are used a lot in the results section (line 422-472). But they are not explained in the introduction, where the term thioglucosidase is used. This could be fixed in many ways, and the authors can decide. I suggest keeping the discussion as it is and adding something like this in the intro line 66. First add a section shift, to put HRP's in a section for themselves. In the myrosinase ITC section, add something like: The isothiocyanate forming thioglucosidase enzymes are traditionally termed myrosinases. Myrosinase is a functional term, and several beta-glucosidase enzymes are myrosinases, i.e., accepting thioglucosides as substrates. The group of genes first characterized in *A. thaliana*, known as TGGs, are commonly referred to as classical myrosinases. The group of genes first characterized in *A. thaliana*, TGGs (Barth and Jander, 2006 Plant J. 46, 549– 562, were termed the

classical myrosinases (I think the first user of this terminology was Nakano et al 2017 Plant journal 10.1111/tpj.13377). Several other genes with myrosinase activity but characterized later can be termed “atypical myrosinases” simply to indicate that they are yet less well biochemically known than the classical myrosinases (Nakano et al., 2017). The atypical myrosinases include the PEN enzymes (Bednarek et al., 2009 Science, 323, 101– 106).

Response: Thank you very much for your comments. We agree with your opinion very much, so we added a paragraph to the revised manuscript based on your comments.

Line 88 versus 89 K-mer or k-mer? (capital k?)

Response: Thank you very much for question. We have uniformly used the lowercase form "k-mer" in the revised manuscript.

Line 211: names of tribes, should they be in italics or not?

Response: Thanks very much for your question. The names of tribes should not be italicized and we have corrected.

Line 452 should “not” be deleted? Double negation in this sentence.

Response: Thanks for your suggestion. We have deleted the word “not” in the revised manuscript.

Line 460: write GS-OX as lower case letters to match the name in figure 8a.

Response: Thanks very much for your suggestion. We have changed accordingly.

Line 461 format explanation of CYP as the other explanations, e.g. (monooxygenase) in parentheses or perhaps (cytochrome P450 clade 81F)

Response: Thanks very much for your suggestion. We have changed accordingly: CYP81F1 (cytochrome P450 monooxygenase 81F).

Legend to figure 8: I would expand with a few explanations as follows: a. simplified diagram of the GSL biosynthesis pathways as elucidated in *Arabidopsis thaliana* of methionine-derived GSLs, indolic GSLs, ...

Reasoning: In Brassica, other gene symbols are known, and in the text (line 406) you specifically point out that you compared the horseradish sequences to *A. thaliana*. Also using methionine derived rather than aliphatic corresponds better to figure 8a and line 399.

Response: Thanks very much for your suggestion. We have changed accordingly:

Artwork figure 3a: The white font is a bit difficult for reading in the left hand side of the figure, could this be improved some way? Also spelling error Caradmine.

Response: Thanks very much for your suggestion. We have improved the figure 3a and corrected the spelling error in the revised figure.

Reference formatting. In many reference titles, species names have not been spelled with capital genus name or italicized, such as 6, 7, 11, 12, 13, 15, 29, 32, 34, 35, 38, 39, 41, 42, 44, 48... please go on yourself for numbers higher than 50.

Response: Thanks very much for your suggestion. We have revised accordingly:

In some reference titles, there are superfluous capital letters inherited from the use of capital letters by some journals in the titles just for show, I suggest removing the superfluous capital letters in 18, 39, 49... please go on yourself for numbers higher than 50.

Response: Thanks very much for your suggestion. We have revised accordingly: